# Min-Max Optimisation for Nonconvex-Nonconcave Functions Using a Random Zeroth-Order Extragradient Algorithm

**Amir Ali Farzin**                                                    *amirali.farzin@anu.edu.au*
*School of Engineering*
*Australian National University & CIICADA Lab*

**Yuen-Man Pun**                                                       *yuenman.pun@anu.edu.au*
*School of Engineering*
*Australian National University & CIICADA Lab*

**Philipp Braun**                                                      *philipp.braun@anu.edu.au*
*School of Engineering*
*Australian National University & CIICADA Lab*

**Antoine Lesage-Landry**                                              *antoine.lesage-landry@polymtl.ca*
*Department of Electrical Engineering*
*Polytechnique Montréal, GERAD & Mila*

**Youssef Diouane**                                                    *youssef.diouane@polymtl.ca*
*Department of Mathematics and Industrial Engineering*
*Polytechnique Montréal & GERAD*

**Iman Shames**                                                        *iman.shames@unimelb.edu.au*
*Department of Electrical and Electronic Engineering*
*University of Melbourne & CIICADA Lab*

**Reviewed on OpenReview:** *https://openreview.net/forum?id=1bxY1uAXyr&noteId=DJ5hL7iFY7*

## Abstract

This study explores the performance of the random Gaussian smoothing Zeroth-Order ExtraGradient (ZO-EG) scheme considering deterministic min-max optimisation problems with possibly NonConvex-NonConcave (NC-NC) objective functions. We consider both unconstrained and constrained, differentiable and non-differentiable settings. We discuss the min-max problem from the point of view of variational inequalities. For the unconstrained problem, we establish the convergence of the ZO-EG algorithm to the neighbourhood of an $\epsilon$-stationary point of the NC-NC objective function, whose radius can be controlled under a variance reduction scheme, along with its complexity. For the constrained problem, we introduce the new notion of proximal variational inequalities and give examples of functions satisfying this property. Moreover, we prove analogous results to the unconstrained case for the constrained problem. For the non-differentiable case, we prove the convergence of the ZO-EG algorithm to a neighbourhood of an $\epsilon$-stationary point of the smoothed version of the objective function, where the radius of the neighbourhood can be controlled, which can be related to the $(\delta, \epsilon)$-Goldstein stationary point of the original objective function.

# 1 Introduction

Many min-max problems that arise in modern machine learning are nonconvex-nonconcave, for example, generative adversarial networks (Goodfellow et al., 2014; Gulrajani et al., 2017), robust neural networks (Madry et al., 2018), and sharpness-aware minimisation (Foret et al., 2021). These min-max problems are generally intractable even for computing an approximate first-order locally optimal solution for smooth objective functions (Diakonikolas et al., 2021), thus structural properties have to be imposed in analyses. The existing literature generally follows two approaches to solving nonconvex-nonconcave min-max optimisation: (i) imposing one-sided or two-sided Polyak-Łojasiewicz conditions (Yang et al., 2020) (or Kurdyka-Łojasiewicz for nonsmooth functions (Zheng et al., 2023)) on the min-max problem; or (ii) addressing the problem from the lens of variational inequalities (Diakonikolas et al., 2021; Pethick et al., 2023).

Regardless of either approach, most existing works require access to the gradient of the oracle, which prohibits its use for a wide range of applications. For example, one can only access the input and output of a Deep Neural Network (DNN) instead of the internal configurations (e.g., the network structure and weights) in most real-world systems. Hence, it is more practical to design black-box attacks to DNNs for robustifying them against adversarial examples (Chen et al., 2017). Another example is Automated Machine Learning tasks, where computing gradients with respect to pipeline configuration parameters is infeasible (Wang et al., 2021). Other applications include hyperparameter tuning (Snoek et al., 2012), reinforcement learning (Salimans et al., 2017), robust training (Moosavi-Dezfooli et al., 2019), network control and management (Chen & Giannakis, 2018), and high-dimensional data processing (Liu et al., 2018).

In this paper, we solve possibly NonConvex-NonConcave (NC-NC) min-max problems via Zeroth-Order (ZO) methods from the perspective of Variational Inequalities (VI). Unlike first-order methods, ZO methods only require access to (often noisy) evaluations of the objective function, thus are applicable to problems for which gradients are costly or even impossible to compute (Maass et al., 2021; Salimans et al., 2017; Bottou et al., 2018); also see (Rios & Sahinidis, 2013; Audet & Hare, 2017) for detailed reviews of these frameworks. As far as we are concerned, the literature on solving NC-NC min-max optimisation problems via ZO methods is very sparse. The only works we noticed are (Xu et al., 2023) and (Anagnostidis et al., 2021), which study the unconstrained differentiable nonconvex-Polyak-Łojasiewicz (NC-PL) min-max problem. Our work considers the min-max problem for both the unconstrained and the constrained setting. We assume the existence of a solution to the weak Minty Variational Inequality (MVI) (Diakonikolas et al., 2021) problem and propose a ZO extragradient method to solve it. The word "extra" refers to the extra oracle evaluation needed in each iteration compared to ZO gradient descent ascent. It is shown that our analysis is also applicable to non-differentiable min-max problems, with a convergence guarantee to a Goldstein stationary point.

## 1.1 Contributions

In this paper, we study the possibly nonconvex-nonconcave deterministic min-max problem of the form

$$\min_{x \in \mathcal{X}} \max_{x \in \mathcal{Y}} f(x, y), \tag{1}$$

where $f \colon \mathbb{R}^n \times \mathbb{R}^m \to \mathbb{R}$ is an integrable objective function. The sets $\mathcal{X} \subset \mathbb{R}^n$ and $\mathcal{Y} \subset \mathbb{R}^m$ are assumed to be nonempty, closed, and convex. To solve the problem, we propose a ZO extragradient algorithm based on Gaussian smoothing. While it is common in the ZO optimisation literature based on Gaussian smoothing to sample random directions from the standard normal distribution $\mathcal{N}(0, \mathbf{I})$, in this work, we consider a more general setting introduced in (Nesterov & Spokoiny, 2017, Section 1.3) where a symmetric and positive definite matrix[1] $B = B^\top \succ 0$ is used to define random directions sampled from $\mathcal{N}(0, B^{-1})$. The performance of the algorithm in both the unconstrained and the constrained setting is analysed. For the unconstrained setting, by assuming the existence of a solution satisfying the weak MVI (introduced in Definition 10), we prove the convergence of the algorithm to a neighbourhood of the $\epsilon$-stationary point of $f$ (whose size depends on the variance), in at most $\mathcal{O}(\epsilon^{-2})$ iterations and function evaluations. For the constrained setting, by assuming the existence of a solution satisfying the *proximal weak MVI* (defined in

---

[1]In Nesterov & Spokoiny (2017), $B$ is allowed to be complex valued and thus the Hermitian matrix $B^*$ is used instead of the transpose $B^\top$. Here, we restrict our attention to real-valued matrices.

Definition 11), we show that the algorithm converges to a neighbourhood of the $\epsilon$-stationary point of $f$ (whose size again depends on the variance), in at most $\mathcal{O}(\epsilon^{-2})$ iterations and function evaluations. The size of the neighbourhood of convergence in both settings can be controlled using variance-reduction techniques in the ZO random oracle. In particular, by employing the variance reduction technique described in Algorithm 2, we establish the convergence of the algorithm to an $\epsilon$-stationary point of $f$ in the unconstrained setting within $\mathcal{O}(\epsilon^{-2})$ iterations and requiring $\mathcal{O}(\epsilon^{-4})$ function evaluations. For the constrained setting, we demonstrate convergence to an $\epsilon$-stationary point of $f$ in $\mathcal{O}(\epsilon^{-2})$ iterations, with $\mathcal{O}(\epsilon^{-6})$ function evaluations.

While most of the prior works assume the differentiability of the objective function of the min-max problem, we show that the assumption can be removed by considering a Gaussian smoothed objective function instead. Assuming the existence of a weak MVI solution for a Gaussian smoothed function $f_\mu$ of $f$, we show that the algorithm converges to a variance dependent neighbourhood of the $\epsilon$-stationary point of $f_\mu$ in at most $\mathcal{O}(\epsilon^{-2})$ iterations and function evaluations, implying convergence to an $\epsilon$-Goldstein stationary point of $f$ (defined in Definition 7). Gaussian smoothing of a function is discussed in (10). As in the smooth setting, using the variance reduction technique in Algorithm 2, we show the convergence of the algorithm to an $\epsilon$-stationary point of $f_\mu$ and an $\epsilon$-Goldstein stationary point of $f$ in $\mathcal{O}(\epsilon^{-2})$ iterations and $\mathcal{O}(\epsilon^{-4})$ function evaluations. Note that in our work, across all considered settings, the bounds on the number of iterations do not explicitly depend on the problem dimension.

## 1.2 Related work

**ZO min-max optimisation:** ZO methods provide a key for solving a host of min-max optimisation problems in which gradient information is not accessible; see, e.g., (Chen et al., 2017; Wang et al., 2021; Snoek et al., 2012; Salimans et al., 2017; Moosavi-Dezfooli et al., 2019). A vast majority of existing literature on ZO min-max optimisation focuses on solving convex-concave or convex-nonconcave min-max problems. For example, Wang et al. (2023) addresses the deterministic and stochastic nonconvex-strongly concave min-max optimisation problem with constraints only on the maximiser. The authors solve the deterministic optimisation problem using ZO gradient descent ascent and ZO gradient descent multi-step ascent methods, both sampled from Gaussian random oracles, and in the deterministic case, prove the convergence of their methods to an $\epsilon$-stationary point in $\mathcal{O}(d\epsilon^{-2})$ and in $\mathcal{O}(d\log(\epsilon^{-1})\epsilon^{-2})$ iterations ($d$ is the problem dimension), respectively. Liu et al. (2020) considers the constrained nonconvex-strongly concave min-max problem and solves it using a ZO projected gradient descent ascent method with uniform sampling vectors. The method is shown to converge to a neighbourhood of an $\epsilon$-stationary point in $\mathcal{O}(\epsilon^{-2})$ iterations.

ZO methods are also developed for stochastic min-max optimisation problems with similar problem structures. Xu et al. (2020) proposes a ZO variance-reduced gradient descent ascent method based on Gaussian sampling vectors for solving the (deterministic or stochastic) unconstrained differentiable nonconvex-strongly concave min-max optimisation problems. The algorithm is proved to converge to an $\epsilon$-stationary point of the objective function in $\mathcal{O}(d\epsilon^{-3})$ iterations. Later, Huang et al. (2022) developed an accelerated ZO momentum descent ascent method based on uniform smoothing estimators for solving stochastic nonconvex-strongly concave min-max optimisation problems, which has been shown to converge to an $\epsilon$-stationary point in $\mathcal{O}(d^{3/4}\epsilon^{-3})$ iterations.

To the best of our knowledge, the only existing works on ZO NC-NC min-max optimisation are (Xu et al., 2023) and (Anagnostidis et al., 2021). In (Xu et al., 2023), the authors study min-max problems for stochastic and deterministic unconstrained differentiable *nonconvex-Polyak-Łojasiewicz* min-max problems using a uniform smoothing random oracle. The authors prove convergence of their approach to an $\epsilon$-stationary point. The authors use ZO alternating gradient descent ascent and ZO variance-reduced alternating gradient descent ascent algorithms and, respectively, in the deterministic case, prove convergence of their approaches to an $\epsilon$-stationary point in $\mathcal{O}(d\epsilon^{-2})$ and $\mathcal{O}(d^2\epsilon^{-2})$ iterations. The authors in (Anagnostidis et al., 2021) consider stochastic unconstrained differentiable *nonconvex-Polyak-Łojasiewicz* min-max problems. They use the direct-search method and prove the convergence of their approaches to an $\epsilon$-stationary point in $\mathcal{O}(\log(\epsilon^{-1})\epsilon^{-2})$ iterations. In this work, we study the class of NC-NC min-max problems for which there exists a solution satisfying the weak MVI, which has been shown to be satisfied for a large class of functions including all min-max problems with objectives that are bilinear, pseudo-convex-concave, quasi-convex-

concave, and star-convex-concave (Diakonikolas et al., 2021), and all unconstrained variationally coherent problems studied in, e.g., Mertikopoulos et al. (2019) and Zhou et al. (2017).

**Variational inequalities:** Finding solutions to VIs is equivalent to finding a first-order Nash equilibrium of the min-max problem (Facchinei, 2003; Song et al., 2020). In particular, a VI with a monotone operator, which has been well investigated, provides a framework in studying convex-concave min-max problems (Nemirovski, 2004). Researchers have spent efforts in reducing the assumption on the monotonicity of the operator, so as to include a larger class of applicable functions. Dang & Lan (2015) focuses on a class of VI problems, referred to as generalised monotone VI problems, that covers both monotone and pseudo-monotone VI problems. Their work discusses a generalised non-Euclidean extragradient method and proves its convergence in $\mathcal{O}(\epsilon^{-2})$ iterations. Song et al. (2020) uses an optimistic dual extrapolation algorithm and proves its convergence to a strong solution in $\mathcal{O}(\epsilon^{-2})$ iterations when the existence of a weak solution is assumed.

Diakonikolas et al. (2021) introduces a class of problems with weak MVI solutions to solve the smooth unconstrained NC-NC min-max problem, which is a weaker assumption than the existence of a weak solution to the VI problem. The assumption is shown to be satisfied by quasiconvex-concave or starconvex-concave min-max problems, and the problems for which the operator $F(x,y) = \begin{bmatrix} \nabla_x f(x,y) \\ -\nabla_y f(x,y) \end{bmatrix}$ is negatively comonotone (Bauschke et al., 2021) or positively cohypomonotone (Combettes & Pennanen, 2004). The authors proposed an extragradient algorithm in an unconstrained setup and proved its convergence to an $\epsilon$-stationary point in $\mathcal{O}(\epsilon^{-2})$ iterations. Later, Pethick et al. (2023) addresses the constrained NC-NC min-max problem. The paper assumes the existence of a solution to the weak MVI with a less restricted parameter range and proposes a new extragradient-type algorithm with fixed and adaptive step sizes. The algorithm is proved to converge to a fixed point in $\mathcal{O}(\epsilon^{-2})$ iterations.

To our knowledge, no previous work has considered solving the min-max problem (1) that satisfies the weak MVI using ZO random oracles.

**Non-differentiable min-max optimisation:** Gradient information is needed when studying first-order min-max optimisation problems, hence non-differentiable min-max optimisation has barely been discussed in the literature. However, because a Gaussian smoothed function always has a Lipschitz continuous gradient as long as the function is itself Lipschitz (Nesterov & Spokoiny, 2017), it hints that ZO smoothing methods may provide a tool to circumvent the computational difficulty caused by the non-differentiability of the objective function. Indeed, Gu & Xu (2024) considers a non-differentiable convex-concave problem and approximates the gradient by taking the average of finite differences of random points in a neighbourhood of the iterate with uniformly sampled vectors. It is proved that the algorithm converges to an $\epsilon$-optimal point in $\mathcal{O}(d\epsilon^{-2})$ iterations. Qiu et al. (2023) considers a non-differentiable nonconvex-strongly concave federated optimisation problem. The authors use a ZO federated averaging algorithm based on sampling from a unit ball and prove the convergence to an $\epsilon$-stationary point of the uniformly smoothed function in $\mathcal{O}(d^8\epsilon^{-2})$ iterations.

**Goldstein subdifferential in ZO optimisation:** The Goldstein subdifferential (defined in Definition 6) has been used in studying the stationarity of a non-differentiable function (Goldstein, 1977). Lin et al. (2022) shows that the gradient of a uniformly smoothed function is an element of the Goldstein subdifferential. The authors then proposed a gradient-free method for solving non-smooth nonconvex minimisation problems and proved its convergence to a $(\delta,\epsilon)$-Goldstein stationary point at a rate of $\mathcal{O}(d^{3/2}\delta^{-1}\epsilon^{-4})$ where $d$ is the problem dimension. Similar convergence results of ZO uniform smoothing methods to a Goldstein stationary point can also be found in the non-smooth nonconvex minimisation literature (Kornowski & Shamir, 2024; Rando et al., 2024). Concurrently, Lei et al. (2024) studies the convergence of a ZO Gaussian smoothing method for a class of locally Lipschitz functions called sub-differentially polynomially bounded functions. It is shown that the gradient of the Gaussian smoothed function lies in a neighbourhood of a Goldstein subdifferential. These results allow us to quantify the stationarity of a solution in a non-differentiable min-max problem.

**Outline:** The paper is organised as follows. Preliminaries and the proposed framework are introduced in Section 2. In Section 3, the main convergence and complexity results related to the proposed algorithm are presented for different settings. Section 4 offers illustrative examples. Lastly, we conclude our paper and

discuss potential future research directions in Section 5. Auxiliary lemmas, proofs of the main theorems, and complementary material can be found in the appendix.

**Notation:** For symmetric positive definite real matrices $B_1 = B_1^\top \succ 0$, $B_2 = B_2^\top \succ 0$, $B_1 \in \mathbb{R}^{n \times n}$, $B_2 \in \mathbb{R}^{m \times m}$, and $B = \begin{bmatrix} B_1 & 0 \\ 0 & B_2 \end{bmatrix}$, we define $n$-, $m$- and $d$-dimensional normed spaces

$$\mathbf{X} = (\mathbb{R}^n, \|x\| = \langle x, B_1 x \rangle^{\frac{1}{2}}), \qquad \mathbf{Y} = (\mathbb{R}^m, \|y\| = \langle y, B_2 y \rangle^{\frac{1}{2}}), \quad \text{and} \quad \mathbf{Z} = (\mathbb{R}^d, \|z\| = \langle z, B z \rangle^{\frac{1}{2}}), \quad (2)$$

where $d = n + m$, respectively. The dual spaces of $\mathbf{X}$, $\mathbf{Y}$, and $\mathbf{Z}$ are denoted by $\mathbf{X}^*$, $\mathbf{Y}^*$, and $\mathbf{Z}^*$, respectively. For all $z \in \mathbf{Z}^*$, the dual norm is denoted by $\|z\|_* = \langle z, B^{-1} z \rangle^{\frac{1}{2}}$, i.e., $\mathbf{Z}^* = (\mathbb{R}^n, \|z\|_* = \langle z, B^{-1} z \rangle^{\frac{1}{2}})$ Additionally, for the space $\mathbf{Z}$, we use the weighted norm

$$|z| \overset{\text{def}}{=} \|z\|_{B^{-1}} = \langle z, B^{-1} B z \rangle^{\frac{1}{2}} = \langle z, z \rangle^{\frac{1}{2}}, \tag{3}$$

which reduces to the standard Euclidean norm in $\mathbb{R}^n$ and specialises to the absolute value of a real number in the scalar case.

The smallest eigenvalue, the largest eigenvalue, and the condition number of the positive definite matrix $B \succ 0$ are denoted by $\underline{\lambda}$, $\overline{\lambda}$, and $\kappa = \frac{\overline{\lambda}}{\underline{\lambda}}$, respectively, and it holds that

$$\underline{\lambda}|z|^2 \leq \|z\|^2 \leq \overline{\lambda}|z|^2 \tag{4}$$

$$\overline{\lambda}^{-1}|z|^2 \leq \|z\|_*^2 \leq \underline{\lambda}^{-1}|z|^2. \tag{5}$$

The ceiling function is denoted by $\lceil \cdot \rceil$, i.e., for $x \in \mathbb{R}$, $x \geq 0$, $\lceil x \rceil = \min\{N \in \mathbb{N} | n \geq x\}$. The projection operator onto a closed convex set $\mathcal{Z} \subset \mathbf{Z}$, is defined as

$$\text{Proj}_{\mathcal{Z}}(\bar{z}) \overset{\text{def}}{=} \arg\min_{z \in \mathcal{Z}} \|z - \bar{z}\|^2. \tag{6}$$

The convex hull of a set of points $S \subset \mathbf{Z}$ is denoted by $\text{conv}(S)$. Let $\mathbb{B}_\delta(z)$ be the closed ball in $\mathbf{Z}$ with centre $z$ and radius $\delta$. The expectation operator with respect to a random variable $u$ is denoted by $E_u[\cdot]$. For $k \in \mathbb{N}$, $u_k \in \mathbf{Z}$, we denote by $\mathcal{U}_k = (u_1, \ldots, u_k)$ a set comprising of independent and identically distributed random vectors. The conditional expectation over $\mathcal{U}_k$ is denoted by $E_{\mathcal{U}_k}[\cdot]$. The identity matrix of appropriate dimension is denoted by $\mathbf{I}$. The diameter of a set $\mathcal{Z} \subset \mathbf{Z}$ is denoted by $D_z$ and is equal to $\sup\{\|z_1 - z_2\| : z_1, z_2 \in \mathcal{Z}\}$. The Minkowski sum of two sets $\mathcal{A}, \mathcal{B} \subset \mathbf{Z}$ is denoted by $\mathcal{A} + \mathcal{B} = \{a + b \mid a \in \mathcal{A}, b \in \mathcal{B}\}$.

## 2 Preliminaries, Problem of Interest, and Algorithm

In this section, we provide the preliminaries for different classes of functions used in this paper. Moreover, the definitions for $\epsilon$-stationary points, generalised gradients, and $(\delta, \epsilon)$-Goldstein stationary points are given. We define different classes of VIs and explain how they are related to min-max problems. Finally, definitions related to Gaussian smoothing ZO oracles are provided, and the main algorithm discussed in this paper is introduced.

### 2.1 Preliminaries and Problem of Interest

For simplicity of notation, we use the definitions $d = n + m \in \mathbb{N}$, $\mathcal{Z} = \mathcal{X} \times \mathcal{Y} \subset \mathbf{Z}$ ($\mathcal{X} \subset \mathbf{X}$, $\mathcal{Y} \subset \mathbf{Y}$ and $\mathbf{Z} = \mathbf{X} \times \mathbf{Y}$), and $z = (x, y)$ to write $f(z) = f(x, y)$ in cases where the properties of function $f$ in (1) are important but the individual components $x$ and $y$ are not.

Regularity of the objective function $f$ in (1) is essential for optimisation algorithms to have convergence guarantees (Nesterov et al., 2018). The Lipschitz continuity, as defined below, is the first of such conditions. We introduce other necessary properties later in this section.

**Definition 1** (Lipschitz continuity). *Let $f : \mathbf{Z} \to \mathbb{R}$ be a continuous function. Then, $f$ is said to be globally Lipschitz if there exists a Lipschitz constant $L_0(f) > 0$ such that*

$$|f(z_1) - f(z_2)| \leq L_0(f)\|z_1 - z_2\| \qquad \forall\ z_1, z_2 \in \mathbf{Z}.$$

*If there exists a positive $\tilde{\delta}$ where the above inequality is satisfied for any $z_1 \in \mathbf{Z}$ and $z_2$ where $\|z_1 - z_2\| \leq \tilde{\delta}$, then $f$ is said to be locally Lipschitz.*

*Moreover, if $f$ is a continuously differentiable function, then the gradient of $f$ is said to be globally Lipschitz if there exists a Lipschitz constant $L_1(f) > 0$ such that*

$$\|\nabla f(z_1) - \nabla f(z_2)\|_* \leq L_1(f)\|z_1 - z_2\| \qquad \forall\ z_1, z_2 \in \mathbf{Z}. \tag{7}$$

Finding the global minimum of a nonconvex optimisation problem, if it exists, is NP-hard (Nemirovskij & Yudin, 1983) and it is known that finding a global saddle point (or Nash equilibrium) of an NC-NC function $f$ is in general intractable (Murty & Kabadi, 1987). Thus, in this paper, instead of finding the saddle points of (1), we mainly focus on finding stationary points of $f$ as described in the following problem statement.

**Problem 1.** *Consider a function $f : \mathcal{Z} \to \mathbb{R}$ along with a nonempty closed convex set $\mathcal{Z} \subset \mathcal{X} \times \mathcal{Y}$. Find the stationary points of $f$.*

In what follows, we discuss various ways of characterising the stationary points of a function under different smoothness conditions.

We start by defining the stationary points of continuously differentiable functions.

**Definition 2** (Stationary points). *For a continuously differentiable function $f$, $z_0 \in \mathbf{Z}$ is a stationary point of $f$ if $\nabla f(z_0) = 0$.*

Similarly, under the same assumptions on $f$, one can define $\epsilon$-stationary points through the condition $\|\nabla f(z_0)\|_* \leq \epsilon$ for $\epsilon \geq 0$. A general definition of $\epsilon$-stationary points is presented below.

**Definition 3** ($\epsilon$-stationary points (Liu et al., 2024)). *Let $f : \mathcal{X} \times \mathcal{Y} \to \mathbb{R}$ be a continuously differentiable function, where $\mathcal{X} \subset \mathbf{X}$ and $\mathcal{Y} \subset \mathbf{Y}$ are nonempty closed convex sets and let $h_1$ and $h_2$ denote positive constants. Then, a point $(x_0, y_0) \in \mathcal{X} \times \mathcal{Y}$ is an $\epsilon$-stationary point of $f$ if $\|\tau(x_0, y_0)\|_* \leq \epsilon$, where*

$$\tau(x_0, y_0) \overset{def}{=} \begin{bmatrix} \frac{1}{h_1}(x_0 - \mathrm{Proj}_{\mathcal{X}}(x_0 - h_1 \nabla_x f(x_0, y_0))) \\ \frac{1}{h_2}(y_0 - \mathrm{Proj}_{\mathcal{Y}}(y_0 + h_2 \nabla_y f(x_0, y_0))) \end{bmatrix}.$$

Recall that the projection operator $\mathrm{Proj}_{\mathcal{X}}(\cdot)$ is defined in (6) in the notation section. We can further extend the definition of stationary points for the case where $f$ is not necessarily continuously differentiable, termed $(\delta, \epsilon)$-Goldstein stationary points. To this aim, we first need to define generalised directional derivatives and generalised gradients (Clarke, 1975).

**Definition 4** (Generalised directional derivative). *Let $f : \mathbf{Z} \to \mathbb{R}$ be a locally Lipschitz continuous function. Given a point $z \in \mathbf{Z}$ and a direction $v \in \mathbf{Z}$, the generalised directional derivative of function $f$ is given by $f^\circ(z; v) \overset{def}{=} \limsup_{z' \to z,\ t\downarrow 0} \frac{1}{t}(f(z' + tv) - f(z'))$. The generalised gradient of $f$ is defined as the set*

$$\partial f(z) \overset{def}{=} \{g \in \mathbf{Z} : \langle g, v \rangle \leq f^\circ(z; v),\ \forall v \in \mathbf{Z}\}.$$

Rademacher's theorem guarantees that any Lipschitz continuous function is differentiable almost everywhere (that is, non-differentiable points are of Lebesgue measure zero) (Evans, 2018). Hence, for any Lipschitz continuous function $f$, there is a simple way to represent the generalised gradient $\partial f(z)$,

$$\partial f(z) = \mathrm{conv}\left(\left\{g \in \mathbf{Z} : g = \lim_{z_k \to z} \nabla f(z_k), \nabla f(z_k) \text{ exists}\right\}\right),$$

which is the convex hull of all limit points of $\nabla f$ over all sequences $(z_k)_{k \in \mathbb{N}}$ such that $z_k \to z$ for $k \to \infty$ and $\nabla f(z_k)$ exists for all $k \in \mathbb{N}$ (Lin et al., 2022). Given the definition of generalised gradients, as a next step towards $(\delta, \epsilon)$-Goldstein stationary points, we need to consider Clarke stationary points (Clarke, 1990).

**Definition 5** (Clarke stationary point). *Given a locally Lipschitz continuous function $f : \mathbf{Z} \to \mathbb{R}$, a Clarke stationary point of $f$ is a point $z \in \mathbf{Z}$ satisfying $0 \in \partial f(z)$. A point $z \in \mathbf{Z}$ is an $\epsilon$-Clarke stationary point if $\min\{\|g\|_* : g \in \partial f(z)\} \leq \epsilon$.*

In Zhang et al. (2020), it is shown that $\epsilon$-Clarke stationary points of a nonsmooth nonconvex function with a fixed $\epsilon \in (0, 1]$ can not be found by any finite-time optimisation algorithm in general. This leads to the definitions of $\delta$-Goldstein subdifferentials and $(\delta, \epsilon)$-Goldstein stationary points.

**Definition 6** ($\delta$-Goldstein subdifferential (Lin et al., 2022)). *Given a point $z \in \mathbf{Z}$ and $\delta \geq 0$, the $\delta$-Goldstein subdifferential of a Lipschitz continuous function $f : \mathbf{Z} \to \mathbb{R}$ at $z$ is given by $\partial_\delta f(z) \stackrel{def}{=}$ $\mathrm{conv}\left(\bigcup_{z' \in \mathbb{B}_\delta(z)} \partial f(z')\right)$.*

The Goldstein subdifferential of $f$ at $z$ is the convex hull of the unions of all generalised gradients at points in a $\delta$-ball around $z$. Accordingly, the $(\delta, \epsilon)$-Goldstein stationary points are defined below.

**Definition 7** ($(\delta, \epsilon)$-Goldstein stationary point). *A point $z \in \mathbf{Z}$ is a $(\delta, \epsilon)$-Goldstein stationary point of a Lipschitz continuous function $f : \mathbf{Z} \to \mathbb{R}$ if $\min\{\|g\|_* : g \in \partial_\delta f(z)\} \leq \epsilon$.*

Note that $(\delta, \epsilon)$-Goldstein stationary points are a weaker notion than $\epsilon$-Clarke stationary points because any $\epsilon$-Clarke stationary point is a $(\delta, \epsilon)$-Goldstein stationary point, but not vice versa. In Zhang et al. (2020), it is shown that the converse holds under the assumption of continuous differentiability and $\lim_{\delta \to 0} \partial_\delta f(z) = \partial f(z)$. Finding a $(\delta, \epsilon)$-Goldstein stationary point in nonsmooth nonconvex optimisation has been shown to be tractable (Tian et al., 2022).

As a next step, we introduce variational inequalities. In particular, instead of solving (1) directly, we find points satisfying these variational inequalities for different operators, which under appropriate continuity assumptions, characterise stationary points of $f$ in the presence of $\mathcal{Z}$ and consequently solutions to Problem 1.

For example, for the case where $f$ is continuously differentiable, the gradient operator of $f$ is defined as

$$F(z) \stackrel{def}{=} \begin{bmatrix} \nabla_x f(x, y) \\ -\nabla_y f(x, y) \end{bmatrix}. \tag{8}$$

Then, a point $z^*$ satisfying Definition 8 below is a stationary point of $f$.

**Definition 8** (Stampacchia variational inequality (Diakonikolas et al., 2021)). *Consider a closed and convex set $\mathcal{Z} \subset \mathbf{Z}$ and an operator $F : \mathbf{Z} \to \mathbf{Z}^*$. Then, we say that $z^* \in \mathcal{Z}$ satisfies the Stampacchia Variational Inequality (SVI) if*

$$\langle F(z^*), z - z^* \rangle \geq 0,$$

*holds for all $z \in \mathcal{Z}$.*

The SVI is in general difficult to verify. Thus, a related and computationally more tractable Minty variational inequality can be used.

**Definition 9** (Minty variational inequality (Diakonikolas et al., 2021)). *Consider a closed and convex set $\mathcal{Z} \subset \mathbf{Z}$ and an operator $F : \mathbf{Z} \to \mathbf{Z}^*$. Then, we say that $z^* \in \mathcal{Z}$ satisfies the Minty Variational Inequality (MVI) if*

$$\langle F(z), z - z^* \rangle \geq 0,$$

*holds for all $z \in \mathcal{Z}$.*

If $F$ is monotone, then every solution to SVI is also a solution to MVI, and the two sets of solutions are equivalent. If $F$ is not monotone, all that can be said is that the set of MVI solutions is a subset of the set of SVI solutions (Kinderlehrer & Stampacchia, 2000). Instead of Definition 9, we will consider a generalisation of MVIs, as discussed in Diakonikolas et al. (2021).

**Definition 10** (Weak Minty variational inequality (Diakonikolas et al., 2021)). *Consider a closed and convex set $\mathcal{Z} \subset \mathbf{Z}$ and a Lipschitz operator $F : \mathbf{Z} \to \mathbf{Z}^*$ with Lipschitz constant $L > 0$. Then, we say that $z^* \in \mathcal{Z}$ satisfies the weak Minty variational inequality if, for some $\rho \in \left[0, \frac{1}{8L\kappa^2}\right)$,*

$$\langle F(z), z - z^* \rangle + \frac{\rho}{2}\|F(z)\|_*^2 \geq 0, \tag{9}$$

*holds for all $z \in \mathcal{Z}$, and where $\kappa$ denotes the condition number of $B$ defined in* (2).

Note that Definition 10 is a generalisation of Definition 9 and it reduces to Definition 9 for $\rho = 0$. For more details, see (Diakonikolas et al., 2021, Section 2.2).

### 2.2 The Zeroth-Order Extragradient Algorithm & Gaussian Smoothing

In this paper, the objective function $f$ in (1) is not necessarily continuously differentiable, or, if $f$ is continuously differentiable, its gradient is not necessarily accessible for computations. For this sake, we will use a function approximation known as Gaussian smoothing (Nesterov & Spokoiny, 2017). Such approximation is continuously differentiable as long as $f$ is integrable. Namely, for a parameter $\mu > 0$, the Gaussian smoothed version of an integrable function $f : \mathbf{Z} \to \mathbb{R}$, is defined as $f_\mu : \mathbf{Z} \to \mathbb{R}$,

$$f_\mu(z) \stackrel{\text{def}}{=} \frac{1}{\psi} \int_{\mathbf{Z}} f(z + \mu u) \mathrm{e}^{-\frac{1}{2}\|u\|^2} \mathrm{d}u, \quad \text{where} \quad \psi \stackrel{\text{def}}{=} \int_{\mathbf{Z}} \mathrm{e}^{-\frac{1}{2}\|u\|^2} \mathrm{d}u = \frac{(2\pi)^{d/2}}{[\det B]^{\frac{1}{2}}}. \tag{10}$$

Here, $u \in \mathbf{Z}$ is sampled from Gaussian distribution $\mathcal{N}(0, B^{-1})$. In (Nesterov & Spokoiny, 2017, Section 2), it is shown that for all $\mu > 0$ and under the assumption that $f$ is integrable, then $f_\mu$ is continuously differentiable. If $f$ is additionally assumed to be globally Lipschitz continuous, then $f_\mu$ is globally Lipschitz continuous with the same Lipschitz constant. The same conclusion can be made with respect to the gradient of the functions $f$ and $f_\mu$.

To approximate the gradient of a function $f$ (for points where the gradient is defined), we define the random oracle $g_\mu : \mathbf{Z} \to \mathbf{Z}^*$ as (Nesterov & Spokoiny, 2017, Section 3)

$$g_\mu(z) = g_\mu(z; B) \stackrel{\text{def}}{=} \frac{f(z + \mu u) - f(z)}{\mu} Bu, \tag{11}$$

where $u \in \mathbf{Z}$ and $B$ are as defined in (2). It is shown in Nesterov & Spokoiny (2017) that $g_\mu$ is an unbiased estimator of $\nabla f_\mu$, i.e., $\nabla f_\mu(z) = E_u[g_\mu(z)]$. The oracle $g_\mu$ allows us to approximate $\nabla f_\mu(z)$ only with function evaluations of the function $f$.

In our proposed framework, we use the simultaneous smoothing for both $x$ and $y$ using a pre-specified smoothing parameter $\mu > 0$, but with independent random vectors $u_1, \hat{u}_1 \in \mathbf{X}$ and $u_2, \hat{u}_2 \in \mathbf{Y}$ sampled from $\mathcal{N}(0, B_1^{-1})$ and $\mathcal{N}(0, B_2^{-1})$. To simplify the notation, we define

$$u \stackrel{\text{def}}{=} \begin{bmatrix} u_1 \\ u_2 \end{bmatrix}, \qquad \hat{u} \stackrel{\text{def}}{=} \begin{bmatrix} \hat{u}_1 \\ \hat{u}_2 \end{bmatrix}, \quad \text{and} \quad B = \begin{bmatrix} B_1 & 0 \\ 0 & B_2 \end{bmatrix}. \tag{12}$$

The common choice of $B$ defined in (12) is the identity matrix $B = \mathbf{I}$, but we will state the main results in Section 3 for the general choice of $B = B^\top \succ 0$. Now that all preliminary definitions have been detailed, we are able to state the zeroth-order extragradient algorithm, as shown in Algorithm 1.

---

**Algorithm 1** Zeroth-Order Extragradient (ZO-EG)

---

1: **Input**: $z_0 = (x_0, y_0) \in \mathcal{Z}$; $N \in \mathbb{N}$; $\{h_{1,k}\}_{k=0}^N, \{h_{2,k}\}_{k=0}^N \subset \mathbb{R}_{>0}$; $\mu > 0$; $B_1 = B_1^\top \succ 0$; $B_2 = B_2^\top \succ 0$
2: **for** $k = 0, \dots, N$ **do**
3:     Sample $\hat{u}_{1,k}$ and $\hat{u}_{2,k}$ from $\mathcal{N}(0, B_1^{-1})$ and $\mathcal{N}(0, B_2^{-1})$
4:     Calculate $G_\mu(z_k)$ using $u = \hat{u}_k$, (14) and (13)
5:     Compute $\hat{z}_k = \text{Proj}_{\mathcal{Z}}(z_k - h_{1,k} G_\mu(z_k))$
6:     Sample $u_{1,k}$ and $u_{2,k}$ from $\mathcal{N}(0, B_1^{-1})$ and $\mathcal{N}(0, B_2^{-1})$
7:     Calculate $G_\mu(\hat{z}_k)$ using $u = u_k$, (14) and (13)
8:     Compute $z_{k+1} = \text{Proj}_{\mathcal{Z}}(z_k - h_{2,k} G_\mu(\hat{z}_k))$
9: **end for**
10: **return** $z_1, \dots, z_N$

---

Algorithm 1 relies on the evaluation of the oracle

$$G_\mu(z) \stackrel{\text{def}}{=} \begin{bmatrix} g_{\mu,x}(z) \\ -g_{\mu,y}(z) \end{bmatrix}, \tag{13}$$

where

$$g_{\mu,x}(z) \stackrel{\text{def}}{=} \frac{f(z + \mu u) - f(z)}{\mu} B_1 u_1 \quad \text{and} \quad g_{\mu,y}(z) \stackrel{\text{def}}{=} \frac{f(z + \mu u) - f(z)}{\mu} B_2 u_2. \tag{14}$$

If we define

$$F_\mu(z) \stackrel{\text{def}}{=} \begin{bmatrix} \nabla_x f_\mu(z) \\ -\nabla_y f_\mu(z) \end{bmatrix} \quad \text{and} \quad \xi(z) \stackrel{\text{def}}{=} G_\mu(z) - F_\mu(z), \tag{15}$$

then from (Nesterov & Spokoiny, 2017, Section 3), it is known that $E_u[\xi(z)] = 0$ for all $z \in \mathcal{Z}$, as $G_\mu(z)$ is an unbiased estimator of $F_\mu(z)$, i.e., with only the evaluations of $f$, we can obtain an unbiased estimation of $F_\mu$. We later use this identity to prove the convergence to a point $\bar{z}$ for which $|F(\bar{z})| \leq \epsilon$ is satisfied.

In Algorithm 1, $z_0$ denotes the initial guess of a stationary point of (1), $\mu > 0$ is the smoothing parameter in (13), (14), $N \in \mathbb{N}$ denotes the number of iterations, and $h_{1,k}$ and $h_{2,k}$ denote positive step sizes for $k \in \{0, \dots, N\}$. The projection steps are only necessary in the constrained case to ensure feasibility, i.e., to ensure that $z_k \in \mathcal{Z}$ for all $k \in \{1, \dots, N\}$.

Algorithm 1 ensures convergence to a variance-dependent neighbourhood of the $\epsilon$-stationary point as shown in the subsequent sections under various assumptions. To reduce the size of the variance-dependent neighbourhood, a variance reduction scheme can be used. Here we use the scheme as outlined in Algorithm 2, which can be found in Balasubramanian & Ghadimi (2022), for example. In Algorithm 1, if in each iteration instead of sampling one $u$ to calculate the corresponding $G_\mu$ defined in (13), we sample $t_k$ directions, then Algorithm 1 changes to Algorithm 2.

---

**Algorithm 2** Variance-Reduced ZO-EG

1: **Input:** $z_0 = (x_0, y_0) \in \mathcal{Z}$; $N \in \mathbb{N}$; $\{h_{1,k}\}_{k=0}^N, \{h_{2,k}\}_{k=0}^N \subset \mathbb{R}_{>0}$; $\mu > 0$; $B_1 = B_1^\top \succ 0$; $B_2 = B_2^\top \succ 0$; $\{t_k\}_{k=0}^N \subset \mathbb{N}$
2: **for** $k = 0, \dots, N$ **do**
3:     Sample $\hat{u}_{1,k}^0, \cdots, \hat{u}_{1,k}^{t_k}$ and $\hat{u}_{2,k}^0, \cdots, \hat{u}_{2,k}^{t_k}$ from $\mathcal{N}(0, B_1^{-1})$ and $\mathcal{N}(0, B_2^{-1})$
4:     Calculate $G_\mu^0(z_k), \cdots, G_\mu^{t_k}(z_k)$ using $u^i = \hat{u}_k^i$, $i = 0, \dots, t_k$, (14) and (13)
5:     Compute $G_\mu(z_k) = \frac{1}{t_k} \sum_{i=0}^{t_k} G_\mu^i(z_k)$
6:     Compute $\hat{z}_k = \text{Proj}_{\mathcal{Z}}(z_k - h_1(k) G_\mu(z_k))$
7:     Sample $u_{1,k}^0, \cdots, u_{1,k}^{t_k}$ and $u_{2,k}^0, \cdots, u_{2,k}^{t_k}$ from $\mathcal{N}(0, B_1^{-1})$ and $\mathcal{N}(0, B_2^{-1})$
8:     Calculate $G_\mu^0(\hat{z}_k), \cdots, G_\mu^{t_k}(\hat{z}_k)$ using $u^i = u_k^i$, $i = 0, \dots, t_k$, (14) and (13)
9:     Compute $G_\mu(\hat{z}_k) = \frac{1}{t_k} \sum_{i=0}^{t_k} G_\mu^i(\hat{z}_k)$
10:    Compute $z_{k+1} = \text{Proj}_{\mathcal{Z}}(z_k - h_2(k) G_\mu(\hat{z}_k))$
11: **end for**
12: **return** $z_1, \dots, z_N$

---

Leveraging this technique the upper bound on the variance of the random oracle will be divided by the number of sampled directions.

Having the stage set up, in the next section, we present the main results. In particular, we analyse the convergence and iteration complexity of Algorithm 1 for three different cases.

## 3 Main Results

In this section, we analyse the convergence and iteration complexity of Algorithm 1 for possibly nonconvex-nonconcave min-max problems. Specifically, Section 3.1 examines the scenario where $f$ is continuously

differentiable and $\mathcal{Z} = \mathbf{Z}$. In Section 3.2, we extend the analysis to the case where $f$ is continuously differentiable but $\mathcal{Z} \neq \mathbf{Z}$ in Problem 1. Finally, Section 3.3 focuses on the case where $\mathcal{Z} = \mathbf{Z}$ and $f$ is non-differentiable. Detailed proofs of the lemmas and theorems are provided in Appendices A and B.

### 3.1 Unconstrained Differentiable Problem

In this subsection, we consider the unconstrained version (1) that corresponds to Problem 1 with $\mathcal{Z} = \mathbf{Z}$. Let us start with the following standard assumption on the variance of the ZO random oracle in the literature of ZO and stochastic optimisation; see, e.g., Maass et al. (2021); Liu et al. (2020); Xu et al. (2020).

**Assumption 1.** *For a fixed $\mu > 0$, the variance of the random oracle $G_\mu(z)$ defined in (13) is upper bounded by $\sigma^2 \geq 0$, i.e.,*

$$E_u[\|G_\mu(z) - F_\mu(z)\|_*^2] \leq \sigma^2, \quad \forall z \in \mathbf{Z}, \tag{16}$$

We assume that Assumption 1 is satisfied throughout the paper. Indeed, a simple calculation shows that

$$
\begin{aligned}
E_u[\|G_\mu(z) - F_\mu(z)\|_*^2] &= E_u[\|G_\mu(z)\|_*^2 + \|F_\mu(z)\|_*^2 - 2\langle G_\mu(z), B^{-1}F_\mu(z)\rangle] \\
&= E_u[\|G_\mu(z)\|_*^2] + \|F_\mu(z)\|_*^2 - 2\|F_\mu(z)\|_*^2 \\
&= E_u[\|G_\mu(z)\|_*^2] - \|F_\mu(z)\|_*^2 \leq E_u[\|G_\mu(z)\|_*^2],
\end{aligned}
\tag{17}
$$

where the first equality follows from expanding the norm, and the second equality holds since $E_u[G_\mu(z)] = F_\mu(z)$. It is shown that $E_u[\|G_\mu(z)\|_*^2] \leq L_0(f)^2(d+4)^2$ for a Lipschitz continuous function $f$ with Lipschitz constant $L_0(f)$ and $E_u[\|G_\mu(z)\|_*^2] \leq \frac{\mu^2}{2}L_1^2(f)(d+6)^3 + 2(d+4)\|F(z)\|_*^2$ for a function $f$ with Lipschitz continuous gradient with constant $L_1(f)$ (Nesterov & Spokoiny, 2017, Theorem 4). Hence, Assumption 1 is not a stringent assumption, particularly when $f$ is Lipschitz continuous or when $f$ has Lipschitz gradients.

**Remark 1.** *Leveraging the variance reduction technique in Algorithm 2, the upper bound on the variance in (16) of the random oracle is*

$$E_u[\|G_\mu(z_k) - F_\mu(z_k)\|_*^2] \leq \frac{\sigma^2}{t_k}.$$

*Thus, by increasing the number of samples $t_k$, the bound on variance decreases. Additionally, note that this variance reduction scheme preserves the property $E_u[G_\mu(z_k)] = F_\mu(z_k)$.*

Next, we need to make an assumption about the existence of a solution for the weak MVI in Definition 10.

**Assumption 2.** *For Problem 1 with $\mathcal{Z} = \mathbf{Z}$, there exists $z^* \in \mathcal{Z}$ such that $F(z)$ defined in (8) satisfies the weak MVI defined in (9).*

Now, we need to analyse the behaviour of $F_\mu$ defined in (15) when Assumption 2 is satisfied. The following lemma presents the properties of $F_\mu$ when Assumption 2 is satisfied.

**Lemma 1.** *Let $f \colon \mathbf{Z} \to \mathbb{R}$ be continuously differentiable with Lipschitz continuous gradient with constant $L_1(f) > 0$. Moreover, let $F_\mu$ be the operator defined in (15) with smoothing parameter $\mu > 0$, and let $\rho$ denote the weak MVI parameter defined in Definition 10. If there exists $z^* \in \mathbf{Z}$ such that Assumption 2 is satisfied, then it holds that*

$$\langle F_\mu(z), z - z^*\rangle + \rho\|F_\mu(z)\|_*^2 + \mu^2 L_1(f)d + \rho\mu^2 L_1^2(f)(d+3)^3 \geq 0, \; \forall z \in \mathbb{R}^d. \tag{18}$$

A proof of Lemma 1 can be found in Appendix A. Using Lemma 1, we can present the main theorem of this subsection. This theorem introduces an upper bound for the average of the expected value of the square norm of the gradient operator of the smoothed function in the sequence generated by Algorithm 1.

**Theorem 1.** *Let $f \colon \mathbf{Z} \to \mathbb{R}$ be continuously differentiable with Lipschitz continuous gradients with constant $L_1(f) > 0$. Let $\sigma^2$ be an upper bound on the variance of the random oracle defined in Assumption 1, $N \geq 0$ be the number of iterations, $F_\mu$ be defined in (15) with smoothing parameter $\mu > 0$, $\mathcal{U}_k = [(u_0, \hat{u}_0), (u_1, \hat{u}_1), \cdots, (u_k, \hat{u}_k)]$, $k \in \{0, \ldots, N\}$, and $\rho$ denotes the weak MVI parameter in Definition 10. Moreover, let $\{z_k\}_{k\geq0}$ and $\{\hat{z}_k\}_{k\geq0}$ be the sequences generated by Algorithm 1, lines 5 and 8,*

*respectively, suppose that Assumption 2 is satisfied, and recall the definitions of the smallest eigenvalue $\underline{\lambda}$, the largest eigenvalue $\overline{\lambda}$ and the condition number $\kappa$ of the positive definite matrix $B$ defined in* (2). *Then, for any iteration $N$, with*

$$h_{1,k} = h_1 \leq \frac{1}{L_1(f)\overline{\lambda}\kappa} \qquad and \qquad h_{2,k} = h_2 \in \left( \sqrt{\frac{2\rho}{L_1(f)\overline{\lambda}\underline{\lambda}\kappa}}, \frac{h_1}{2} \right], \tag{19}$$

*we have*

$$
\begin{aligned}
\frac{1}{N+1} \sum_{k=0}^{N} E_{\mathcal{U}_k}[|F_\mu(\hat{z}_k)|^2] \leq &\frac{2\overline{\lambda}\underline{\lambda}L_1(f)\kappa|z_0 - z^*|^2}{(\overline{\lambda}\underline{\lambda}L_1(f)\kappa h_2^2 - 2\rho)(N+1)} + \frac{2\mu^2\underline{\lambda}L_1(f)d}{(\overline{\lambda}\underline{\lambda}L_1(f)\kappa h_2^2 - 2\rho)} \\
&+ \frac{2\mu^2\underline{\lambda}L_1(f)^2\rho(d+3)^3}{(\overline{\lambda}\underline{\lambda}L_1(f)\kappa h_2^2 - 2\rho)} + \frac{3\underline{\lambda}\sigma^2}{L_1(f)\kappa(\overline{\lambda}\underline{\lambda}L_1(f)\kappa h_2^2 - 2\rho)}.
\end{aligned}
\tag{20}
$$

A proof of Theorem 1 can be found in Appendix B. Given the upper bound provided by Theorem 1, the first right-hand side term of (20) becomes arbitrarily small for $N \to \infty$. The second term, in turns, can become arbitrarily small if $\mu \to 0$. The last term depends on the variance of the random oracle, defined in Assumption 1, which becomes arbitrarily small by using a variance reduction scheme, see Algorithm 2 and Remark 1.

In Theorem 1, the positive definite matrix $B$ defines a degree of freedom, whose eigenvalues and condition number have a significant impact on the step size selection in (19) and on the upper bound in (20). An optimal selection of $B$ in terms of (19) and (20) is out of the scope of this paper. Nevertheless, a discussion based on heuristic arguments is added in Appendix C.1, indicating that an optimal selection of $B$ in Theorem 1 implies $B$ to have a condition number $\kappa = 1$ (but $B$ does not necessarily need to be the identity matrix). Building on the discussion in Appendix C.1, the following corollary restates the result of Theorem 1 for the special case where $B$ is a scalar multiple of the identity matrix. In this setting, the random directions $u_k$ and $\hat{u}_k$ in Algorithm 1 are sampled from $\mathcal{N}(0, \lambda\mathbf{I})$ for some $\lambda > 0$.

**Corollary 1.** *Let the assumptions of Theorem 1 be satisfied with $B = \lambda\mathbf{I}$ for $\lambda > 0$. Then, it holds that*

$$
\begin{aligned}
\frac{1}{N+1} \sum_{k=0}^{N} E_{\mathcal{U}_k}[|F_\mu(\hat{z}_k)|^2] \leq &\frac{2\lambda^2 L_1(f)|z_0 - z^*|^2}{(\lambda^2 L_1(f)h_2^2 - 2\rho)(N+1)} + \frac{2\mu^2\lambda L_1(f)d}{(\lambda^2 L_1(f)h_2^2 - 2\rho)} \\
&+ \frac{2\mu^2\lambda L_1(f)^2\rho(d+3)^3}{(\lambda^2 L_1(f)h_2^2 - 2\rho)} + \frac{3\lambda\sigma^2}{L_1(f)(\lambda^2 L_1(f)h_2^2 - 2\rho)}.
\end{aligned}
$$

The proof of Corollary 1 is the same as the proof of Theorem 1 with $\underline{\lambda} = \overline{\lambda} = \lambda$ and $\kappa = 1$. The next corollary gives a guideline on how to choose the number of iterations and the smoothing parameter $\mu$, for a given specific measure of performance $\epsilon$ and for diagonal matrices $B \succ 0$.

**Corollary 2.** *Let the assumptions of Theorem 1 be satisfied with $B = \lambda\mathbf{I}$ for $\lambda > 0$ and let $r_0 = \|z_0 - z^*\|$. For a given $\epsilon > 0$, if*

$$\mu \leq \left( \frac{(\lambda^2 L_1(f)h_2^2 - 2\rho)}{4\lambda L_1(f)d + 4\lambda L_1^2(f)\rho(d+3)^3} \right)^{\frac{1}{2}} \epsilon \quad and \quad N \geq \left\lceil \left( \frac{4\lambda^2 L_1(f)r_0^2}{(\lambda^2 L_1(f)h_2^2 - 2\rho)} \right) \epsilon^{-2} - 1 \right\rceil,$$

*then,*

$$\frac{1}{N+1} \sum_{k=0}^{N} E_{\mathcal{U}_k}[|F_\mu(\hat{z}_k)|^2] \leq \epsilon^2 + \frac{3\lambda\sigma^2}{(\lambda^2 L_1^2(f)h_2^2 - 2\rho L_1(f))}.$$

A proof of Corollary 2 can be found in Appendix B. Considering Definition 3, to show that the sequence generated by Algorithm 1 converges to an $\epsilon$-stationary point, $\|F(\hat{z}_k)\|$ needs to be bounded. Based on Theorem 1 and Corollary 2, the following corollary introduces an upper bound of the average of the expected value of the squared norm of the gradient operator $F$, defined in (8), over the sequence generated by Algorithm 1.

**Corollary 3.** *Let the assumptions of Theorem 1 be satisfied with $B = \lambda\mathbf{I}$ for $\lambda > 0$ and let*

$$\mu \leq \min\left\{\frac{\epsilon}{\sqrt{2}\lambda L_1(f)(d+3)^{\frac{3}{2}}}, \left(\frac{\lambda^2 L_1(f)h_2^2 - 2\rho}{16\lambda L_1(f)d + 16\lambda L_1^2(f)\rho(d+3)^3}\right)^{\frac{1}{2}}\epsilon\right\} \;\; and \;\; N \geq \left\lceil\left(\frac{8\lambda^2 L_1(f)r_0^2}{\lambda^2 L_1(f)h_2^2 - 2\rho}\right)\epsilon^{-2} - 1\right\rceil.$$

*Then, it holds that*

$$\frac{1}{N+1}\sum_{k=0}^{N} E_{\mathcal{U}_k}[|F(\hat{z}_k)|^2] \leq \epsilon^2 + \frac{6\lambda\sigma^2}{(\lambda^2 L_1^2(f)h_2^2 - 2\rho L_1(f))}.$$

The proof of the Corollary 3 can be found in Appendix B. In light of Corollary 3, it can be seen that the sequence generated by Algorithm 1 is guaranteed to converge to a neighbourhood of the $\epsilon$-stationary points of $f$ in expectation. Additionally, the size of the neighbourhood can be made arbitrarily small using the variance reduction scheme in Algorithm 2. The next theorem introduces an upper bound for the average of the expected value of the square norm of the gradient operator of the smoothed function in the sequence generated by Algorithm 2.

**Theorem 2.** *Let $f\colon \mathbf{Z} \to \mathbb{R}$ be continuously differentiable with Lipschitz continuous gradients with constant $L_1(f) > 0$. Let $\sigma^2$ be an upper bound on the variance of the random oracle defined in Assumption 1, $N \geq 0$ be the number of iterations, $t_k \in \mathbb{N}$ be the number of samples in each iteration of Algorithm 2, $F_\mu$ be defined in (15) with smoothing parameter $\mu > 0$, $\mathcal{U}_k = [(u_0, \hat{u}_0), (u_1, \hat{u}_1), \cdots, (u_k, \hat{u}_k)]$, $k \in \{0, \ldots, N\}$, and $\rho$ denotes the weak MVI parameter in Definition 10. Moreover, let $\{z_k\}_{k\geq 0}$ and $\{\hat{z}_k\}_{k\geq 0}$ be the sequences generated by Algorithm 2, lines 6 and 10, respectively, and suppose that Assumption 2 is satisfied. Then, for any iteration $N$, with*

$$h_{1,k} = h_1 \leq \frac{1}{L_1(f)\overline{\lambda}\kappa} \quad and \quad h_{2,k} = h_2 \in \left(\sqrt{\frac{2\rho}{L_1(f)\overline{\lambda}\underline{\lambda}\kappa}}, \frac{h_1}{2}\right],$$

*we have*

$$\frac{1}{N+1}\sum_{k=0}^{N} E_{\mathcal{U}_k}[|F_\mu(\hat{z}_k)|^2] \leq \frac{2\overline{\lambda}\underline{\lambda}L_1(f)\kappa|z_0 - z^*|^2}{(\overline{\lambda}\underline{\lambda}L_1(f)\kappa h_2^2 - 2\rho)(N+1)} + \frac{2\underline{\lambda}\mu^2 L_1(f)d}{(\overline{\lambda}\underline{\lambda}L_1(f)\kappa h_2^2 - 2\rho)} + \frac{2\underline{\lambda}\mu^2 L_1(f)^2\rho(d+3)^3}{(\overline{\lambda}\underline{\lambda}L_1(f)\kappa h_2^2 - 2\rho)}$$

$$+ \frac{3\underline{\lambda}\sigma^2}{L_1(f)\kappa(\overline{\lambda}\underline{\lambda}L_1(f)\kappa h_2^2 - 2\rho)}\frac{1}{(N+1)}\sum_{k=0}^{N}\frac{1}{t_k}. \tag{21}$$

A proof of Theorem 2 can be found in Appendix B. As in the case of Theorem 1, a discussion on the selection of $B$ can be found in Appendix C.1. The next corollary gives a guideline on how to choose the number of iterations and the smoothing parameter $\mu$, for a given specific measure of performance $\epsilon$ to guarantee convergence to an $\epsilon$-stationary point of the objective function.

**Corollary 4.** *Let the assumptions of Theorem 2 be satisfied with $B = \lambda\mathbf{I}$ for $\lambda > 0$. Moreover, let*

$$t_k = t \geq \left\lceil\frac{18\lambda\sigma^2}{\lambda^2 L_1(f)(L_1(f)h_2^2 - 2\rho)}\epsilon^{-2}\right\rceil, \qquad N \geq \left\lceil\left(\frac{12\lambda^2 L_1(f)r_0^2}{(\lambda^2 L_1(f)h_2^2 - 2\rho)}\right)\epsilon^{-2} - 1\right\rceil, \qquad and$$

$$\mu \leq \min\left\{\frac{\epsilon}{\sqrt{3}\lambda L_1(f)(d+3)^{\frac{3}{2}}}, \left(\frac{\lambda^2 L_1(f)h_2^2 - 2\rho}{24\lambda L_1(f)d + 24\lambda L_1^2(f)\rho(d+3)^3}\right)^{\frac{1}{2}}\epsilon\right\}.$$

*Then, it holds that*

$$\frac{1}{N+1}\sum_{k=0}^{N} E_{\mathcal{U}_k}[|F(\hat{z}_k)|^2] \leq \epsilon^2.$$

A proof of the Corollary 4 can be found in Appendix B. Based on Corollaries 3 and 4, and under appropriate parameter selection, we observe the following: employing Algorithm 1 guarantees convergence to a

neighbourhood—whose size depends on the variance of the stochastic oracle—of an $\epsilon$-stationary point of the objective function, within $\mathcal{O}(\epsilon^{-2})$ iterations and $\mathcal{O}(\epsilon^{-2})$ function evaluations. In contrast, utilising Algorithm 2, convergence to an actual $\epsilon$-stationary point (rather than a variance-dependent neighbourhood) is guaranteed in $\mathcal{O}(\epsilon^{-2})$ iterations and $\mathcal{O}(\epsilon^{-4})$ function evaluations.

**Remark 2.** *In Algorithm 2, if we set $B = \lambda\mathbf{I}$ (with $\lambda > 0$) and $t_k = k + 1$, then from (21), we get*

$$\frac{1}{N+1}\sum_{k=0}^{N}E_{\mathcal{U}_k}[|F_\mu(\hat{z}_k)|^2] \leq \frac{2\lambda^2 L_1(f)|z_0 - z^*|^2}{(\lambda^2 L_1(f)h_2^2 - 2\rho)(N+1)} + \frac{2\lambda L_1(f)d + 2\lambda L_1^2(f)\rho(d+3)^3}{(\lambda^2 L_1(f)h_2^2 - 2\rho)}\mu^2$$
$$+ \frac{(\ln(N+1)+1)3\lambda\sigma^2}{(N+1)(\lambda^2 L_1^2(f)h_2^2 - 2\rho L_1(f))}.$$

*Thus, to obtain $\frac{1}{N+1}\sum_{k=0}^{N}E_{\mathcal{U}_k}[|F_\mu(\hat{z}_k)|^2] \leq \epsilon^2$,*

$$N \geq \max\left\{\left\lceil\left(\frac{6\lambda^2 L_1(f)r_0^2}{(\lambda^2 L_1(f)h_2^2 - 2\rho)} + \beta\right)\epsilon^{-2} - 1\right\rceil, \left\lceil\beta\epsilon^{-2}\ln(\beta\epsilon^{-2}) - 1\right\rceil\right\}$$

*is required and where $\beta = \frac{9\lambda\sigma^2}{(\lambda 62 L_1^2(f)h_2^2 - 2\rho L_1(f))}$ and $\epsilon \leq 1$. Hence, compared to Corollary 4, the dependency of the number of iterations on $\epsilon$ changes and a total of $2(N+1)(N+2)$ function evaluations are required.*

In cases where specific properties of the objective function (such as Lipschitz constant $L_1(f)$ of the gradient or $\rho$ corresponding to the weak MVI) are unknown or can only be approximated, $\mu$ can be chosen independently of the objective function's properties. The following remark provides a guideline for selecting $\mu$ and $N$ to achieve a performance comparable to that of Corollary 2, in the case that $\mu$ is independent of the function's properties.

**Remark 3.** *Theorem 1's analysis can be repeated for the case where the smoothing parameter $\mu$ is iteration-dependent and satisfies $\mu_k = \frac{l}{k+1}$, for some positive scalar $l$. For this case, under the additional assumption that $B = \mathbf{I}$, (20) becomes*

$$\frac{1}{N+1}\sum_{k=0}^{N}E_{\mathcal{U}_k}[\|F_{\mu_k}(\hat{z}_k)\|^2] \leq \frac{L_1(f)\|z_0 - z^*\|^2}{(L_1(f)h_2^2 - 2\rho)(N+1)} + \frac{L_1(f)d + L_1^2(f)\rho(d+3)^3}{(L_1(f)h_2^2 - 2\rho)}\frac{l^2\pi^2}{6(N+1)} \tag{22}$$
$$+ \frac{3\sigma^2}{(L_1^2(f)h_2^2 - 2\rho L_1(f))}.$$

*Then, for a given tolerance $\epsilon > 0$, if*

$$N \geq \left\lceil\left(\frac{2L_1(f)r_0^2}{(L_1(f)h_2^2 - 2\rho)} + \frac{L_1(f)d + L_1^2(f)\rho(d+3)^3}{(L_1(f)h_2^2 - 2\rho)}\frac{l^2\pi^2}{6}\right)\epsilon^{-2} - 1\right\rceil,$$

*we have*

$$\frac{1}{N+1}\sum_{k=0}^{N}E_{\mathcal{U}_k}[\|F_{\mu_k}(\hat{z}_k)\|^2] \leq \epsilon^2 + \frac{3\sigma^2}{(L_1^2(f)h_2^2 - 2\rho L_1(f))}.$$

*As can be seen, if $l$ is selected independently of $d$, then the number of iterations to achieve a tolerance of $\epsilon$ is of order $\mathcal{O}(d^3\epsilon^{-2})$. However, it is possible to reduce the power of $d$ in the complexity order by choosing $l$ appropriately. For example, if $l = \frac{1}{d}$, the number of iterations to achieve a tolerance of $\epsilon$ is of order $\mathcal{O}(d\epsilon^{-2})$. For the sake of comparison, in Anagnostidis et al. (2021), the authors extended the direct search algorithm of Vicente (2013) and analysed the unconstrained differentiable NC-PL min-max problem and showed that the complexity order of the direct search algorithm for computing an $\epsilon$-stationary point is $\mathcal{O}(d^2\epsilon^{-2}\log(\epsilon^{-1}))$.*

## 3.2 Constrained Differentiable Problem

Here, we study the performance of Algorithm 1 for solving the constrained version of Problem 1 where $\mathcal{Z} \subset \mathbf{Z}$ is a convex compact set with $D_z$ as its diameter. To ensure that the iterates stay in the constraint

set, projection steps are needed. In this case, Problem 1, with $f$ as its objective function and $\mathcal{Z}$ as its constraint set, can be reformulated as an unconstrained problem with $\Gamma(z)$ as its objective function, where

$$\Gamma(z) \stackrel{\text{def}}{=} f(z) + \mathcal{I}_{\mathcal{Z}}(z) \quad \text{and} \quad \mathcal{I}_{\mathcal{Z}}(z) \stackrel{\text{def}}{=} \mathcal{I}_{\mathcal{X}}(x) - \mathcal{I}_{\mathcal{Y}}(y) \quad \text{with} \quad I_{\mathcal{Z}}(z) \stackrel{\text{def}}{=} \begin{cases} 0 & z \in \mathcal{Z}, \\ \infty & z \notin \mathcal{Z}. \end{cases} \tag{23}$$

It is easy to see that $\Gamma : \mathbf{Z} \to \mathbb{R} \cup \{\infty\}$ is not differentiable and its gradient is not defined everywhere. Thus, we can not use Definitions 9 and 10 with the gradient of $\Gamma$. To proceed and to analyse stationary points of $f$ in the sense of Definition 3, we define operator $Q_\ell$ as follows:

$$Q_\ell(z, a, F(\bar{z})) \stackrel{\text{def}}{=} -\frac{1}{a}(\text{Prox}_\ell(z - aF(\bar{z})) - z), \qquad \forall z, \bar{z} \in \mathcal{Z}. \tag{24}$$

Here, $a$ is a positive scalar and $\text{Prox}_\ell(\bar{z}) \stackrel{\text{def}}{=} \arg\min_z(\|z - \bar{z}\|^2 + \ell(z))$ for a proper and lower semicontinuous function $\ell$.

For the instances where $\ell = \mathcal{I}_{\mathcal{Z}}$ and $\text{Prox}_\ell = \text{Proj}_{\mathcal{Z}}$, we recover $\tau$ defined in Definition 3. Next, we define the proximal (weak) Minty variational inequality, analogous to Definitions 9 and 10, for the analysis of Algorithm 1 in the constrained case.

**Definition 11** (Proximal (weak) Minty variational inequality). *Consider a closed and convex set $\mathcal{Z} \subset \mathbf{Z}$, a Lipschitz operator $F : \mathbf{Z} \to \mathbf{Z}$ with Lipschitz constant $L > 0$, and a possibly non-differentiable convex function $\ell$. Then $z^* \in \mathcal{Z}$ is said to satisfy the proximal Minty variational inequality if*

$$\langle Q_\ell(z, a, F(\bar{z})), \bar{z} - z^* \rangle \geq 0, \tag{25}$$

*holds for all $z, \bar{z} \in \mathcal{Z}$.*

*Moreover, $z^* \in \mathcal{Z}$ is said to satisfy the proximal weak Minty variational inequality if*

$$\langle Q_\ell(z, a, F(\bar{z})), z - z^* \rangle + \frac{\rho}{2}\|Q_\ell(z, a, F(\bar{z}))\|_*^2 \geq 0, \quad \rho \in \left[0, \frac{1}{24L\kappa^2}\right), \tag{26}$$

*holds for all $z, \bar{z} \in \mathcal{Z}$, where operator $Q_\ell$ is defined in (24), and $\kappa$ denotes the condition number of $B$.*

By comparing Definition 11 with Definitions 9 and 10, it follows that, if function $\ell$ is constant or $\mathcal{Z} = \mathbf{Z}$ (as noted in Remark 7), the proximal (weak) MVI simplifies to the (weak) MVI.

We now discuss examples of functions satisfying the proximal MVI defined in (25). Consider $f(x, y) = xy$ with $\mathcal{Z} = \{z = (x, y) \mid x \geq 0, y \geq 0\}$ and $\ell = \mathcal{I}_{\mathcal{Z}}$, then $f$ satisfies the proximal MVI definition with $z^* = (0, 0)$. Similarly, the functions $f(x, y) = x^n y^m$ $(n, m > 0)$ with $\mathcal{Z} = \{z = (x, y) \mid x \geq 0, y \geq 0\}$ and $\ell = \mathcal{I}_{\mathcal{Z}}$ satisfy the definition of proximal MVI with $z^* = (0, 0)$. More generally, it can be shown that $f(x, y) = x^\top A y + c^\top x + d^\top y$, where $x, c \in \mathbb{R}^n$, $y, d \in \mathbb{R}^m$, $A \in \mathbb{R}^{n \times m}$, with nonnegative $A, c, d$, $\mathcal{Z} = \{z = (x, y) \mid x \geq 0, y \geq 0\}$ and $\ell = \mathcal{I}_{\mathcal{Z}}$ meets the definition of the proximal MVI (25) with $z^* = (0, 0)$.

Before proceeding further, we define the following auxiliary function

$$P_{\mathcal{Z}}(z, h, g(\bar{z})) \stackrel{\text{def}}{=} \frac{1}{h}\left[z - \text{Proj}_{\mathcal{Z}}(z - hg(\bar{z}))\right], \tag{27}$$

where $h$ is a positive scalar. We note that when $\ell$ is the indicator function, then $P_{\mathcal{Z}}(z, h, g(\bar{z})) = Q_\ell(z, h, g(\bar{z}))$. Moreover, let $F$, $G_\mu$, and $F_\mu$ be defined in (8), (13), and (15). Also, let $z_k$, $\hat{z}_k$, $h_{1,k}$, and $h_{2,k}$ be adopted from Algorithm 1. Then we can define below auxiliary variables:

$$s_k \stackrel{\text{def}}{=} P_{\mathcal{Z}}(z_k, h_{1,k}, G_\mu(z_k)), \qquad \hat{s}_k \stackrel{\text{def}}{=} P_{\mathcal{Z}}(z_k, h_{2,k}, G_\mu(\hat{z}_k)). \tag{28}$$

Hence, using above auxiliary variables in the constrained case of Problem 1, then the update steps in lines 5 and 8 in Algorithm 1 can be written as

$$z_{k+1} = z_k - h_{2,k}\hat{s}_k \quad \text{and} \quad \hat{z}_k = z_k - h_{1,k}s_k. \tag{29}$$

To proceed, we need to make an assumption about the existence of a solution for the proximal weak MVI in Definition 11.

**Assumption 3.** *For Problem 1 with $\mathcal{Z} \subset \mathbf{Z}$, compact and convex, and $\ell = \mathcal{I}_{\mathcal{Z}}$ defining the indicator function, there exists $z^* \in \mathcal{Z}$ such that $F(z)$ defined in (8) satisfies the proximal weak MVI defined in (26).*

Next, the main lemma of this subsection is presented. This result is analogous to Lemma 1 in the unconstrained setting but adapted for the constrained case. The lemma characterises $s_k$ and $\hat{s}_k$ under the assumption that there exists a $z^*$ satisfying Assumption 3.

**Lemma 2.** *Let $f(z)$ defined in Problem 1, be continuously differentiable with Lipschitz continuous gradient with constant $L_1(f) > 0$. Moreover, let $\hat{s}_k$ and $s_k$ be defined in (28), $\xi_k \stackrel{\text{def}}{=} G_\mu(z_k) - F_\mu(z_k)$, $\hat{\xi}_k \stackrel{\text{def}}{=} G_\mu(\hat{z}_k) - F_\mu(\hat{z}_k)$, $G_\mu$ and $F_\mu$ be defined in (13) and (15) with smoothing parameter $\mu > 0$, $\rho$ denote the proximal weak MVI parameter defined in Definition 11, $\kappa$ denote the condition number of $B$ in (2) and $D_z$ be the diameter of $\mathcal{Z} \subset \mathbb{R}^d$. If there exists $z^* \in \mathcal{Z}$ such that Assumption 3 is satisfied, then it holds that*

$$\langle s_k, z_k - z^* \rangle + \rho \|s_k\|_*^2 + \frac{\mu}{2} D_z \kappa L_1(f)(d+3)^{\frac{3}{2}} + D_z \kappa \|\xi_k\|_* + \frac{\mu^2}{2} \rho \kappa^2 L_1^2(f)(d+3)^3 + 2\rho\kappa^2 \|\xi_k\|_*^2 \geq 0, \quad (30)$$

$$\langle \hat{s}_k, \hat{z}_k - z^* \rangle + \rho \|\hat{s}_k\|_*^2 + \frac{\mu}{2} D_z \kappa L_1(f)(d+3)^{\frac{3}{2}} + D_z \kappa \|\hat{\xi}_k\|_* + \frac{\mu^2}{2} \rho \kappa^2 L_1^2(f)(d+3)^3 + 2\rho\kappa^2 \|\hat{\xi}_k\|_*^2 \geq 0. \quad (31)$$

A proof of Lemma 2 is provided in Appendix A. In the sequel, using Lemma 2, we can present the main theorem of this subsection. This theorem introduces an upper bound on the average of the expected value of the Euclidean norm of $s_k$ defined in (28). Considering the formulation in (29), this theorem is analogous to Theorem 1 in the unconstrained setting but adapted for the constrained case.

**Theorem 3.** *Let $f(z)$, defined in Problem 1, be continuously differentiable with Lipschitz continuous gradient with constant $L_1(f) > 0$. Let $\sigma^2$ be an upper bound on variance of the random oracle defined in Assumption 1, $N \geq 0$ be the number of iterations, $s_k$ be defined in (28) with smoothing parameter $\mu > 0$, $\mathcal{U}_k = [(u_0, \hat{u}_0), (u_1, \hat{u}_1), \cdots, (u_k, \hat{u}_k)]$, $k \in \{0, \ldots, N\}$, $\rho$ denotes the proximal weak MVI parameter defined in Definition 11, and $D_z$ be diameter of the compact and convex set $\mathcal{Z} \subset \mathbb{R}^d$. Moreover, let $\{z_k\}_{k \geq 0}$ and $\{\hat{z}_k\}_{k \geq 0}$ be the sequences generated by Algorithm 1, lines 5 and 8, respectively, suppose that Assumption 2 is satisfied, and recall the definitions of the smallest eigenvalue $\underline{\lambda}$, the largest eigenvalue $\overline{\lambda}$ and the condition number $\kappa$ of the positive definite matrix $B$ defined in (2). Then, for any iteration $N$, with $h_{1,k} = h_{2,k} = h$ and $h \in \left( \sqrt{\frac{6\rho}{L_1(f)\kappa\overline{\lambda}\underline{\lambda}}}, \frac{1}{2L_1(f)\kappa\overline{\lambda}} \right]$, we have*

$$\frac{1}{N+1} \sum_{k=0}^{N} E_{\mathcal{U}_k}[|s_k|^2] \leq \frac{2L_1(f)\underline{\lambda}\overline{\lambda}\kappa|z_0 - z^*|^2}{(L_1(f)h^2\underline{\lambda}\overline{\lambda}\kappa - 6\rho)(N+1)} + \frac{\mu D_z \underline{\lambda}\kappa L_1(f)(d+3)^{3/2}}{L_1(f)h^2\underline{\lambda}\overline{\lambda}\kappa - 6\rho} + \frac{\mu^2 \rho \underline{\lambda}\kappa^2 L_1(f)^2(d+3)^3}{L_1(f)h^2\underline{\lambda}\overline{\lambda}\kappa - 6\rho}$$
$$+ \frac{(36\rho\kappa^2\underline{\lambda} + \frac{4\underline{\lambda}}{L_1(f)})\sigma^2}{L_1(f)h^2\underline{\lambda}\overline{\lambda}\kappa - 6\rho} + \frac{2D_z\underline{\lambda}\kappa\sigma}{L_1(f)h^2\underline{\lambda}\overline{\lambda}\kappa - 6\rho}. \quad (32)$$

A proof of Theorem 3 is provided in Appendix B. Given the upper bound of Theorem 3, the first term on the right-hand side of (32) becomes arbitrarily small as $N \to \infty$. The second and third terms become arbitrarily small for $\mu \to 0$. The last two terms are dependent on the variance of the random oracle, defined in Assumption 1, which becomes arbitrarily small by using the variance reduction scheme in Algorithm 2. As in the unconstrained setting, an optimal selection of $B$ is out of the scope of this paper, but a discussion on the selection of $B$ can be found in Appendix C.2. As in Section 3.1, and motivated through the discussion in Appendix C.2, we focus on diagonal matrices $B = \lambda\mathbf{I}$, $\lambda > 0$, in the remainder of this section.

**Corollary 5.** *Let the assumptions of Theorem 3 be satisfied with $B = \lambda\mathbf{I}$ for $\lambda > 0$. Then, we have*

$$\frac{1}{N+1} \sum_{k=0}^{N} E_{\mathcal{U}_k}[|s_k|^2] \leq \frac{2\lambda^2 L_1(f)|z_0 - z^*|^2}{(\lambda^2 L_1(f)h^2 - 6\rho)(N+1)} + \frac{\mu\lambda D_z L_1(f)(d+3)^{3/2}}{\lambda^2 L_1(f)h^2 - 6\rho} + \frac{\mu^2\lambda\rho L_1(f)^2(d+3)^3}{\lambda^2 L_1(f)h^2 - 6\rho}$$
$$+ \frac{(36\rho + \frac{4}{L_1(f)})\lambda\sigma^2}{\lambda^2 L_1(f)h^2 - 6\rho} + \frac{2D_z\lambda\sigma}{\lambda^2 L_1(f)h^2 - 6\rho}. \quad (33)$$

The proof of Corollary 5 is the same as the proof of Theorem 3 with $\underline{\lambda} = \overline{\lambda} = \lambda$ and $\kappa = 1$. The next corollary gives a guideline on how to choose the number of iterations and the smoothing parameter provided a specific measure of performance $\epsilon$ and for diagonal matrices $B \succ 0$.

**Corollary 6.** *Let $\hat{s}_k$ be defined in (28), with $G_\mu$ defined in (13), and adopt the assumptions of Theorem 3 with $B = \lambda\mathbf{I}$ for $\lambda > 0$. Let $r_0 = \|z_0 - z^*\|$, $a \stackrel{\text{def}}{=} \frac{\rho\lambda L_1^2(f)(d+3)^3}{\lambda^2 L_1(f)h^2 - 6\rho}$, and $b \stackrel{\text{def}}{=} \frac{\lambda L_1(f)D_z(d+3)^{\frac{3}{2}}}{\lambda^2 L_1(f)h^2 - 6\rho}$. For a given $\epsilon > 0$, if*

$$\mu \leq \frac{-b + \sqrt{b^2 + 2a\epsilon^2}}{2a} \qquad and \qquad N \geq \left\lceil \left( \frac{4\lambda^2 L_1(f)r_0^2}{\lambda^2 L_1(f)h^2 - 6\rho} \right)\epsilon^{-2} - 1 \right\rceil,$$

*then,*

$$\frac{1}{N+1}\sum_{k=0}^{N} E_{\mathcal{U}_k}[|s_k|^2] \leq \epsilon^2 + \frac{(36\rho + \frac{4}{L_1(f)})\lambda\sigma^2}{\lambda^2 L_1(f)h^2 - 6\rho} + \frac{2D_z\lambda\sigma}{\lambda^2 L_1(f)h^2 - 6\rho}.$$

A proof of the Corollary 6 can be found in Appendix B. To proceed to the next result, we need the auxiliary variable below:

$$p_k \stackrel{\text{def}}{=} P_{\mathcal{Z}}(z_k, h_{2,k}, F(z_k)). \tag{34}$$

Considering Definition 3 and (27), we observe that $p_k$ can be written as $p_k = \tau(x_k, y_k)$. To show that the sequence generated by Algorithm 1 converges to an $\epsilon$-stationary point of $f$, it is needed to bound $|p_k|$. Based on Theorem 3, the next corollary provides an upper bound for the average expected value of $\hat{p}_k$ defined in (34).

**Corollary 7.** *Let $p_k$ be defined in (34), adopt the assumptions of Theorem 3 with $B = \lambda\mathbf{I}$ for $\lambda > 0$ and let $a \stackrel{\text{def}}{=} \frac{4\rho\lambda L_1^2(f)(d+3)^3}{\lambda^2 L_1(f)h^2 - 6\rho}$, $b \stackrel{\text{def}}{=} \frac{4\lambda L_1(f)D_z(d+3)^{\frac{3}{2}}}{\lambda^2 L_1(f)h^2 - 6\rho}$,*

$$\mu \leq \min\left\{ \frac{-b + \sqrt{b^2 + a\epsilon^2}}{2a}, \frac{\epsilon}{\sqrt{2}\lambda L_1(f)(d+3)^{\frac{3}{2}}} \right\} \qquad and \qquad N \geq \left\lceil \left( \frac{16\lambda^2 L_1(f)r_0^2}{\lambda^2 L_1(f)h^2 - 6\rho} \right)\epsilon^{-2} - 1 \right\rceil.$$

*Then, the following bound holds:*

$$\frac{1}{N+1}\sum_{k=0}^{N} E_{\mathcal{U}_k}[|p_k|^2] \leq \epsilon^2 + \left( \frac{4(36\rho + \frac{4}{L_1(f)})}{\lambda^2 L_1(f)h^2 - 6\rho} + 4 \right)\lambda\sigma^2 + \frac{8D_z\lambda\sigma}{\lambda^2 L_1(f)h^2 - 6\rho}.$$

A proof of the Corollary 7 can be found in Appendix B. Taking into account Definition 3 and Corollary 7, the projected Gaussian smoothing ZO estimate generated by Algorithm 1 is guaranteed to converge to a neighbourhood of the $\epsilon$-stationary points of $f$ in terms of the expected value. We further note that this neighbourhood can be ensured to be arbitrarily small using the variance reduction technique in Algorithm 2. The next theorem introduces an upper bound for the average of the expected value of the Euclidean norm of $s_k$ defined in (28) with respect to the sequence generated by Algorithm 2.

**Theorem 4.** *Let $f(z)$, defined in Problem 1, be continuously differentiable with Lipschitz continuous gradient with constant $L_1(f) > 0$. Let $\sigma^2$ be an upper bound on the variance of the random oracle defined in Assumption 1, $N \geq 0$ be the number of iterations, $t_k$ be the number of samples in each iteration of Algorithm 2, $s_k$ be defined in (28) with smoothing parameter $\mu > 0$, $\mathcal{U}_k = [(u_0, \hat{u}_0), (u_1, \hat{u}_1), \cdots, (u_k, \hat{u}_k)]$, $k \in \{0, \ldots, N\}$, $\rho$ denotes the proximal weak MVI parameter defined in Definition 11, and $D_z$ be diameter of the compact and convex set $\mathcal{Z} \subset \mathbb{R}^d$. Moreover, let $\{z_k\}_{k\geq 0}$ and $\{\hat{z}_k\}_{k\geq 0}$ be the sequences generated by Algorithm 1, lines 5 and 8, respectively, suppose that Assumption 2 is satisfied and recall the definition of the smallest eigenvalue $\underline{\lambda}$, the largest eigenvalue $\overline{\lambda}$ and the condition number $\kappa$ of the positive definite matrix $B$ defined in (2). Then,*

*for any iteration $N$, with $h_{1,k} = h_{2,k} = h$ and $h \in \left( \sqrt{\frac{6\rho}{L_1(f)\kappa\overline{\lambda}\underline{\lambda}}}, \frac{1}{2L_1(f)\kappa\overline{\lambda}} \right]$, we have*

$$\frac{1}{N+1}\sum_{k=0}^{N} E_{\mathcal{U}_k}[|s_k|^2] \leq \frac{2L_1(f)\underline{\lambda}\overline{\lambda}\kappa|z_0 - z^*|^2}{(L_1(f)h^2\underline{\lambda}\overline{\lambda}\kappa - 6\rho)(N+1)} + \frac{\mu D_z \underline{\lambda}\kappa L_1(f)(d+3)^{3/2}}{L_1(f)h^2\underline{\lambda}\overline{\lambda}\kappa - 6\rho} + \frac{\mu^2 \rho \underline{\lambda}\kappa^2 L_1(f)^2(d+3)^3}{L_1(f)h^2\underline{\lambda}\overline{\lambda}\kappa - 6\rho}$$
$$+ \frac{(36\rho\kappa^2\underline{\lambda} + \frac{4\underline{\lambda}}{L_1(f)})\sigma^2}{L_1(f)h^2\underline{\lambda}\overline{\lambda}\kappa - 6\rho}\frac{1}{N+1}\sum_{k=0}^{N}\frac{1}{t_k} + \frac{2D_z\underline{\lambda}\kappa\sigma}{L_1(f)h^2\underline{\lambda}\overline{\lambda}\kappa - 6\rho}\frac{1}{N+1}\sum_{k=0}^{N}\frac{1}{\sqrt{t_k}}. \quad (35)$$

A proof of Theorem 4 can be found in Appendix B. A discussion on the step size selection $h$ and the selection of $B$ can be found in Appendix C.2. The next corollary gives a guideline on how to choose the number of iterations and the smoothing parameter $\mu$, for a given specific measure of performance $\epsilon$ to guarantee the convergence to an $\epsilon$-stationary point of the objective function.

**Corollary 8.** *Let $p_k$ be defined in (34), adopt the assumptions of Theorem 4 with $B = \lambda\mathbf{I}$ for $\lambda > 0$. Moreover, define*

$$a \stackrel{\text{def}}{=} \frac{4\rho\lambda L_1^2(f)(d+3)^3}{\lambda^2 L_1(f)h^2 - 6\rho}, \quad b \stackrel{\text{def}}{=} \frac{4\lambda L_1(f)D_z(d+3)^{\frac{3}{2}}}{\lambda^2 L_1(f)h^2 - 6\rho}, \quad c \stackrel{\text{def}}{=} \frac{(36\rho + \frac{4}{L_1} + 4)\lambda\sigma^2}{\lambda^2 L_1 h^2 - 6\rho}, \quad d \stackrel{\text{def}}{=} \frac{2D_z\lambda\sigma}{\lambda^2 L_1 h^2 - 6\rho},$$

*and let*

$$t_k = t \geq 32\lceil \max\{c\epsilon^{-2}, d^2\epsilon^{-4}\}\rceil,$$
$$\mu \leq \min\left\{\frac{-b + \sqrt{b^2 + a\epsilon^2}}{2a}, \frac{\epsilon}{\sqrt{2}\lambda L_1(f)(d+3)^{\frac{3}{2}}}\right\} \quad \text{and} \quad N \geq \left\lceil \left(\frac{32\lambda^2 L_1(f)r_0^2}{\lambda^2 L_1(f)h^2 - 6\rho}\right)\epsilon^{-2} - 1 \right\rceil.$$

*Then, the following bound holds:*

$$\frac{1}{N+1}\sum_{k=0}^{N} E_{\mathcal{U}_k}[|p_k|^2] \leq \epsilon^2.$$

A proof of the Corollary 8 can be found in Appendix B. Based on Corollaries 7 and 8, and under appropriate parameter choices, the following conclusions can be drawn: employing Algorithm 1 ensures convergence to a neighbourhood—whose radius depends on the variance of the stochastic oracle—of an $\epsilon$-stationary point of the objective function, within $\mathcal{O}(\epsilon^{-2})$ iterations and $\mathcal{O}(\epsilon^{-2})$ function evaluations. In contrast, using Algorithm 2 guarantees convergence to an $\epsilon$-stationary point (i.e., not merely a variance-dependent neighbourhood), with the same iteration complexity $\mathcal{O}(\epsilon^{-2})$, but at the cost of $\mathcal{O}(\epsilon^{-6})$ function evaluations.

**Remark 4.** *In Pethick et al. (2023), the authors addressed the NC-NC min-max problem using a first-order extragradient algorithm with adaptive and constant step sizes. In their approach, they assume the existence of a solution to the weak MVI with respect to the operator $v = F + A$, where $F$ is the gradient operator and $A$ is the sub-differential operator of the indicator function. It is worth noting that both Assumption 3 and their assumption simplify to the weak MVI with respect to the gradient operator in the unconstrained case. Beyond this, there is no direct relationship between the assumptions, as each encompasses different classes of problems.*

**Remark 5.** *The results presented in Section 3.2 are restricted to convex and compact constraint sets. However, it is well known that if the objective function is coercive, then there exists a compact set containing the optimal solution. Consequently, any constrained problem with an unbounded convex constraint set and a coercive objective can be equivalently reformulated as a problem with a convex compact constraint set. We refer the reader to (Calafiore & El Ghaoui, 2014, Lemma 8.5) for additional information on extensions to coercive objective functions. Extending the current results to settings with unbounded constraint sets of infinite volume and non-coercive objective functions remains an important direction for future research.*

### 3.3 Unconstrained Non-differentiable Problem

The smoothed function $f_\mu$ defined in (10) has several nice properties that can circumvent the difficulties associated with solving non-differentiable problems. Among these the following two play a critical role in one's ability to solve these problems. First, it is known that $f_\mu$ is differentiable regardless of the differentiability of $f$ (Nesterov & Spokoiny, 2017, Section 2). Second, if $f$ is Lipschitz continuous, then $f_\mu$ has Lipschitz continuous gradients with its Lipschitz constant explicitly expressed in the following lemma.

**Lemma 3** ((Nesterov & Spokoiny, 2017, Lemma 2)). *Let $f : \mathbf{Z} \to \mathbb{R}$ be Lipschitz continuous with constant $L_0(f) > 0$ and $f_\mu$ be defined in (10). Then $f_\mu$'s gradient is Lipschitz continuous with $L_1(f_\mu) = \frac{d^{1/2}}{\mu} L_0(f)$.*

Moreover, the existing literature has characterised the relation between the stationary points of a smoothed function $f_\mu$ and the Goldstein stationary points of the original function. Specifically, (Lin et al., 2022, Theorem 3.1) proves that $\nabla f_\delta(z) \in \partial_\delta f(z)$ for any $z \in \mathbb{R}^d$, where $f_\delta(z) = E_{u\sim\mathbb{P}}[f(z + \delta u)]$ is the uniform smoothing with $\mathbb{P}$ being a uniform distribution on a unit ball. Lei et al. (2024) derives similar results for Gaussian smoothing of a class of functions, called Subdifferentially Polynomially Bounded, which includes global Lipschitz continuous functions as a special case.

**Lemma 4** ((Lei et al., 2024, Theorem 3.6 and Remark 3.7)). *Let $f : \mathbf{Z} \to \mathbb{R}$ be a Lipschitz continuous function with constant $L_0(f) > 0$ and $B = \mathbf{I}$ defined in (12) be the identity. Let $f_\mu : \mathbf{Z} \to \mathbb{R}$ be defined according to (10) and let $\partial_\delta f$ be the $\delta$-Goldstein subdifferential defined through Definition 6. For $0 < \delta < 1$, $0 < \gamma \le \min\{5L_0(f), 1\}$, and $\mu \le \frac{\delta}{\sqrt{d\pi e}}(\frac{\gamma}{4L_0(f)})^{1/d}$, it holds that*

$$\nabla f_\mu(z) \in \partial_\delta f(z) + \mathbb{B}_\gamma(0) \quad \forall z \in \mathbf{Z}.$$

These results motivate us to study the convergence of ZO-EG in non-differentiable min-max optimisation via the smoothed function $f_\mu$. Towards that end, in the following we make an assumption on the existence of a solution to the weak MVI with respect to $\mathcal{Z} = \mathbf{Z}$ and $F_\mu$, defined in (15).

**Assumption 4.** *Consider Problem 1 with $\mathcal{Z} = \mathbf{Z}$. Let $f : \mathbf{Z} \to \mathbb{R}$ be a Lipschitz continuous function, $F_\mu(z)$ be defined in (15), $L_1(f_\mu)$ be the Lipschitz constant of the gradients of $f_\mu$ defined in (10) and let $\kappa$ denote the condition number of the matrix $B$ defined in (2). For all $z \in \mathcal{Z}$, there exist $z^* \in \mathcal{Z}$ such that*

$$\langle F_\mu(z), z - z^* \rangle + \frac{\rho}{2}\|F_\mu(z)\|_*^2 \ge 0, \quad \rho \in \left[0, \frac{1}{4L_1(f_\mu)\kappa^2}\right).$$

From Lemma 3, we see that as long as $f$ is Lipschitz continuous, $L_1(f_\mu)$ is well-defined and can be expressed in terms of $L_0(f)$. Hence, Assumption 4 is well-defined for studying non-differentiable min-max optimisation problems. One simple but non-differentiable example that satisfies Assumption 4 is $f(x, y) = |x| - |y|$, for $x, y \in \mathbb{R}$ and $\mathcal{Z} = \mathbb{R}^2$. We leave the proof to Appendix D.

Having this set-up, we can discuss the convergence of Algorithm 1 when the objective function is non-differntiable.

**Theorem 5.** *Let $f(z)$, defined in Problem 1, be Lipschitz continuous with constant $L_0(f) > 0$ and recall the definitions of the smallest eigenvalue $\underline{\lambda}$, the largest eigenvalue $\overline{\lambda}$ and the condition number $\kappa$ of the positive definite matrix $B$ defined in (2). Let $\sigma^2$ be an upper bound on variance of the random oracle defined in Assumption 1, $N \ge 0$ be the number of iterations, $F_\mu$ be defined in (15) with smoothing parameter $\mu > 0$, $\mathcal{U}_k = [(u_0, \hat{u}_0), (u_1, \hat{u}_1), \cdots, (u_k, \hat{u}_k)]$, $k \in \{0, \ldots, N\}$, $\rho$ denotes the weak MVI parameter defined in Assumption 4, and $L_1(f_\mu)$ be the Lipschitz constant of the gradient of $f_\mu$. Moreover, let $\{z_k\}_{k\ge0}$ and $\{\hat{z}_k\}_{k\ge0}$ be the sequences generated by Algorithm 1 (see lines 5 and 8) and suppose Assumption 4 is satisfied. Then, for any number of iterations $N$, with $h_{1,k} = h_1 \le \frac{1}{L_1(f_\mu)\overline{\lambda}\kappa}$ and $h_{2,k} = h_2 \in \left(\sqrt{\frac{\rho}{L_1(f_\mu)\overline{\lambda}\underline{\lambda}\kappa}}, \frac{h_1}{2}\right]$, we have*

$$\frac{1}{N+1}\sum_{k=0}^{N} E_{\mathcal{U}_k}[|F_\mu(\hat{z}_k)|^2] \le \frac{2L_1(f_\mu)\overline{\lambda}\underline{\lambda}\kappa|z_0 - z^*|^2}{(L_1(f_\mu)\overline{\lambda}\underline{\lambda}\kappa h_2^2 - \rho)(N+1)} + \frac{3\underline{\lambda}}{L_1(f_\mu)\kappa(L_1(f_\mu)\overline{\lambda}\underline{\lambda}\kappa h_2^2 - \rho)}\sigma^2. \tag{36}$$

A proof of Theorem 5 is provided in Appendix B. Given the upper bound of Theorem 5, the first term on the right-hand side of (36) becomes arbitrarily small for $N \to \infty$. The second term is dependent on the variance of the random oracle, defined in Assumption 1, which becomes arbitrarily small by using the variance reduction scheme in Algorithm 2. As before, we refer to Appendix C.3 for a discussion on the selection of the matrix $B$ and continue with results for the special case $B$ is defined as the identity matrix $B = \mathbf{I}$.

**Corollary 9.** *Consider the assumptions of Theorem 5 and let $B$ defined in (12) be the identity matrix $B = \mathbf{I}$. Then, we have*

$$\frac{1}{N+1} \sum_{k=0}^{N} E_{\mathcal{U}_k}[\|F_\mu(\hat{z}_k)\|^2] \leq \frac{2L_1(f_\mu)\|z_0 - z^*\|^2}{(L_1(f_\mu)h_2^2 - \rho)(N+1)} + \frac{3}{L_1(f_\mu)(L_1(f_\mu)h_2^2 - \rho)}\sigma^2. \tag{37}$$

The proof of Corollary 9 is the same as the proof of Theorem 5 with $\|z\| = \|z\|_B = \|z\|_*$ for all $z \in \mathbf{Z}$ and $\underline{\lambda} = \overline{\lambda} = 1$. The next corollary provides a guideline for choosing the hyperparameters of Theorem 5, given a specific measure of performance $\epsilon$.

**Corollary 10.** *Adopt the assumptions of Theorem 5 and let $B$ defined in (12) be the identity matrix $B = \mathbf{I}$. Let $\mu > 0$ be the smoothing parameter, $r_0 = \|z_0 - z^*\|$, and the step sizes to be $h_{1,k} = h_1 \leq \frac{1}{L_1(f_\mu)\overline{\lambda}\kappa}$ and $h_{2,k} = h_2 \in \left(\sqrt{\frac{\rho}{L_1(f_\mu)\overline{\lambda}\underline{\lambda}\kappa}}, \frac{h_1}{2}\right]$ with $L_1(f_\mu) = \frac{d^{1/2}L_0(f)}{\mu}$. For a given $\epsilon > 0$, if*

$$N \geq \left\lceil \left(\frac{2r_0^2 L_1(f_\mu)}{(L_1(f_\mu)h_2^2 - \rho)}\right) \epsilon^{-2} - 1 \right\rceil \qquad then \qquad \frac{1}{N+1}\sum_{k=0}^{N} E_{\mathcal{U}_k}[\|F_\mu(\hat{z}_k)\|^2] \leq \epsilon^2 + \frac{3}{L_1(f_\mu)(L_1(f_\mu)h_2^2 - \rho)}\sigma^2.$$

A proof of Corollary 10 is given in Appendix B. Considering Theorem 5, and Corollary 10, it can be concluded that the sequence generated by Algorithm 1 is guaranteed to converge to a neighbourhood of the $\epsilon$-stationary points of $f_\mu$ in the expected sense. The size of the neighbourhood can be made arbitrarily small using the variance reduction technique. Moreover, leveraging Lemma 4, $\nabla f_\mu$ belongs to a neighbourhood of $\delta$-Goldstein subdifferential, whose size becomes arbitrarily small by choosing appropriate parameters. Thus, this convergence result means that the point is a $(\delta, \bar{\epsilon})$-Goldstein stationary point of $f$, defined in Definition 7. The result is presented in the following corollary.

**Corollary 11.** *Adopt the assumptions of Lemma 4 and Theorem 5 and let $B$, as defined in (12), be the identity matrix $B = \mathbf{I}$. Let $0 < \delta < 1$, $r_0 = \|z_0 - z^*\|$, and the step sizes to be $h_{1,k} = h_1 \leq \frac{1}{L_1(f_\mu)\overline{\lambda}\kappa}$ and $h_{2,k} = h_2 \in \left(\sqrt{\frac{\rho}{L_1(f_\mu)\overline{\lambda}\underline{\lambda}\kappa}}, \frac{h_1}{2}\right]$. For a given $\epsilon > 0$, if*

$$\mu \leq \frac{\delta}{\sqrt{d\pi e}}\left(\frac{\epsilon}{8L_0(f)}\right)^{1/d} \quad and \quad N \geq \left\lceil \left(\frac{8r_0^2 L_1(f_\mu)}{(L_1(f_\mu)h_2^2 - \rho)}\right)\epsilon^{-2} - 1 \right\rceil,$$

*then there exists $z_k \in \mathbb{R}^d$, $k \in \{0, \ldots, N\}$, in the sequence generated by Algorithm 1 which is a $(\delta, \bar{\epsilon})$-Goldstein stationary point of $f$, where $\bar{\epsilon} = \epsilon + \sqrt{\frac{3}{L_1(f_\mu)(L_1(f_\mu)h_2^2 - \rho))}}\sigma$ in the expected sense.*

A proof of the Corollary 11 is given in Appendix B. Note that in Corollary 11, $\sigma$ is the upper bound on the variance of the random oracle, defined in Assumption 1, which becomes arbitrarily small by using the variance reduction scheme in Algorithm 2. The next theorem introduces an upper bound for the average of the expected value of the square norm of the gradient operator of the smoothed function in the sequence generated by Algorithm 2.

**Theorem 6.** *Let $f(z)$, defined in Problem 1, be Lipschitz continuous with constant $L_0(f) > 0$ and recall the definitions of the smallest eigenvalue $\underline{\lambda}$, the largest eigenvalue $\overline{\lambda}$ and the condition number $\kappa$ of the positive definite matrix $B$ defined in (2). Let $\sigma^2$ be an upper bound on the variance of the random oracle defined in Assumption 1, $N \geq 0$ be the number of iterations, $t_k$ be the number of samples in each iteration of Algorithm 2, $F_\mu$ be defined in (15) with smoothing parameter $\mu > 0$, $\mathcal{U}_k = [(u_0, \hat{u}_0), (u_1, \hat{u}_1), \cdots, (u_k, \hat{u}_k)]$,*

$k \in \{0, \ldots, N\}$, $\rho$ denotes the weak MVI parameter defined in Assumption 4, and $L_1(f_\mu)$ be the Lipschitz constant of the gradient of $f_\mu$. Moreover, let $\{z_k\}_{k \geq 0}$ and $\{\hat{z}_k\}_{k \geq 0}$ be the sequences generated by Algorithm 1 (see lines 5 and 8) and suppose Assumption 4 is satisfied. Then, for any number of iterations $N$, with $h_{1,k} = h_1 \leq \frac{1}{L_1(f_\mu)\overline{\lambda}\kappa}$ and $h_{2,k} = h_2 \in \left( \sqrt{\frac{\rho}{L_1(f_\mu)\overline{\lambda}\underline{\lambda}\kappa}}, \frac{h_1}{2} \right]$, we have

$$\frac{1}{N+1} \sum_{k=0}^{N} E_{\mathcal{U}_k}[|F_\mu(\hat{z}_k)|^2] \leq \frac{2L_1(f_\mu)\overline{\lambda}\underline{\lambda}\kappa|z_0 - z^*|^2}{(L_1(f_\mu)\overline{\lambda}\underline{\lambda}\kappa h_2^2 - \rho)(N+1)} + \frac{3\underline{\lambda}\sigma^2}{L_1(f_\mu)\kappa(L_1(f_\mu)\overline{\lambda}\underline{\lambda}\kappa h_2^2 - \rho)} \frac{1}{N+1} \sum_{k=0}^{N} \frac{1}{t_k}. \quad (38)$$

A proof of Theorem 6 can be found in Appendix B and for the step size selection and a selection of the matrix $B$ we refer again to Appendix C.3. The next corollary gives a guideline on how to choose the number of iterations, the smoothing parameter $\mu$, and the number of samples in each iteration of Algorithm 2, for a given specific measure of performance $\epsilon$ to guarantee the convergence to a $(\delta, \epsilon)$-Goldstein stationary point of the objective function.

**Corollary 12.** *Adopt the assumptions of Lemma 4 and Theorem 6 and let $B$ defined in* (12) *be the identity matrix $B = \mathbf{I}$. Let $0 < \delta < 1$, $r_0 = \|z_0 - z^*\|$, and the step sizes to be $h_{1,k} = h_1 \leq \frac{1}{L_1(f_\mu)\overline{\lambda}\kappa}$ and $h_{2,k} = h_2 \in \left( \sqrt{\frac{\rho}{L_1(f_\mu)\overline{\lambda}\underline{\lambda}\kappa}}, \frac{h_1}{2} \right]$. For a given $\epsilon > 0$, if*

$$\mu \leq \frac{\delta}{\sqrt{d\pi e}} \left( \frac{\epsilon}{8L_0(f)} \right)^{1/d}, \quad N \geq \left\lceil \left( \frac{16r_0^2 L_1(f_\mu)}{(L_1(f_\mu)h_2^2 - \rho)} \right) \epsilon^{-2} - 1 \right\rceil, \quad and \quad t_k = t \geq \left\lceil \frac{24\sigma^2}{L_1(f_\mu)(L_1(f_\mu)h_2^2 - \rho)} \epsilon^{-2} \right\rceil,$$

*then there exists $z_k \in \mathbb{R}^d$, $k \in \{0, \ldots, N\}$, in the sequence generated by Algorithm 1 which is a $(\delta, \epsilon)$-Goldstein stationary point of $f$ in the expected sense.*

A proof of the Corollary 12 can be found in Appendix B. In light of Corollaries 11 and 12, and under appropriate parameter selection, it can be established that employing Algorithm 1 with $\mathcal{O}(\epsilon^{-2})$ iterations and $\mathcal{O}(\epsilon^{-2})$ function evaluations ensures convergence to a neighbourhood—dependent on the variance of the random oracle—of a $(\delta, \epsilon)$-Goldstein stationary point of the objective function. In contrast, utilising Algorithm 2 with $\mathcal{O}(\epsilon^{-2})$ iterations and $\mathcal{O}(\epsilon^{-4})$ function evaluations guarantees convergence to a true $(\delta, \epsilon)$-Goldstein stationary point, independent of the oracle variance.

**Remark 6.** *In this paper, we have discussed unconstrained non-differentiable min-max optimisation by assuming the existence of solutions of the weak MVI based on the smoothed function $f_\mu$. Similar assumptions can be made via the proximal weak MVI when constrained non-differentiable min-max optimisation is studied. For the sake of brevity, we omit the detailed discussion here.*

In the next section, we provide a general discussion on Algorithm 1 and the main results presented in Sections 3.1, 3.2, and 3.3.

### 3.4 Other oracles, the effect of noisy function evaluation, and the selection of $B$

In this section, we give some general explanations on Algorithm 1, the main theorems, and the results of this study. First, we focus on the choice of the random oracle defined in (11). The random Gaussian oracle defined in (11) is a forward approximation oracle. Other types of random Gaussian oracles, such as the central difference approximation, i.e.,

$$\bar{g}_\mu(z) = \frac{f(z + \mu u) - f(z - \mu u)}{2\mu} Bu,$$

and the backward approximation, i.e.,

$$\tilde{g}_\mu(z) = \frac{f(z) - f(z - \mu u)}{\mu} Bu,$$

are used in the literature of ZO optimisation (see Nesterov & Spokoiny (2017); Malladi et al. (2023), for example). We should note that the proofs of all the results in this work only rely on

(i) the unbiasedness of the random oracle as an estimation of the gradient of the Gaussian smoothed version of the objective function; and

(ii) the boundedness of the variance of the random oracle.

Any random oracle satisfying the aforementioned properties can be employed as a substitute for the forward approximation oracle defined in (11), without affecting the convergence bounds established in Sections 3.1, 3.2, and 3.3. Although the forward approximation oracle is prevalent in the literature Nesterov & Spokoiny (2017); Liu et al. (2020); Wang et al. (2023); Farzin & Shames (2024); Xu et al. (2020); Huang et al. (2022); Xu et al. (2023), the central difference approximation often provides higher accuracy by eliminating the first-order term in the Taylor expansion. Empirically, the central difference approximation oracle may outperform both forward and backward approximations, particularly in scenarios with noisy feedback, as demonstrated in Section 4.3.

In this paper, we have considered noiseless function evaluations. In the presence of noisy evaluations—modelled as function values corrupted by additive noise $\gamma$ with zero mean and bounded variance, i.e., $E[\gamma] = 0$ and $E[\gamma^2] < \infty$—the variance bound associated with the forward approximation oracle defined in (11) scales by a factor of $\frac{1}{\mu^2}$. In contrast, when employing the central difference approximation, the variance bound improves, scaling it by $\frac{1}{4\mu^2}$. Although it should be noted that in each iteration of Algorithm 2, the central difference approximation requires $4t_k$ function evaluations, whereas the forward or backward difference approximations require $2t_k + 2$ function evaluations, where $t_k$ denotes the number of sampled directions. The practical benefits of the central difference approximation over the forward approximation are demonstrated numerically in Section 4.3. A rigorous theoretical analysis addressing the noisy feedback scenario remains an important direction for future research.

Regarding the choice of the matrix $B$ defined in (12), it is common in the literature to select $B$ as the identity matrix Balasubramanian & Ghadimi (2022); Malladi et al. (2023); Wang et al. (2023); Xu et al. (2020). However, our main theorems in Sections 3.1, 3.2, and 3.3 are stated for a general positive definite matrix $B = B^* \succ 0$. A detailed discussions on the selection of the minimum and maximum eigenvalues of $B$ are provided in Appendix C. The impact of different choices of $B$ is further investigated numerically in Section 4.3. Extending the theoretical analysis of the choice of $B$ constitutes an important direction for future work.

Comparing the update rule in (Nesterov & Spokoiny, 2017, (66)) with the update rules in Algorithms 1 and 2 of this work, one key difference is that the random oracle in (Nesterov & Spokoiny, 2017, (66)) is multiplied by $B^{-1}$, whereas in our work it is not. Furthermore, the results in Nesterov & Spokoiny (2017) are expressed in terms of the primal and dual norms, while our results are formulated using the $|\cdot| = \|\cdot\|_{B^{-1}}$ norm. This difference in formulation explains why the eigenvalues and condition number of $B$ do not appear in the final results of Nesterov & Spokoiny (2017). Specifically, the use of primal-dual norms and multiplication by $B^{-1}$ absorb the influence of $B$ in their bounds. However, translating their results from primal-dual norms into the $|\cdot|$ norm would generally lead to more conservative bounds compared to deriving the results directly in the $|\cdot|$ norm, as we do in this work. Our motivation for using the $|\cdot|$ norm is to make the decay of the iterates explicit without masking the influence of $B$. To illustrate this point, suppose $|z|^2 \leq C$ for some constant $C > 0$. Then, using the dual norm $\|\cdot\|_*$, we have $\|z\|_*^2 \leq \frac{C}{\lambda}$. This implies that by choosing $\underline{\lambda}$ sufficiently large, one can make $\|z\|_*^2$ arbitrarily small. This highlights how the $\|\cdot\|_*$ norm can obscure the true behaviour of the algorithm unless $B$ is explicitly accounted for. For more details, see Appendix E. It is worth noting that the per-iteration complexity of Algorithm 2 is lower than that of Algorithm 3, as it avoids the additional matrix-vector multiplication step. Furthermore, in Algorithms 1 and 2, the matrix $B$ is treated as a hyperparameter, and its selection influences the algorithm's performance with respect to the Euclidean norm of the gradient operator. In contrast, the hyperparameter choice in Algorithm 3 is independent of the choice of $B$, and for a fixed hyperparameter configuration, the algorithm guarantees consistent performance across all choices of $B$, measured with respect to the $B$-weighted primal and dual norms.

Next, we discuss the relationship between second-order optimality conditions and the results of this study. Prior works such as Daskalakis & Panageas (2018); Jin et al. (2020); Farzin et al. (2025); Cai et al. (2024) characterise local convergence behaviour and limit points of certain first-order algorithms, analysing second-

order optimality conditions to relate these limit points to local Nash equilibria, local min-max points, or $\Phi$-equilibria. To the best of our knowledge, the second-order optimality properties of limit points produced by ZO algorithms have not yet been investigated. In this study, we focused on establishing first-order optimality properties of the sequence generated by the ZO-EG algorithm. Examining second-order optimality constitutes a promising avenue for future research.

## 4 Numerical examples

In this section, we evaluate the performance of Algorithm 1 via numerical experiments. First, ZO-EG is applied to three toy functions, and the trajectory of iterates for each case is analysed. Second, a robust underdetermined least squares problem is studied, and the convergence trajectory of Algorithm 1 is analysed and compared with other algorithms. The third example is concerned with a data poisoning attack on a logistic regression problem. Algorithm 1 is applied to this problem and it is shown how it compromises the prediction accuracy of a logistic regressor. The performance of the algorithm is compared to that of the direct search (DS) algorithm from (Anagnostidis et al., 2021). In the fourth example, a classifier neural network is trained and a corresponding empirical risk minimisation problem is solved using Algorithm 1. Finally, a version of a lane merging problem, which can be formulated as min-max problem, is implemented and solved using ZO-EG. In all the examples below, we set $B_1^{-1} = B_2^{-1} = \mathbf{I}$, unless stated otherwise.

### 4.1 Low Dimensional Toy Problems

In this section, we apply Algorithm 1 to the following three functions:

$$f_1(x, y) = 2x^2 - 2y^2 + 4xy + 10\sin(xy), \tag{39}$$

$$f_2(x, y) = \log(1 + e^x) + 3xy - \log(1 + e^y), \tag{40}$$

$$f_3(x, y) = |x^3 - 1| - |y^3 + 1|. \tag{41}$$

We consider the functions $f_1$, $f_2$ and $f_3$ as the objective functions of the min-max Problem 1. Function $f_1(x, y)$ is smooth and nonconvex-nonconcave, and thus fits into the setting discussed in Section 3.1. Function $f_2(x, y)$ is smooth and nonconvex-nonconcave and is considered in a constrained setting where $\mathcal{X} \stackrel{\text{def}}{=} \{x \in \mathbb{R} : |x| \leq 3\}$ and $\mathcal{Y} \stackrel{\text{def}}{=} \{y \in \mathbb{R} : |y| \leq 2\}$ and thus, the corresponding theory, is covered in Section 3.2. Function $f_3(x, y)$ is non-differentiable and nonconvex-nonconcave. Hence, we can use the theory of Section 3.3 to study the performance of Algorithm 1. For (39) and (41), we choose $h_1 = 2 \times 10^{-3}$, $h_2 = 10^{-3}$, and $\mu = 10^{-6}$. For (40), we choose $h_1 = h_2 = 10^{-3}$ and $\mu = 10^{-6}$. The sequence $\{x_k, y_k\}_{k \geq 0}$ for objective functions $f_1$ and $f_2$ with two initial values $(5, -7)$ and $(-7, 5)$ and for objective function $f_3$ with two initial values $(7, -1)$ and $(1, 7)$ are shown in Figure 1. For (41), Algorithm 1 is initialised through values where $f_1$ is non-differentiable. As expected from the theoretical results, from Figure 1, we observe that Algorithm 1 successfully converges to the stationary point of the objective function for all three cases.

### 4.2 Robust Least-Squares Problem

We illustrate the behaviour of Algorithm 1 when applied to a robust least-squares (RLS) problem. Slightly deviating from the notation so far, let $A \in \mathbb{R}^{n \times m}$ be the coefficient matrix and $y_0 \in \mathbb{R}^n$ be the noisy measurement vector for $n, m \in \mathbb{N}$. We assume that $y_0$ is subject to a bounded additive deterministic perturbation $\delta \in \mathbb{R}^n$. The RLS problem can be formulated as (El Ghaoui & Lebret, 1997)

$$\min_x \max_{|\delta| \leq \rho} \quad |Ax - y_0 + \delta|^2.$$

This problem has a compact convex constraint set $\mathbb{B}_\rho(0)$ with respect to the optimisation variable $\delta$. We set $\rho = 5$, $n = 150$, $m = 250$ and sample the elements of $A$ and $y_0$ from $\mathcal{N}(0, 1)$. In Algorithm 1 we choose $h_1 = h_2 = 10^{-5}$ and $\mu = 10^{-9}$. For comparison, we solve the same problem with Gradient Descent Ascent (GDA) and min-max Direct Search (DS) Anagnostidis et al. (2021) algorithms. The sequence of the objective function values is plotted against the execution time (0.5 sec), the number of iterations, and the

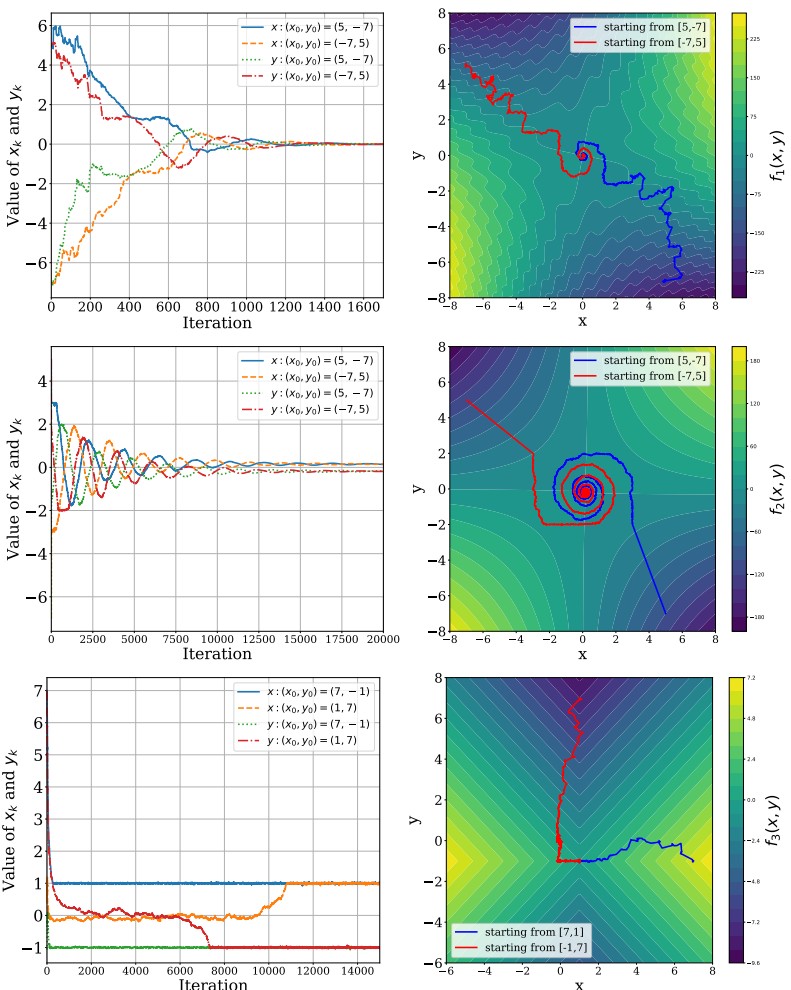

Figure 1: The trajectories of iterates generated by Algorithm 1 applied to functions $f_1, f_2, f_3$ in Section 4.1.

number of function calls for ZO-EG, GDA, and DS in Figure 2. In each iteration of ZO-EG, there are 2 oracle calls (i.e., 4 function evaluations), whereas the number of function evaluations per iteration in DS can vary. Notably, for this particular example, both ZO-EG and DS reached their target within 0.5 seconds, but DS made minimal progress. The average and the standard deviation over 10 runs of the wall-clock times, as

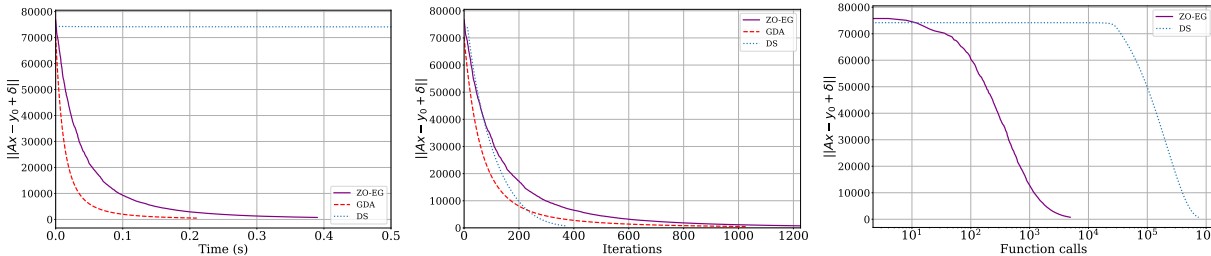

Figure 2: On the left, objective function value versus the execution time, in the middle, objective function value versus the number of iterations, on the right, objective function value versus the number of function calls for RLS problem of Section 4.2.

measured by the python `time` package, for each algorithm to yield an iterate that results in the function value

Table 1: Average wall-clock time to reach $0.5\%\|Ax_0 - y_0 + \delta_0\|$ for 10 runs for the RLS problem in Section 4.2

|  | ZO-EG | GDA | DS |
|---|---|---|---|
| Average Wall-Clock Time (s) | 0.39 | 0.21 | 24.56 |
| Standard Deviation | 0.13 | 0.04 | 3.80 |

of $0.5\%\|Ax_0 - y_0 + \delta_0\|$ are presented Table 1. Note that GDA, as a first-order method, uses the gradient of the objective function while ZO-EG only has access to the function values. Also, in each iteration of GDA, two gradient evaluations are needed, while we need four function evaluations in each iteration of ZO-EG.

### 4.3 Choice of $B$ and the Oracle, Effect of Output Noise, and Impact of Increased Sample Size

In this section, we illustrate the behaviour of Algorithms 1 and 2 applied to the RLS problem described in Section 4.2. First, we analyse the effect of the choice of matrix $B$ defined in (12). Then, assuming noisy function evaluations, we study the performance of Algorithms 1 and 2 using forward, backward, and central difference approximation oracles with varying numbers of samples per iteration. We set $\rho = 5$, $n = 150$, $m = 250$, and sample the elements of $A$ and $y_0$ from $\mathcal{N}(0, 1)$. In both algorithms, we fix $h_1 = h_2 = 10^{-5}$, $N = 5000$, and $\mu = 10^{-5}$. We run Algorithm 1 with five different choices of $B$ defined in (12). Specifically, $B_1$ is a diagonal matrix with all diagonal entries equal to 10, $B_2$ is a diagonal matrix with all diagonal entries equal to 0.1, $B_3$ is diagonal with entries randomly chosen between $[0.1, 10]$ (yielding a condition number $\kappa = 98.87$), $B_4$ is diagonal with half of the diagonal entries equal to 10 and the other half equal to 1, randomly placed, and $B_5$ is the identity matrix. Figure 3 plots the average objective function value over five runs. It can be seen that, for this example, the choices of $B$ with condition number equal to 1 perform better and yield smoother convergence.

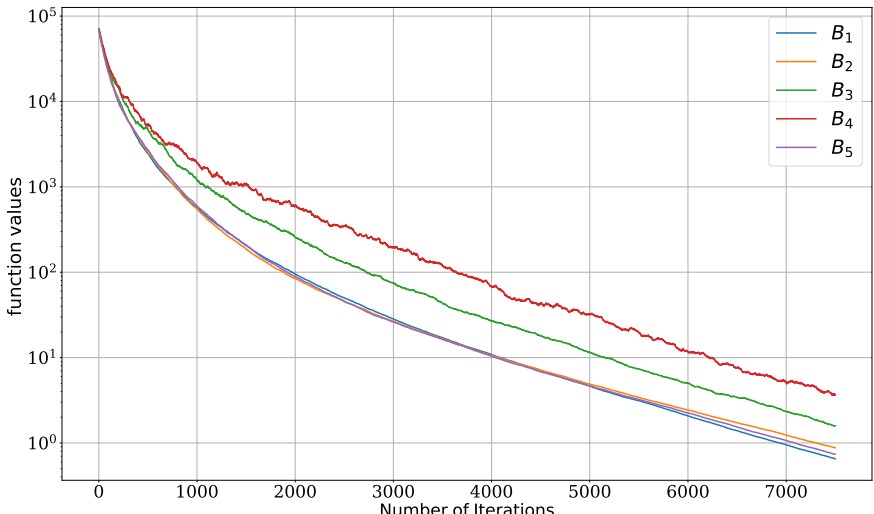

Figure 3: Objective function values over the generated sequence by Algorithm 1 with different choice of $B$ for the example of Section 4.3.

Next, we test Algorithms 1 and 2 in the case of noisy feedback. The parameter selection is as mentioned above, and we choose $B$ as the identity matrix. First, we consider output noise sampled from $\mathcal{N}(0, 10^{-3})$, $\mathcal{N}(0, 10^{-2})$, and $\mathcal{N}(0, 10^{-1})$, and test Algorithms 1 and 2 with forward, backward, and central difference approximation oracles. The number of samples per iteration of Algorithm 2 is set to 100. The average objective function values over 3 runs are plotted in Figure 4. It can be observed that when the noise level is small, there is little difference among the oracles; however, as the noise scale increases, the central difference oracle outperforms the forward and backward oracles. Additionally, in all three noise cases, the variance reduction technique employed in Algorithm 2 improves convergence by reducing variance, leading to better

and more accurate results. We repeated the test with noise sampled from $\mathcal{N}(0, 10^{-1})$, but with $\mu = 10^{-4}$. The average objective function values over 3 runs are plotted in Figure 5. Comparing the bottom row of Figure 4 with Figure 5 reveals the inverse dependency of variance on $\mu$; under the same noise conditions, a larger smoothing parameter leads to more accurate convergence.

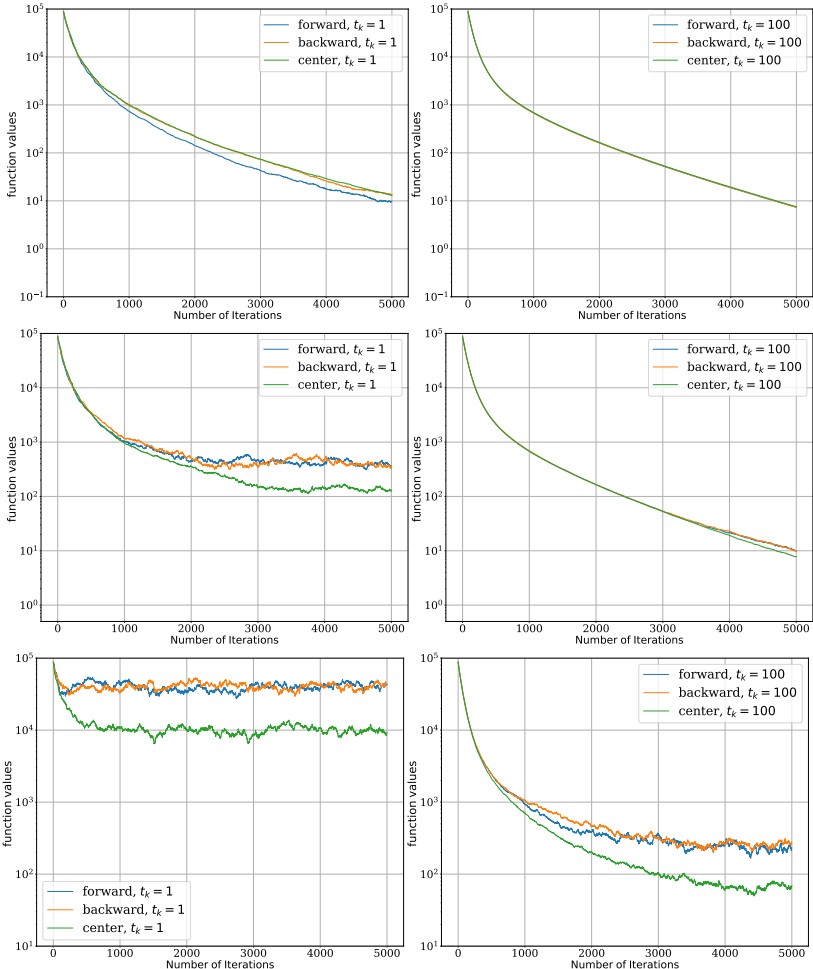

Figure 4: Performance of Algorithms 1 (left column) and 2 (right column) under additive Gaussian output noise in the example of Section 4.3. The plots show the objective function values over the generated sequence for different noise levels. From top to bottom, the noise is sampled from $\mathcal{N}(0, 10^{-3})$, $\mathcal{N}(0, 10^{-2})$, and $\mathcal{N}(0, 10^{-1})$, respectively.

## 4.4 Data Poisoning attack on Logistic Regression

Following the examples in Huang et al. (2022); Liu et al. (2020), as a next example, we consider a poisoning attack scenario where a fraction of the samples is corrupted by an additive perturbation vector aiming to compromise the training process and, consequently, deteriorate the prediction accuracy. This problem can be formulated as

$$\max_{\|x\|_\infty \leq \zeta} \min_y \quad h(x, y; D_p) + h(0, y; D_t) + \lambda\|y\|^2,$$

where $D_p$ is the corrupted data set and $D_t$ is the clean data set, $\zeta > 0$ is the maximum allowed perturbation magnitude, $\lambda > 0$ is a regularisation parameter, $y$ is the model parameter, and $x$ is the corruption vector. Note that this max-min problem can be reformulated as a min-max problem: $\min_{\|x\|_\infty \leq \zeta} \max_y f(x, y)$ with

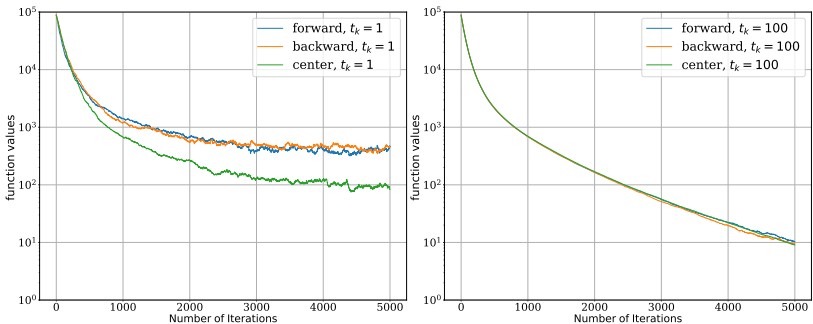

Figure 5: Performance of Algorithms 1 and 2 with $\mu = 10^{-4}$, under additive Gaussian output noise sampled from $\mathcal{N}(0, 10^{-1})$ in the example of Section 4.3.

$f(x, y) \overset{\text{def}}{=} -(h(x, y; D_p) + h(0, y; D_t) + \lambda\|y\|^2)$. The corruption ratio is set to 15%. We consider a binary cross-entropy loss function, i.e., $h(x, y; D) = -\frac{1}{\text{card}(D)} \sum_{(a_i, b_i) \in D} [b_i \log(g(x, y; a_i)) + (1 - b_i) \log(1 - g(x, y; a_i))]$ and $g(x, y; a_i) = \frac{1}{1 + \exp(-(x + a_i)^\top y)}$, where $\text{card}(D)$ denotes the cardinality of the set $D$.

In the experiment, we generate 500 samples. Specifically, we randomly draw the feature vectors $a_i \in \mathbb{R}^{20}$ ($n = m = 20$) from $\mathcal{N}(0, \mathbf{I})$. Label $b_i = 1$ if $\frac{1}{1 + \exp(-(a_i^\top \theta + v_i))} \geq \frac{1}{2}$, otherwise $b_i = 0$. Moreover, $\theta$ and $v_i$ are sampled from $\mathcal{N}(0, 1)$ and $\mathcal{N}(0, 10^{-3})$, respectively, for $i = 1, \ldots, 500$. We let $\lambda = 0.001$ and $\zeta = 10$. DS is implemented with the same parameter setting as the first experiment of (Anagnostidis et al., 2021) as a comparison. ZO-EG is run for 12000 iterations with $h_1 = h_2 = 10^{-3}$ and DS is run for 330 iterations. We note that, for this particular example, on average, each iteration of ZO-EG takes $6.9 \times 10^{-3}$ seconds and each iteration of DS takes 0.24 seconds. The average evaluation accuracy over 20 runs versus the number of iterations, the wall-clock execution time, and number of function calls for both algorithms are plotted in Figure 6. In each iteration of ZO-EG, there are 2 oracle calls (i.e., 4 function evaluations), whereas the number of function evaluations per iteration in DS can vary. We note that ZO-EG's improved performance

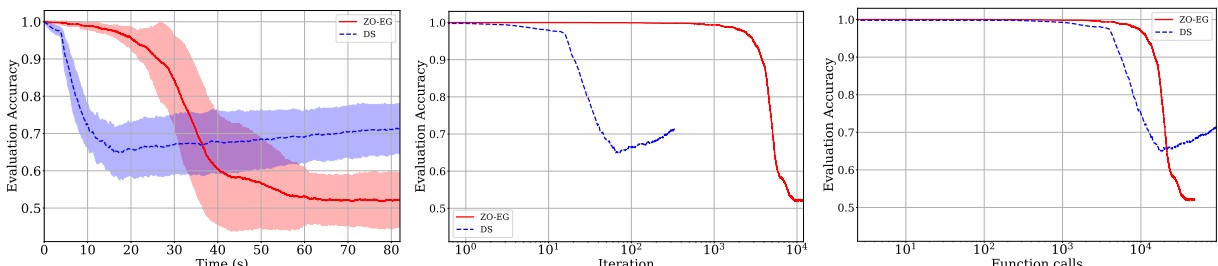

Figure 6: On the left, evaluation accuracy versus the execution time, in the middle, evaluation accuracy versus the number of iterations, and on the right, evaluation accuracy versus the number of function calls for the poisoning attack example of Section 4.4.

(doing so in an increased amount of time) in comparison to DS, i.e., ZO-EG successfully decreases the prediction accuracy to a lower level than DS, as it is the goal of poisoning attacks.

## 4.5 Robust Optimisation

The problem of empirical risk minimisation for a specific class of a binary classification problems is formulated as (Anagnostidis et al., 2021)

$$\min_\theta \max_p \quad -\sum_{i=1}^m p_i[y_i \log(\hat{y}(X_i; \theta)) + (1 - y_i) \log(1 - \hat{y}(X_i; \theta))] - \lambda \sum_{i=1}^m \left(p_i - \frac{1}{m}\right)^2,$$

where $X_i \in \mathbb{R}^v$, $i \in \{1, \ldots, m\}$, are the data points, $\theta \in \mathbb{R}^n$ are the network parameters, and $\hat{y}(X_i; \theta), y_i \in \mathbb{R}^m$ are the predicted and the true class of data points $X_i$, respectively. Moreover, $p \in \mathbb{R}^m$ denotes the weights assigned to each data point. The positive scalar $\lambda$ is the regularisation parameter. The Wisconsin breast cancer data set[2] is considered for this test. This dataset has 569 instances and each instance has 30 ($v = 30$) features. Specifically, we consider the case where the predicted class of a data point $X$, $\hat{y}(X; \theta)$, is generated by a neural network with a hidden layer of size 50 and the LeakyReLU activation function with $n = 1601$ and $m = 513$. We let $\lambda = 0.05$, $h_1 = 10^{-2}$, $h_2 = 10^{-3}$, and $\mu = 10^{-5}$. The min-max DS algorithm is implemented using the same setting as the first test of Anagnostidis et al. (2021) for comparison purposes. In Figure 7, the evolution of the zero-one error and the total error are plotted against the wall-clock time. The algorithm is trained using a 10-fold cross-validation process (Refaeilzadeh et al., 2009). It can be observed that the

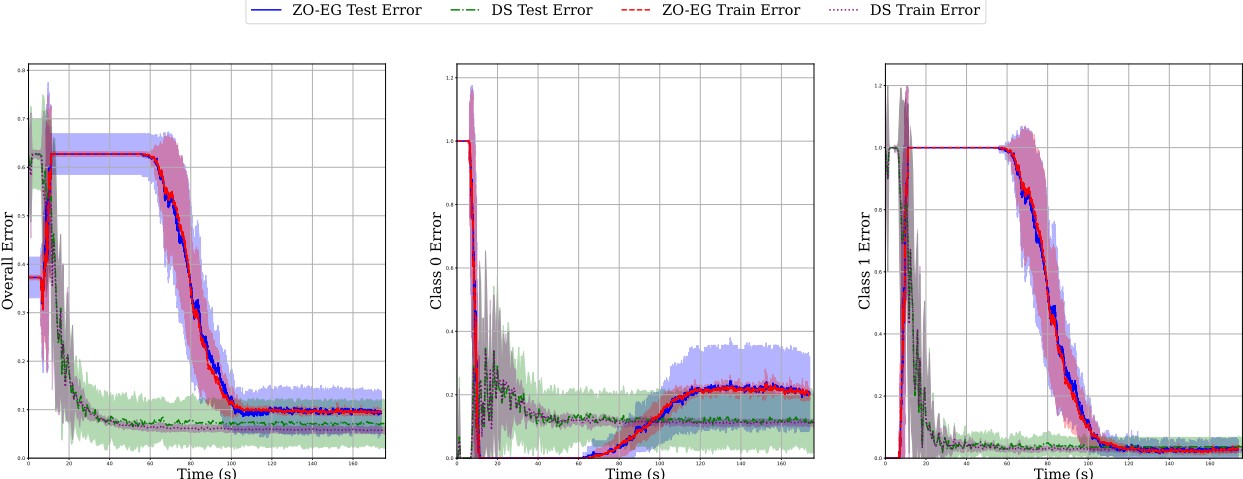

Figure 7: Numerical simulations for the binary classification problem of Section 4.5.

steady-state performance of the algorithms is the same for class 1 and DS performs slightly better for class 0. However, DS leads to a faster decrease in the transient in comparison to ZO-EG. This behaviour is almost similar to that of GDA (Anagnostidis et al., 2021, Appendix D1, Figure 4). It is important to note that this objective function explicitly satisfies a more stringent assumption; it is strongly concave in $p$.

## 4.6 Lane Merging

In this subsection, we present a numerical example of a lane merging problem formulated as a min-max optimisation problem. This scenario involves two cars. The first aims to maximise its velocity while staying in its lane, and the second aims to perform a lane merging maneuver. Both vehicles aim to avoid collision. The kinematic bicycle model is used to simulate the cars' dynamics, represented as:

$$\frac{\mathrm{d}}{\mathrm{d}t} s_i(t) = f(t, s_i(t), u_i(t)) = \begin{bmatrix} v_i(t)\cos(\theta_i(t)) \\ v_i(t)\sin(\theta_i(t)) \\ v_i(t)\tan(\delta_i(t))/L \\ a_i(t) \end{bmatrix}, \quad s_i(t) = \begin{bmatrix} x_i(t) \\ y_i(t) \\ \theta_i(t) \\ v_i(t) \end{bmatrix}, \quad u_i(t) = \begin{bmatrix} a_i(t) \\ \delta_i(t) \end{bmatrix},$$

where $i \in \{1, 2\}$ indicates first or second car, $(x_i, y_i)$ represents the position, $\theta_i$ the heading angle, $v_i$ the velocity, $\delta_i$ the steering angle, $a_i$ the longitudinal acceleration, and $L$ is the car's wheelbase. The input for the first car is defined as $u_1(t) = \begin{bmatrix} a_1(t) & 0 \end{bmatrix}^\top$, while the second car's input is $u_2(t) = \begin{bmatrix} a_2(t) & \delta_2(t) \end{bmatrix}^\top$.

To solve the problem using numerical optimisation, the continuous-time system is discretised using the fourth order Runge-Kutta (RK4) method with a fixed time step $\Delta t > 0$. In RK4, an update is calculated using

---

[2]https://archive.ics.uci.edu/ml/datasets/Breast+Cancer+Wisconsin+(Diagnostic)

four intermediate evaluations of $f$ at different points within the time step:

$$
\begin{aligned}
&K_{1,i,k} = f\big(t_k, s_{i,k}, u_i(t_k)\big), &&K_{2,i,k} = f\big(t_k + \tfrac{\Delta t}{2}, s_{i,k} + \tfrac{\Delta t}{2} K_{1,i,k}, u_i(t_k + \tfrac{\Delta t}{2})\big), \\
&K_{3,i,k} = f\big(t_k + \tfrac{\Delta t}{2}, s_{i,k} + \tfrac{\Delta t}{2} K_{2,i,k}, u_i(t_k + \tfrac{\Delta t}{2})\big), &&K_{4,i,k} = f\big(t_k + \Delta t, s_{i,k} + \Delta t K_{3,i,k}, u_i(t_k + \Delta t)\big).
\end{aligned}
\tag{42}
$$

Here, $s_{i,k}$ denotes the approximation of $s_i(t_k + t)$, $t_k = k\Delta t$, and assuming that the initial time satisfies $t = 0$, the resulting discrete-time dynamics are:

$$
s_{i,k+1} = s_{i,k} + \tfrac{\Delta t}{6}\big(K_{1,i,k} + 2K_{2,i,k} + 2K_{3,i,k} + K_{4,i,k}\big).
$$

where $k \in \mathbb{N}$ denotes the discrete time steps. The discrete-time objective functions for the two cars are:

$$
\begin{aligned}
\Gamma_1(k, s_{1,k}, s_{2,k}) &= \tfrac{1}{2}v_{1,k}^2 - 2\exp\big(-\big((x_{1,k} - x_{2,k})^2 + (y_{1,k} - y_{2,k})^2\big)\big), \\
\Gamma_2(k, s_{1,k}, s_{2,k}) &= \exp\big(-\big((x_{1,k} - x_{2,k})^2 + (y_{1,k} - y_{2,k})^2\big)\big) + 10\big(y_{2,k} - y_{\text{target}}\big)^2,
\end{aligned}
$$

where $y_{\text{target}}$ is the $y$-coordinate of the target lane. These objective functions penalise proximity between the cars to avoid collisions, encourage the first car to increase its velocity, and encourage the second car to reach the target lane. This problem can be solved as an open-loop non-cooperative game over a time horizon of $T > 0$, where we use the parameter $T = 20$ for the numerical example. The control inputs are parametrised using $\Phi = 50$ uniformly spaced control points leading to a sampling time of $\Delta t = 0.4$ second. The inputs are continuous and piecewise linear by assumption, i.e., $u_i(t_k + \tfrac{\Delta t}{2}) = \tfrac{1}{2}(u_i(t_k) + u_i(t_k + \Delta t)))$ is used in (42) for $i \in \{1, 2\}$ and $k \in \mathbb{N}$. The optimisation problem is formulated as:

$$
\begin{aligned}
\min_{U_2 \in \Pi} \max_{U_1 \in \Pi} \quad & \sum_{k=0}^{\Phi-1} \Gamma_1(k, s_{1,k}, s_{2,k}) + \Gamma_2(k, s_{1,k}, s_{2,k}), \\
\text{subject to} \quad & s_{i,k+1} = s_{i,k} + \frac{\Delta t}{6}\big(K_{1,i,k} + 2K_{2,i,k} + 2K_{3,i,k} + K_{4,i,k}\big), \ \forall i \in \{1, 2\},
\end{aligned}
$$

where $U_i = \{u_i(t_k) | \ k \in \{0, \ldots, \Phi - 1\}\}$, $i \in \{1, 2\}$, and $\Pi = \{(a, \delta) \mid a \in [a_{\min}, a_{\max}], \delta \in [\delta_{\min}, \delta_{\max}]\}$. The input dimensions are 50 for $U_1$ and 100 for $U_2$. The steering input for the first car is fixed and does not contribute to the dimensionality.

We implement the ZO-EG algorithm with $y_{\text{target}} = 5$, $\mu = 10^{-6}$, $h_1 = 2 \times 10^{-9}$, $h_2 = 10^{-9}$, and $N = 4000$. The initial states of the cars are $s_1(0) = [0, 5, 0, 2]^\top$ for the first car and $s_2(0) = [5, 0, 0, 3]^\top$ for the second car. Figure 8 shows the evolution of the objective function values over the iterations. The initial and final positions and paths of the cars are depicted in Figure 9.

We also investigate the proximal MVI in this context. Using ZO-EG with the same time horizon $T = 20$, $\Phi = 20$ control values (sampling time $\Delta t = 1$ second), the same step sizes and smoothing parameter, and $N = 7500$ iterations, we generate a candidate point $u_p = (u_{1p}, u_{2p})$ for the answer to the proximal MVI problem (candidate for $z^*$). To evaluate the proximal MVI condition $\langle Q_\ell(u, h_2, F(\bar{u})), \bar{u} - u_p \rangle \geq 0$, where $\ell$ is the indicator function of $\Pi$, we sample 1000 samples for $u$ and $\bar{u}$ separately. All points are sampled from a normal distribution centred at $u_p$. For the acceleration inputs, the covariance matrix is set to $0.1\mathbf{I}$, while for the steering inputs, it is $0.01\mathbf{I}$. Among all tested points, the proximal MVI product is positive, as shown in Figure 10.

## 5 Conclusion And Future Research Directions

The performance of Gaussian ZO random oracles on finding stationary points of nonconvex-nonconcave functions, with or without constraints, differentiable or non-differentiable objective functions, was explored. For the unconstrained problem, the convergence and complexity bounds of the ZO-EG algorithm were studied when applied to nonconvex-nonconcave objective functions. For the constrained problem, we introduced the notion of proximal variational inequalities and established convergence and complexity bounds of the ZO-EG algorithm. We also considered non-differentiable objective functions and obtained convergence and complexity bounds of finding the stationary point of a smoothed function and related that to the original function

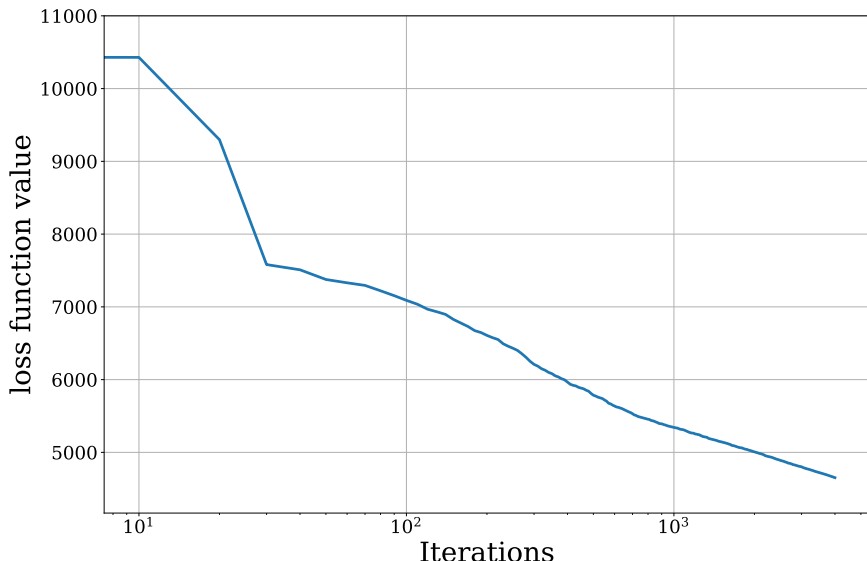

Figure 8: Objective function values versus iterations.

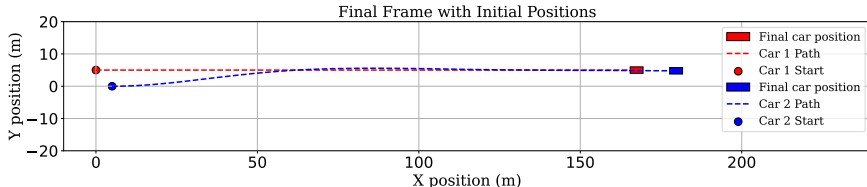

Figure 9: Initial and final positions and paths of the cars.

using the existing literature and the definition of $(\delta,\epsilon)$-Goldstein stationary points. A number of numerical examples were presented to illustrate the findings. A future research direction includes the exploration of the constrained case where the diameter of the constrained set is unbounded. Another potential direction is to study the non-differentiable case, assuming local instead of global Lipschitz continuity properties. Another promising direction for future research is to analyse the second-order optimality conditions of the limit points generated by the ZO-EG algorithm.

### Acknowledgments

We sincerely thank the anonymous reviewers and the Action Editor for their constructive feedback and insights, which significantly strengthened this manuscript. P. Braun and I. Shames are supported by the Australian Research Council through a Discovery Project under Grant DP250101763 and the United States Air Force Office of Scientific Research under Grant No. FA2386-24-1-4014. A. Lesage-Landry and Y. Diouane are supported by the National Science and Engineering Research Council of Canada (NSERC).

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

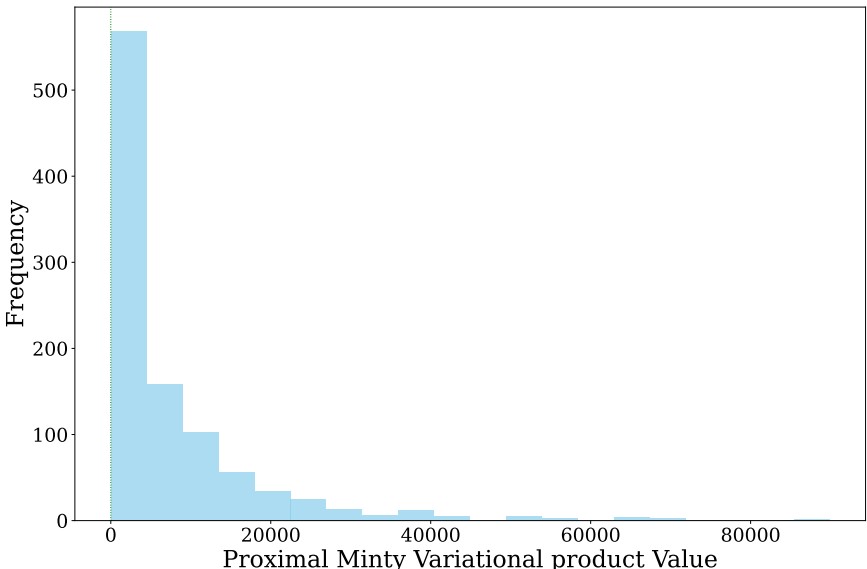

Figure 10: Distribution of proximal MVI values around the optimal solution.

Krishnakumar Balasubramanian and Saeed Ghadimi. Zeroth-order nonconvex stochastic optimization: Handling constraints, high dimensionality, and saddle points. *Foundations of Computational Mathematics*, 22 (1):35–76, 2022.

Heinz H Bauschke, Walaa M Moursi, and Xianfu Wang. Generalized monotone operators and their averaged resolvents. *Mathematical Programming*, 189:55–74, 2021.

Léon Bottou, Frank E Curtis, and Jorge Nocedal. Optimization methods for large-scale machine learning. *SIAM review*, 60(2):223–311, 2018.

Stephen Boyd. Convex optimization. *Cambridge UP*, 2004.

Yang Cai, Constantinos Daskalakis, Haipeng Luo, Chen-Yu Wei, and Weiqiang Zheng. On tractable $\phi$-equilibria in non-concave games. *arXiv preprint arXiv:2403.08171*, 2024.

Giuseppe C Calafiore and Laurent El Ghaoui. *Optimization models*. Cambridge university press, 2014.

Pin-Yu Chen, Huan Zhang, Yash Sharma, Jinfeng Yi, and Cho-Jui Hsieh. Zoo: Zeroth order optimization based black-box attacks to deep neural networks without training substitute models. In *Proceedings of the 10th ACM workshop on artificial intelligence and security*, pp. 15–26, 2017.

Tianyi Chen and Georgios B Giannakis. Bandit convex optimization for scalable and dynamic iot management. *IEEE Internet of Things Journal*, 6(1):1276–1286, 2018.

Frank H Clarke. Generalized gradients and applications. *Transactions of the American Mathematical Society*, 205:247–262, 1975.

Frank H Clarke. *Optimization and Nonsmooth Analysis*. SIAM, 1990.

Patrick L Combettes and Teemu Pennanen. Proximal methods for cohypomonotone operators. *SIAM journal on control and optimization*, 43(2):731–742, 2004.

Cong D Dang and Guanghui Lan. On the convergence properties of non-euclidean extragradient methods for variational inequalities with generalized monotone operators. *Computational Optimization and applications*, 60:277–310, 2015.

Constantinos Daskalakis and Ioannis Panageas. The limit points of (optimistic) gradient descent in min-max optimization. *Advances in neural information processing systems*, 31, 2018.

Jelena Diakonikolas, Constantinos Daskalakis, and Michael I Jordan. Efficient methods for structured nonconvex-nonconcave min-max optimization. In *International Conference on Artificial Intelligence and Statistics*, pp. 2746–2754. PMLR, 2021.

Laurent El Ghaoui and Hervé Lebret. Robust solutions to least-squares problems with uncertain data. *SIAM Journal on Matrix Analysis and Applications*, 18(4):1035–1064, 1997.

Lawrence Craig Evans. *Measure Theory and Fine Properties of Functions*. Routledge, 2018.

F Facchinei. *Finite-dimensional Variational Inequalities and Complementarity Problems*. Springer, 2003.

Amir Ali Farzin and Iman Shames. Minimisation of polyak-łojasewicz functions using random zeroth-order oracles. In *2024 European Control Conference (ECC)*, pp. 3207–3212. IEEE, 2024.

Amir Ali Farzin, Yuen-Man Pun, Philipp Braun, and Iman Shames. Properties of fixed points of generalised extra gradient methods applied to min-max problems. *IEEE Control Systems Letters*, 2025.

Pierre Foret, Ariel Kleiner, Hossein Mobahi, and Behnam Neyshabur. Sharpness-aware minimization for efficiently improving generalization. In *International Conference on Learning Representations*, 2021.

Saeed Ghadimi, Guanghui Lan, and Hongchao Zhang. Mini-batch stochastic approximation methods for nonconvex stochastic composite optimization. *Mathematical Programming*, 155(1-2):267–305, 2016.

Allen A Goldstein. Optimization of lipschitz continuous functions. *Mathematical Programming*, 13:14–22, 1977.

Ian Goodfellow, Jean Pouget-Abadie, Mehdi Mirza, Bing Xu, David Warde-Farley, Sherjil Ozair, Aaron Courville, and Yoshua Bengio. Generative adversarial nets. *Advances in Neural Information Processing Systems*, 27, 2014.

Zhihao Gu and Zi Xu. Zeroth-order stochastic mirror descent algorithms for minimax excess risk optimization. *arXiv preprint arXiv:2408.12209*, 2024.

Ishaan Gulrajani, Faruk Ahmed, Martin Arjovsky, Vincent Dumoulin, and Aaron C Courville. Improved training of wasserstein gans. *Advances in Neural Information Processing Systems*, 30, 2017.

Feihu Huang, Shangqian Gao, Jian Pei, and Heng Huang. Accelerated zeroth-order and first-order momentum methods from mini to minimax optimization. *The Journal of Machine Learning Research*, 23(1):1616–1685, 2022.

Chi Jin, Praneeth Netrapalli, and Michael Jordan. What is local optimality in nonconvex-nonconcave minimax optimization? In *International conference on machine learning*, pp. 4880–4889. PMLR, 2020.

David Kinderlehrer and Guido Stampacchia. *An introduction to Variational Inequalities and Their Applications*. SIAM, 2000.

Guy Kornowski and Ohad Shamir. An algorithm with optimal dimension-dependence for zero-order nonsmooth nonconvex stochastic optimization. *Journal of Machine Learning Research*, 25(122):1–14, 2024.

Ming Lei, Ting Kei Pong, Shuqin Sun, and Man-Chung Yue. Subdifferentially polynomially bounded functions and gaussian smoothing-based zeroth-order optimization. *arXiv preprint arXiv:2405.04150*, 2024.

Tianyi Lin, Zeyu Zheng, and Michael Jordan. Gradient-free methods for deterministic and stochastic nonsmooth nonconvex optimization. *Advances in Neural Information Processing Systems*, 35:26160–26175, 2022.

Sijia Liu, Jie Chen, Pin-Yu Chen, and Alfred Hero. Zeroth-order online alternating direction method of multipliers: Convergence analysis and applications. In *International Conference on Artificial Intelligence and Statistics*, pp. 288–297. PMLR, 2018.

Sijia Liu, Songtao Lu, Xiangyi Chen, Yao Feng, Kaidi Xu, Abdullah Al-Dujaili, Mingyi Hong, and Una-May O'Reilly. Min-max optimization without gradients: Convergence and applications to black-box evasion and poisoning attacks. In *International conference on machine learning*, pp. 6282–6293. PMLR, 2020.

Yuanyuan Liu, Fanhua Shang, Weixin An, Junhao Liu, Hongying Liu, and Zhouchen Lin. A single-loop accelerated extra-gradient difference algorithm with improved complexity bounds for constrained minimax optimization. *Advances in Neural Information Processing Systems*, 36, 2024.

Alejandro I Maass, Chris Manzie, Iman Shames, and Hayato Nakada. Zeroth-order optimization on subsets of symmetric matrices with application to mpc tuning. *IEEE Transactions on Control Systems Technology*, 30(4):1654–1667, 2021.

Aleksander Madry, Aleksandar Makelov, Ludwig Schmidt, Dimitris Tsipras, and Adrian Vladu. Towards deep learning models resistant to adversarial attacks. In *International Conference on Learning Representations*, 2018.

Sadhika Malladi, Tianyu Gao, Eshaan Nichani, Alex Damian, Jason D Lee, Danqi Chen, and Sanjeev Arora. Fine-tuning language models with just forward passes. *Advances in Neural Information Processing Systems*, 36:53038–53075, 2023.

Panayotis Mertikopoulos, Bruno Lecouat, Houssam Zenati, Chuan-Sheng Foo, Vijay Chandrasekhar, and Georgios Piliouras. Optimistic mirror descent in saddle-point problems: Going the extra (gradient) mile. In *ICLR 2019-7th International Conference on Learning Representations*, pp. 1–23, 2019.

Seyed-Mohsen Moosavi-Dezfooli, Alhussein Fawzi, Jonathan Uesato, and Pascal Frossard. Robustness via curvature regularization, and vice versa. In *Proceedings of the IEEE/CVF Conference on Computer Vision and Pattern Recognition*, pp. 9078–9086, 2019.

Katta G Murty and Santosh N Kabadi. Some np-complete problems in quadratic and nonlinear programming. *Mathematical Programming*, 39:117–129, 1987.

Arkadi Nemirovski. Prox-method with rate of convergence o (1/t) for variational inequalities with lipschitz continuous monotone operators and smooth convex-concave saddle point problems. *SIAM Journal on Optimization*, 15(1):229–251, 2004.

Arkadij Semenovič Nemirovskij and David Borisovich Yudin. *Problem Complexity and Method Efficiency in Optimization*. Wiley-Interscience, 1983.

Yurii Nesterov. Introductory lectures on convex programming volume i: Basic course. *Lecture notes*, 3(4):5, 1998.

Yurii Nesterov and Vladimir Spokoiny. Random gradient-free minimization of convex functions. *Foundations of Computational Mathematics*, 17(2):527–566, 2017. ISSN 1615-3375, 1615-3383. doi: 10.1007/s10208-015-9296-2. URL http://link.springer.com/10.1007/s10208-015-9296-2.

Yurii Nesterov et al. *Lectures on Convex Optimization*, volume 137. Springer, 2018.

Thomas Pethick, Puya Latafat, Panagiotis Patrinos, Olivier Fercoq, and Volkan Cevher. Escaping limit cycles: Global convergence for constrained nonconvex-nonconcave minimax problems. *arXiv preprint arXiv:2302.09831*, 2023.

Yuyang Qiu, Uday Shanbhag, and Farzad Yousefian. Zeroth-order methods for nondifferentiable, nonconvex, and hierarchical federated optimization. *Advances in Neural Information Processing Systems*, 36, 2023.

Marco Rando, Cesare Molinari, Lorenzo Rosasco, and Silvia Villa. An optimal structured zeroth-order algorithm for non-smooth optimization. *Advances in Neural Information Processing Systems*, 36, 2024.

Payam Refaeilzadeh, Lei Tang, and Huan Liu. *Cross-Validation*, pp. 532–538. Springer US, Boston, MA, 2009. ISBN 978-0-387-39940-9. doi: 10.1007/978-0-387-39940-9_565. URL https://doi.org/10.1007/978-0-387-39940-9_565.

Luis Miguel Rios and Nikolaos V Sahinidis. Derivative-free optimization: a review of algorithms and comparison of software implementations. *Journal of Global Optimization*, 56:1247–1293, 2013.

Tim Salimans, Jonathan Ho, Xi Chen, Szymon Sidor, and Ilya Sutskever. Evolution strategies as a scalable alternative to reinforcement learning. *arXiv preprint arXiv:1703.03864*, 2017.

Jasper Snoek, Hugo Larochelle, and Ryan P Adams. Practical bayesian optimization of machine learning algorithms. *Advances in neural information processing systems*, 25, 2012.

Chaobing Song, Zhengyuan Zhou, Yichao Zhou, Yong Jiang, and Yi Ma. Optimistic dual extrapolation for coherent non-monotone variational inequalities. *Advances in Neural Information Processing Systems*, 33: 14303–14314, 2020.

Lai Tian, Kaiwen Zhou, and Anthony Man-Cho So. On the finite-time complexity and practical computation of approximate stationarity concepts of lipschitz functions. In *International Conference on Machine Learning*, pp. 21360–21379. PMLR, 2022.

Luís Nunes Vicente. Worst case complexity of direct search. *EURO Journal on Computational Optimization*, 1(1):143–153, 2013.

Chi Wang, Qingyun Wu, Markus Weimer, and Erkang Zhu. Flaml: A fast and lightweight automl library. *Proceedings of Machine Learning and Systems*, 3:434–447, 2021.

Zhongruo Wang, Krishnakumar Balasubramanian, Shiqian Ma, and Meisam Razaviyayn. Zeroth-order algorithms for nonconvex–strongly-concave minimax problems with improved complexities. *Journal of Global Optimization*, 87(2):709–740, 2023.

Tengyu Xu, Zhe Wang, Yingbin Liang, and H Vincent Poor. Gradient free minimax optimization: Variance reduction and faster convergence. *arXiv preprint arXiv:2006.09361*, 2020.

Zi Xu, Zi-Qi Wang, Jun-Lin Wang, and Yu-Hong Dai. Zeroth-order alternating gradient descent ascent algorithms for a class of nonconvex-nonconcave minimax problems. *Journal of Machine Learning Research*, 24(313):1–25, 2023.

Junchi Yang, Negar Kiyavash, and Niao He. Global convergence and variance-reduced optimization for a class of nonconvex-nonconcave minimax problems. *arXiv preprint arXiv:2002.09621*, 2020.

Jingzhao Zhang, Hongzhou Lin, Stefanie Jegelka, Suvrit Sra, and Ali Jadbabaie. Complexity of finding stationary points of nonconvex nonsmooth functions. In *International Conference on Machine Learning*, pp. 11173–11182. PMLR, 2020.

Taoli Zheng, Linglingzhi Zhu, Anthony Man-Cho So, José Blanchet, and Jiajin Li. Universal gradient descent ascent method for nonconvex-nonconcave minimax optimization. *Advances in Neural Information Processing Systems*, 36:54075–54110, 2023.

Zhengyuan Zhou, Panayotis Mertikopoulos, Nicholas Bambos, Stephen Boyd, and Peter W Glynn. Stochastic mirror descent in variationally coherent optimization problems. *Advances in Neural Information Processing Systems*, 30, 2017.

## A    Complementary Lemmas, Corollaries and Remarks

The following lemmas are adopted from Nesterov & Spokoiny (2017). The results are used in the proofs of the main results.

**Lemma 5** ((Nesterov & Spokoiny, 2017, Lemma 1)). *Let $u \in \mathbf{Z}$ be sampled from Gaussian distribution $\mathcal{N}(0, B^{-1})$. If we define $M_p \stackrel{\text{def}}{=} \frac{1}{\kappa} \int \|u\|^p e^{-\frac{1}{2}\|u\|^2} \mathrm{d}u$. For $p \in [0, 2]$ we have $M_p \leq d^{\frac{p}{2}}$. For $p \geq 2$ we have $d^{\frac{p}{2}} \leq M_p \leq (p+d)^{\frac{p}{2}}$.*

**Lemma 6** ((Nesterov & Spokoiny, 2017, Theorem 1)). *Let $f : \mathbf{Z} \to \mathbb{R}$ be continuously differentiable with Lipschitz gradients with constant $L_1(f) > 0$, and $f_\mu$ be defined in (10). Then*

$$|f_\mu(z) - f(z)| \leq \frac{\mu^2}{2} L_1(f) d, \quad \forall z \in \mathbf{Z}, \tag{43}$$

*where $f_\mu$ is Gaussian smoothed version of $f$.*

**Lemma 7** ((Nesterov & Spokoiny, 2017, Lemma 3)). *Let $f : \mathbf{Z} \to \mathbb{R}$ be continuously differentiable with Lipschitz continuous gradient with constant $L_1(f) > 0$, and $f_\mu$ be defined in (10). Then*

$$\|\nabla f_\mu(z) - \nabla f(z)\|_* \leq \frac{\mu}{2} L_1(f)(d+3)^{\frac{3}{2}}, \quad \forall z \in \mathbf{Z}. \tag{44}$$

We continue with a proof of Lemma 1 introduced in Section 3.1.

*Proof of Lemma 1.* Since $z^* \in \mathbf{Z}$ satisfies the weak MVI (9) by Assumption 2, we know that $\langle F(z), z - z^* \rangle + \frac{\rho}{2}\|F(z)\|_*^2 \geq 0$, for all $z \in \mathbf{Z}$. Hence replacing $z$ with $z + \mu u$ in (9), we have

$$0 \leq \langle F(z + \mu u), z + \mu u - z^* \rangle + \frac{\rho}{2}\|F(z + \mu u)\|_*^2$$
$$= \langle F(z + \mu u), z - z^* \rangle + \langle F(z + \mu u), \mu u \rangle + \frac{\rho}{2}\|F(z + \mu u)\|_*^2. \tag{45}$$

Also, considering that the gradient of $f$ is Lipschitz continuous, we have (Nesterov, 1998)

$$f(x_1, y) \leq f(x_2, y) + \langle \nabla_x f(x_2, y), x_1 - x_2 \rangle + \frac{L_1}{2}\|x_1 - x_2\|^2$$

and

$$f(x, y_1) \leq f(x, y_2) + \langle \nabla_y f(x, _2 y), y_1 - y_2 \rangle + \frac{L_1}{2}\|y_1 - y_2\|^2.$$

Considering above inequalities, we get

$$f(x, y + \mu u_2) \leq f(x + \mu u_1, y + \mu u_2) + \langle \nabla_x f(x + \mu u_1, y + \mu u_2), -\mu u_1 \rangle + \frac{L_1 \mu^2}{2}\|u_1\|^2.$$

Since $f$ and $-f$ satisfy the same Lipschitz continuity properties, it holds that

$$-f(x + \mu u_1, y) \leq -f(x + \mu u_1, y + \mu u_2) + \langle -\nabla_y f(x + \mu u_1, y + \mu u_2), -\mu u_2 \rangle + \frac{L_1 \mu^2}{2}\|u_2\|^2.$$

Thus

$$\langle \nabla_x f(x + \mu u_1, y + \mu u_2), \mu u_1 \rangle \leq f(x + \mu u_1, y + \mu u_2) - f(x, y + \mu u_2) + \frac{L_1 \mu^2}{2}\|u_1\|^2$$

and

$$\langle -\nabla_y f(x + \mu u_1, y + \mu u_2), \mu u_2 \rangle \leq f(x + \mu u_1, y) - f(x + \mu u_1, y + \mu u_2) + \frac{L_1 \mu^2}{2}\|u_2\|^2.$$

Adding the above two inequalities, we have

$$\langle F(z + \mu u), \mu u \rangle \leq \frac{L_1 \mu^2}{2}\|u\|^2 + f(x + \mu u_1, y) - f(x, y + \mu u_2),$$

where $z = (x, y)$ and $u = [u_1, u_2]$. Now adding and subtracting $f(x, y)$ we have

$$\langle F(z + \mu u), \mu u \rangle \leq \frac{L_1 \mu^2}{2} \|u\|^2 + (f(x + \mu u_1, y) - f(x, y)) + (f(x, y) - f(x, y + \mu u_2)). \tag{46}$$

Moreover, recalling the definition of $F_\mu$ in (15), it holds that

$$\begin{aligned}
\|F(z + \mu u)\|_*^2 &= \|F_\mu(z) + (F(z) - F_\mu(z)) + (F(z + \mu u) - F(z))\|_*^2 \\
&\leq 2\|F_\mu(z)\|_*^2 + 2\|(F(z) - F_\mu(z)) + (F(z + \mu u) - F(z))\|_*^2 \\
&\leq 2\|F_\mu(z)\|_*^2 + 4\|(F(z) - F_\mu(z))\|_*^2 + 4\|(F(z + \mu u) - F(z))\|_*^2 \\
&\leq 2\|F_\mu(z)\|_*^2 + 4\|(F(z) - F_\mu(z))\|_*^2 + 4\mu^2 L_1^2(f)\|u\|^2,
\end{aligned}$$

where the last inequality is due to the Lipschitz continuity of $F$. Thus considering Lemma 7, it holds that

$$\|F(z + \mu u)\|_*^2 \leq 2\|F_\mu(z)\|_*^2 + \mu^2 L_1^2(d + 3)^3 + 4\mu^2 L_1^2(f)\|u\|^2. \tag{47}$$

Substituting (46) and (47) in (45), we have

$$\begin{aligned}
0 \leq &\langle F(z + \mu u), z - z^* \rangle + \frac{L_1 \mu^2}{2} \|u\|^2 + \rho \|F_\mu(z)\|_*^2 + \frac{\rho}{2} \mu^2 L_1^2(d + 3)^3 + 2\rho\mu^2 L_1^2 \|u\|^2 \\
&+ (f(x + \mu u_1, y) - f(x, y)) + (f(x, y) - f(x, y + \mu u_2)). \tag{48}
\end{aligned}$$

Computing the expected value with respect to $u$ the estimate

$$\begin{aligned}
0 \leq &\langle F_\mu(z), z - z^* \rangle + \frac{L_1 \mu^2 d}{2} + \rho \|F_\mu(z)\|_*^2 + \frac{\rho}{2} \mu^2 L_1^2(d + 3)^3 + 2\rho\mu^2 L_1^2 d \\
&+ (f_{\mu,x}(x, y) - f(x, y)) + (f(x, y) - f_{\mu,y}(x, y)).
\end{aligned}$$

is obtained and where $f_{\mu,x} = E_{u_1}[f(x + \mu u_1, y)]$ and $f_{\mu,y} = E_{u_2}[f(x, y + \mu u_2)]$ are the Gaussian smoothed functions of $f$ with respect to only $x$ and $y$, respectively. Using Lemma 6 we have

$$0 \leq \langle F_\mu(z), z - z^* \rangle + \frac{L_1 \mu^2 d}{2} + \rho \|F_\mu(z)\|_*^2 + \frac{\rho}{2} \mu^2 L_1^2(d + 3)^3 + 2\rho\mu^2 L_1^2 d + \frac{L_1 \mu^2 n}{2} + \frac{L_1 \mu^2 m}{2}.$$

With $n + m = d$ and using the fact that $2d + \frac{(d+3)^3}{2} \leq (d + 3)^3$ for all $d \geq 2$, the last expression can be simplified to

$$\langle F_\mu(z), z - z^* \rangle + \rho \|F_\mu(z)\|_*^2 + \mu^2 L_1(f)d + \rho\mu^2 L_1^2(f)(d + 3)^3 \geq 0, \tag{49}$$

which completes the proof. $\qquad\square$

**Remark 7.** *Extending $Q_\ell(x, a, F(\bar{z}))$, we have*

$$Q_\ell(z, a, F(\bar{z})) = -\frac{1}{a}(\arg\min_x [\|x - z\|^2 + 2a\langle F(\bar{z}), B(x - z)\rangle + \ell(x)] - z).$$

*Then we have*

$$\begin{aligned}
x' &= \arg\min_x [\|x - z\|^2 + 2a\langle F(\bar{z}), B(x - z)\rangle + \ell(x)] \\
&= \arg\min_x [\frac{1}{a^2}\|x - z\|^2 + \frac{2}{a}\langle F(\bar{z}), B(x - z)\rangle + \frac{1}{a^2}(\ell(x))] \\
&= \arg\min_x [\frac{1}{a^2}\|x - z\|^2 + \frac{2}{a}\langle F(\bar{z}), B(x - z)\rangle + \|F(\bar{z})\|^2 + \frac{1}{a^2}(\ell(x)))] \\
&= \arg\min_x [\|\frac{1}{a}(x - z) + F(\bar{z})\|^2 + \frac{1}{a^2}(\ell(x))] \\
&= \arg\min_x [\|x - (z - aF(\bar{z}))\|2 + \ell(x)]
\end{aligned}$$

*Thus, if $\ell(x) = 0$ or $\ell$ is a constant, then $x' = z - aF(\bar{z})$ and $Q_\ell(z, a, F(\bar{z})) = F(\bar{z})$ independent of positive scalar $a$.*

Before we continue with a proof of Lemma 2, we introduce the auxiliary variables

$$v_k \overset{\text{def}}{=} P_{\mathcal{Z}}(z_k, h_{1,k}, F_\mu(z_k)), \quad \hat{v}_k \overset{\text{def}}{=} P_{\mathcal{Z}}(z_k, h_{2,k}, F_\mu(\hat{z}_k)), \quad p_k \overset{\text{def}}{=} P_{\mathcal{Z}}(z_k, h_{1,k}, F(z_k)), \tag{50}$$

to simplify the presentation in the following.

*Proof of Lemma 2.* Let $s_k$ and $\hat{s}_k$, be defined in (28), $\hat{p}_k$ be defined in (34), and $v_k$, $\hat{v}_k$, and $p_k$ be defined in (50). Considering Algorithm 1 and $\Gamma(z)$ along with $\mathcal{Z}$ as defined in Section 3.2 when $\ell(z) = I_{\mathcal{Z}}(z)$, we have $\text{Prox}_\ell(\cdot) = \text{Proj}_{\mathcal{Z}}(\cdot)$, $Q_\ell(z_k, h_1, F(z_k)) = p_k$ and $Q_\ell(z_k, h_2, F(\hat{z}_k)) = \hat{p}_k$. Thus, when Assumption 3 is satisfied, we have

$$\langle p_k, z_k - z^* \rangle + \frac{\rho}{2}\|p_k\|_*^2 \geq 0, \qquad \langle \hat{p}_k, \hat{z}_k - z^* \rangle + \frac{\rho}{2}\|\hat{p}_k\|_*^2 \geq 0.$$

Considering the above inequalities, we have

$$
\begin{aligned}
0 &\leq \langle p_k, z_k - z^* \rangle + \frac{\rho}{2}\|p_k\|_*^2 \\
&= \langle s_k, z_k - z^* \rangle + \langle p_k - v_k, z_k - z^* \rangle + \langle v_k - s_k, z_k - z^* \rangle + \frac{\rho}{2}\|s_k + (p_k - v_k) + (v_k - s_k)\|_*^2 \\
&\leq \langle s_k, z_k - z^* \rangle + |p_k - v_k||z_k - z^*| + \langle v_k - s_k, z_k - z^* \rangle + \rho\|s_k\|_*^2 + \rho\|(p_k - v_k) + (v_k - s_k)\|_*^2 \\
&\leq \langle s_k, z_k - z^* \rangle + D_z\kappa\|F(z_k) - F_\mu(z_k)\|_* + D_z\kappa\|\xi_k\|_* + \rho\|s_k\|_*^2 + 2\rho\kappa^2\|F(z_k) - F_\mu(z_k)\|_*^2 + 2\kappa^2\rho\|\xi_k\|_*^2 \\
&\leq \langle s_k, z_k - z^* \rangle + \frac{\mu}{2}D_z\kappa L_1(f)(d+3)^{3/2} + D_z\kappa\|\xi_k\|_* + \rho\|s_k\|_*^2 + \frac{\mu^2}{2}\rho\kappa^2 L_1^2(f)(d+3)^3 + 2\rho\kappa^2\|\xi_k\|_*^2. \tag{51}
\end{aligned}
$$

The third inequality is due to $\|p_k - v_k\| \leq \|F(z_k) - F_\mu(z_k)\|$ and $\|v_k - s_k\| \leq \|F_\mu(z_k) - G_\mu(z_k)\|$ which can be obtained directly from the non-expansiveness of the projection operator. The last inequality is due to Lemma 7. Inequality (51) proves (30). A proof of (31) follows the same arguments. $\qquad\square$

# B    Proof of Theorems and Corollaries

In this section, we give proofs of the main results presented in this paper.

*Proof of Theorem 1.* Considering Lemma 1, $h_2 > 0$, and letting

$$\xi_k := G_\mu(z_k) - F_\mu(z_k) \quad \text{and} \quad \hat{\xi}_k := G_\mu(\hat{z}_k) - F_\mu(\hat{z}_k) \tag{52}$$

(with $E_{u_k}[\hat{\xi}_k] = 0$ and $E_{\hat{u}_k}[\xi_k] = 0$), we have

$$
\begin{aligned}
0 &\leq h_2\langle F_\mu(\hat{z}_k), \hat{z}_k - z^* \rangle + h_2\rho\|F_\mu(\hat{z}_k)\|_*^2 + h_2\mu^2 L_1 d + h_2\rho\mu^2 L_1^2(d+3)^3 \\
&= h_2\langle G_\mu(\hat{z}_k), \hat{z}_k - z^* \rangle - h_2\langle \hat{\xi}_k, \hat{z}_k - z^* \rangle + h_2\rho\|F_\mu(\hat{z}_k)\|_*^2 + h_2\mu^2 L_1 d + h_2\rho\mu^2 L_1^2(d+3)^3 \\
&= h_2\langle G_\mu(\hat{z}_k), z_{k+1} - z^* \rangle + h_2\langle G_\mu(\hat{z}_k), \hat{z}_k - z_{k+1} \rangle + h_2\langle G_\mu(\hat{z}_k) - G_\mu(\hat{z}_k), \hat{z}_k - z_{k+1} \rangle \tag{53a} \\
&\quad - h_2\langle \hat{\xi}_k, \hat{z}_k - z^* \rangle + h_2\rho\|F_\mu(\hat{z}_k)\|_*^2 + h_2\mu^2 L_1 d + h_2\rho\mu^2 L_1^2(d+3)^3 \tag{53b}
\end{aligned}
$$

Here, we have additionally used $L_1 = L_1(f)$ to shorten the expressions. As a next step, we derive a bound for the three terms in (53a). Considering $\mathcal{Z} = \mathbf{Z}$ and from Algorithm 1 line 8, we know that $z_k - z_{k+1} = h_2 G_\mu(\hat{z}_k)$ as $\text{Proj}_{\mathcal{Z}}(z) = z$. Thus, it holds that

$$
\begin{aligned}
h_2\langle G_\mu(\hat{z}_k), z_{k+1} - z^* \rangle &= \langle z_k - z_{k+1}, z_{k+1} - z^* \rangle \\
&= \frac{1}{2}|z^* - z_k|^2 - \frac{1}{2}|z^* - z_{k+1}|^2 - \frac{1}{2}|z_k - z_{k+1}|^2. \tag{54}
\end{aligned}
$$

Similarly, from Algorithm 1 line 5, we know that $z_k - \hat{z}_k = h_1 G_\mu(z_k)$ and thus the estimate

$$
\begin{aligned}
h_2 \langle G_\mu(z_k), \hat{z}_k - z_{k+1} \rangle &= \frac{h_2}{h_1} \langle z_k - \hat{z}_k, \hat{z}_k - z_{k+1} \rangle \\
&= \frac{h_2}{2h_1} (|z_k - z_{k+1}|^2 - |z_k - \hat{z}_k|^2 - |z_{k+1} - \hat{z}_k|^2) \\
&= \frac{h_2}{2h_1} |z_k - z_{k+1}|^2 - \frac{h_2}{2h_1\overline{\lambda}} \|z_k - \hat{z}_k\|^2 - \frac{h_2}{2h_1\overline{\lambda}} \|z_{k+1} - \hat{z}_k\|^2)
\end{aligned}
\tag{55}
$$

is obtained. For the third term in (53a), we use the fact that the gradient of $f$ is Lipschitz continuous. Hence, for any $\alpha > 0$, the chain of inequalities

$$
\begin{aligned}
h_2 \langle G_\mu(\hat{z}_k) - G_\mu(z_k), \hat{z}_k - z_{k+1} \rangle &\le h_2 |F_\mu(\hat{z}_k) - F_\mu(z_k)| |\hat{z}_k - z_{k+1}| + h_2 \langle \hat{\xi}_k - \xi_k, \hat{z}_k - z_{k+1} \rangle \\
&\le h_2 L_1 \kappa \|\hat{z}_k - z_k\| \|\hat{z}_k - z_{k+1}\| + h_2 \langle \hat{\xi}_k - \xi_k, \hat{z}_k - z_{k+1} \rangle \\
&\le \frac{h_2 L_1 \kappa \alpha}{2} \|\hat{z}_k - z_k\|^2 + \frac{h_2 L_1 \kappa}{2\alpha} \|\hat{z}_k - z_{k+1}\|^2 + h_2 \langle \hat{\xi}_k - \xi_k, \hat{z}_k - z_{k+1} \rangle,
\end{aligned}
\tag{56}
$$

is satisfied.

Substituting (54), (55), and (56) in (53a), letting $r_k = |z_k - z^*|$, and noting that $z_k - z_{k+1} = h_2 G_\mu(\hat{z}_k)$ and $z_k - \hat{z}_k = h_1 G_\mu(z_k)$, we get

$$
\begin{aligned}
0 \le\ & \frac{1}{2}(r_k^2 - r_{k+1}^2) + \left(\frac{h_2}{2h_1} - \frac{1}{2}\right) |z_k - z_{k+1}|^2 + \left(\frac{h_2 L_1 \kappa \alpha}{2} - \frac{h_2}{2h_1\overline{\lambda}}\right) \|\hat{z}_k - z_k\|^2 + h_2 \rho \|F_\mu(\hat{z}_k)\|_*^2 \\
& + \left(\frac{h_2 L_1 \kappa}{2\alpha} - \frac{h_2}{2h_1\overline{\lambda}}\right) \|\hat{z}_k - z_{k+1}\|^2 - h_2 \langle \hat{\xi}_k, \hat{z}_k - z^* \rangle + h_2 \mu^2 L_1 d + h_2 \rho \mu^2 L_1^2 (d+3)^3 \\
& + h_2 \langle \hat{\xi}_k - \xi_k, \hat{z}_k - z_{k+1} \rangle \\
=\ & \frac{1}{2}(r_k^2 - r_{k+1}^2) + h_2^2 \left(\frac{h_2}{2h_1} - \frac{1}{2}\right) |G_\mu(\hat{z}_k)|^2 + h_1^2 \left(\frac{h_2 L_1 \kappa \alpha}{2} - \frac{h_2}{2h_1\overline{\lambda}}\right) \|G_\mu(z_k)\|^2 + h_2 \rho \|F_\mu(\hat{z}_k)\|_*^2 \\
& + \left(\frac{h_2 L_1 \kappa}{2\alpha} - \frac{h_2}{2h_1\overline{\lambda}}\right) \|\hat{z}_k - z_{k+1}\|^2 - h_2 \langle \hat{\xi}_k, \hat{z}_k - z^* \rangle + h_2 \mu^2 L_1 d + h_2 \rho \mu^2 L_1^2 (d+3)^3 \\
& + h_2 \langle \hat{\xi}_k - \xi_k, \hat{z}_k - z_{k+1} \rangle.
\end{aligned}
\tag{57}
$$

Rearranging the above terms we have

$$
\begin{aligned}
h_2^2 \left(\frac{1}{2} - \frac{h_2}{2h_1}\right) |G_\mu(\hat{z}_k)|^2 \le\ & \frac{1}{2}(r_k^2 - r_{k+1}^2) + h_1^2 \left(\frac{h_2 L_1 \kappa \alpha}{2} - \frac{h_2}{2h_1\overline{\lambda}}\right) \|G_\mu(z_k)\|^2 + h_2 \rho \|F_\mu(\hat{z}_k)\|_*^2 \\
& + \left(\frac{h_2 L_1 \kappa}{2\alpha} - \frac{h_2}{2h_1\overline{\lambda}}\right) \|\hat{z}_k - z_{k+1}\|^2 - h_2 \langle \hat{\xi}_k, \hat{z}_k - z^* \rangle + h_2 \mu^2 L_1 d + h_2 \rho \mu^2 L_1^2 (d+3)^3 \\
& + h_2 \langle \hat{\xi}_k - \xi_k, h_2 G_\mu(\hat{z}_k) - h_1 G_\mu(z_k) \rangle.
\end{aligned}
\tag{58}
$$

Choosing $h_1 \le \frac{1}{\overline{\lambda} L_1 \kappa}$ and $\alpha = 1$ ensures that the second and third right-hand side terms of (58) are non-positive. For $\frac{1}{2} - \frac{h_2}{2h_1} > 0$ to hold, $h_2$ needs to satisfy $h_2 < h_1$. Considering these facts, we choose $\sqrt{\frac{2\rho}{L_1 \underline{\lambda} \overline{\lambda} \kappa}} \le h_2 \le \frac{h_1}{2}$. Considering Definition 10, $\rho \in [0, \frac{1}{8L\kappa^2})$ and we can guarantee that there exists $h_1$ and $h_2$ such that $\sqrt{\frac{2\rho}{L_1 \underline{\lambda} \overline{\lambda} \kappa}} \le h_2 \le \frac{h_1}{2}$. Thus, we have

$$
\begin{aligned}
\frac{h_2^2}{4} |G_\mu(\hat{z}_k)|^2 - \frac{\rho}{2L_1 \underline{\lambda} \overline{\lambda} \kappa} |F_\mu(\hat{z}_k)|^2 \le\ & \frac{1}{2}(r_k^2 - r_{k+1}^2) - h_2 \langle \hat{\xi}_k, \hat{z}_k - z^* \rangle + \frac{\mu^2 d}{2\overline{\lambda}\kappa} + \frac{\rho}{2\overline{\lambda}\kappa} \mu^2 L_1 (d+3)^3 \\
& + h_2 \langle \hat{\xi}_k - \xi_k, h_2 G_\mu(\hat{z}_k) - h_1 G_\mu(z_k) \rangle \\
\le\ & \frac{1}{2}(r_k^2 - r_{k+1}^2) + h_2 \langle \hat{\xi}_k, z^* - z_k + h_1 G_\mu(z_k) \rangle + \frac{\mu^2 d}{2\overline{\lambda}\kappa} + \frac{\rho}{2\overline{\lambda}\kappa} \mu^2 L_1 (d+3)^3 \\
& + h_2 \langle \hat{\xi}_k - \xi_k, h_2 G_\mu(\hat{z}_k) - h_1 G_\mu(z_k) \rangle.
\end{aligned}
\tag{59}
$$

The second inequality is due to the fact that $\hat{z}_k = z_k - h_1 G_\mu(z_k)$ considering Algorithm 1 line 5. For the last term of the inequality above, we have

$$
\begin{aligned}
h_2\langle \hat{\xi}_k - \xi_k, h_2 G_\mu(\hat{z}_k) - h_1 G(z_k)\rangle &= h_2\langle \hat{\xi}_k - \xi_k, h_2(\hat{\xi}_k + F_\mu(\hat{z}_k)) - h_1(\xi_k + F_\mu(z_k))\rangle \\
&= h_2\langle \hat{\xi}_k - \xi_k, h_2\hat{\xi}_k - h_1\xi_k\rangle + h_2\langle \hat{\xi}_k - \xi_k, h_2 F_\mu(\hat{z}_k) - h_1 F_\mu(z_k)\rangle.
\end{aligned} \tag{60}
$$

For $h_2\langle \hat{\xi}_k - \xi_k, h_2\hat{\xi}_k - h_1\xi_k\rangle$, we have

$$
h_2\langle \hat{\xi}_k - \xi_k, h_2\hat{\xi}_k - h_1\xi_k\rangle = h_2^2|\hat{\xi}_k|^2 + h_2 h_1|\xi_k|^2 + h_2\langle \xi_k, -h_2\hat{\xi}_k\rangle + h_2\langle \hat{\xi}_k, -h_1\xi_k\rangle. \tag{61}
$$

Substituting (60) and (61) in (59), we have

$$
\begin{aligned}
\frac{h_2^2}{4}|G_\mu(\hat{z}_k)|^2 - \frac{\rho}{2L_1\underline{\lambda}\overline{\lambda}\kappa}|F_\mu(\hat{z}_k)|^2 &\leq \frac{1}{2}(r_k^2 - r_{k+1}^2) + \frac{\mu^2 d}{2\overline{\lambda}\kappa} + \frac{\rho}{2\overline{\lambda}\kappa}\mu^2 L_1(d+3)^3 + h_2^2|\hat{\xi}_k|^2 + h_2 h_1|\xi_k|^2 \\
&\quad + h_2\langle \hat{\xi}_k, z^* - z_k + h_1 G_\mu(z_k)\rangle + h_2\langle \hat{\xi}_k - \xi_k, h_2 F_\mu(\hat{z}_k) - h_1 F_\mu(z_k)\rangle + h_2\langle \xi_k, -h_2\hat{\xi}_k\rangle + h_2\langle \hat{\xi}_k, -h_1\xi_k\rangle.
\end{aligned} \tag{62}
$$

From Jensen's inequality, we know that $E_{u_k}[|G_\mu(\hat{z}_k)|]^2 \leq E_{u_k}[|G_\mu(\hat{z}_k)|^2]$. Additionally, it can be concluded that $E_{u_k}[|G_\mu(\hat{z}_k)|] \geq |E_{u_k}[G_\mu(\hat{z}_k)]| = |F_\mu(\hat{z}_k)|$, and thus

$$
E_{u_k}[|G_\mu(\hat{z}_k)|^2] \geq |F_\mu(\hat{z}_k)|^2. \tag{63}
$$

Using this inequality we can lower bound $E_{u_k}[|G_\mu(\hat{z}_k)|^2]$ and by taking the expected value of (62) with respect to $u_k$ and then with respect to $\hat{u}_k$, noting that $E_{u_k}[\hat{\xi}_k] = 0$, $E_{u_k}[|\hat{\xi}_k|^2] \leq \overline{\lambda}E_{u_k}[\|\hat{\xi}_k\|_*^2] \leq \overline{\lambda}\sigma^2$, $E_{\hat{u}_k}[\xi_k] = 0$, and $E_{\hat{u}_k}[|\xi_k|^2] \leq \overline{\lambda}E_{\hat{u}_k}[\|\xi_k\|_*^2] \leq \overline{\lambda}\sigma^2$, we have

$$
\begin{aligned}
\left(\frac{h_2^2}{4} - \frac{\rho}{2L_1\underline{\lambda}\overline{\lambda}\kappa}\right) E_{u_k,\hat{u}_k}[|F_\mu(\hat{z}_k)|^2] &\leq \frac{1}{2}(r_k^2 - E_{u_k,\hat{u}_k}[r_{k+1}^2]) + \frac{\mu^2 d}{2\overline{\lambda}\kappa} + \frac{\rho}{2\overline{\lambda}\kappa}\mu^2 L_1(d+3)^3 \\
&\quad + \frac{1}{4L_1^2\overline{\lambda}\kappa^2}\sigma^2 + \frac{1}{2L_1^2\overline{\lambda}\kappa^2}\sigma^2.
\end{aligned} \tag{64}
$$

Since $u_k$ and $\hat{u}_k$ are independent by assumption, $\xi_k$, $z_k$, and $\hat{z}_k$ are independent of $u_k$ ($E_{u_k}[\hat{\xi}_k] = 0$ and $E_{u_k}[\xi_k] = \xi_k$,) the expected value of the last four terms of (62) with respected to $u_k$ are zero. The technique of taking the expectation with respect to the last sampled random variable first, and then with respect to the history has been leveraged before in ZO optimisation, e.g., (Nesterov & Spokoiny, 2017, (67)).

Next, let $\mathcal{U}_k = [(u_0, \hat{u}_0), (u_1, \hat{u}_1), \cdots, (u_k, \hat{u}_k)]$ for $k \in \{0, \ldots, N\}$. Computing the expected value of (64) with respect to $\mathcal{U}_{k-1}$, letting $\phi_k = E_{\mathcal{U}_{k-1}}[r_k^2]$ and $\phi_0^2 = r_0^2$, we have

$$
\left(\frac{h_2^2}{4} - \frac{\rho}{2L_1\underline{\lambda}\overline{\lambda}\kappa}\right) E_{\mathcal{U}_k}[|F_\mu(\hat{z}_k)|^2] \leq \frac{1}{2}(\phi_k^2 - \phi_{k+1}^2) + \frac{\mu^2 d}{2\overline{\lambda}\kappa} + \frac{\rho}{2\overline{\lambda}\kappa}\mu^2 L_1(d+3)^3 + \frac{1}{4L_1^2\overline{\lambda}\kappa^2}\sigma^2 + \frac{1}{2L_1^2\overline{\lambda}\kappa^2}\sigma^2. \tag{65}
$$

Summing (65) from $k = 0$ to $k = N$, and dividing it by $N + 1$, yields

$$
\begin{aligned}
\frac{1}{N+1}\sum_{k=0}^{N} E_{\mathcal{U}_k}[|F_\mu(\hat{z}_k)|^2] &\leq \frac{2\underline{\lambda}\overline{\lambda}L_1\kappa|z_0 - z^*|^2}{(\overline{\lambda}\underline{\lambda}L_1\kappa h_2^2 - 2\rho)(N+1)} + \frac{2\mu^2\underline{\lambda}L_1 d}{(\overline{\lambda}\underline{\lambda}L_1\kappa h_2^2 - 2\rho)} + \frac{2\mu^2\underline{\lambda}L_1^2\rho(d+3)^3}{(\overline{\lambda}\underline{\lambda}L_1\kappa h_2^2 - 2\rho)} \\
&\quad + \frac{3\underline{\lambda}\sigma^2}{L_1\kappa(\overline{\lambda}\underline{\lambda}L_1\kappa h_2^2 - 2\rho)},
\end{aligned} \tag{66}
$$

which completes the proof $\qquad\square$

*Proof of Corollary 2.* Adopting the hypothesis of Theorem 1 (and $L_1 = L_1(f)$) and letting $B$ defined in (12) be $B = \lambda \mathbf{I}$ with $\lambda > 0$, we have

$$\frac{1}{N+1} \sum_{k=0}^{N} E_{\mathcal{U}_k}[|F_\mu(\hat{z}_k)|^2] \le \frac{2\lambda^2 L_1 |z_0 - z^*|^2}{(\lambda^2 L_1 h_2^2 - 2\rho)(N+1)} + \frac{2\lambda L_1 d + 2\lambda L_1^2 \rho(d+3)^3}{(\lambda^2 L_1 h_2^2 - 2\rho)} \mu^2 + \frac{3\lambda\sigma^2}{L_1(\lambda^2 L_1 h_2^2 - 2\rho)}.$$

We want to obtain a guideline on how to choose the parameters $N$ and $\mu$, given a measure of performance $\epsilon$, in order to bound the above inequality by $\epsilon$. Thus, by bounding terms $\frac{2\lambda^2 L_1 \|z_0 - z^*\|^2}{(\lambda^2 L_1 h_2^2 - 2\rho)(N+1)}$ and $\frac{2L_1 d + 2L_1^2 \rho(d+3)^3}{(\lambda^2 L_1 h_2^2 - 2\rho)} \lambda\mu^2$ separately by $\frac{\epsilon^2}{2}$, we obtain the lower bound on the number of iterations $N$ and upper bound on smoothing parameter $\mu$. Thus if

$$\mu \le \left( \frac{(\lambda^2 L_1 h_2^2 - 2\rho)}{4\lambda L_1 d + 4\lambda L_1^2 \rho(d+3)^3} \right)^{\frac{1}{2}} \epsilon \quad \text{and} \quad N \ge \left\lceil \left( \frac{4\lambda^2 L_1 r_0^2}{(\lambda^2 L_1 h_2^2 - 2\rho)} \right) \epsilon^{-2} - 1 \right\rceil,$$

then,

$$\frac{1}{N+1} \sum_{k=0}^{N} E_{\mathcal{U}_k}[|F_\mu(\hat{z}_k)|^2] \le \epsilon^2 + \frac{3\lambda\sigma^2}{(\lambda^2 L_1^2 h_2^2 - 2\rho L_1)},$$

which completes the proof. □

*Proof of Corollary 3.* Considering Lemma 7 (and $L_1 = L_1(f)$) and letting $B$ defined in (12) be $\lambda\mathbf{I}$ where $\lambda > 0$, it can be seen that

$$\frac{1}{N+1} \sum_{k=0}^{N} E_{\mathcal{U}_k}[|F(\hat{z}_k)|^2] \le \frac{2}{N+1} \sum_{k=0}^{N} E_{\mathcal{U}_k}[|F_\mu(\hat{z}_k)|^2] + \frac{\mu^2}{2} \lambda L_1^2 (d+3)^3.$$

Considering Theorem 1 and (20), if

$$\mu \le \min \left\{ \frac{\epsilon}{\sqrt{2}\lambda L_1(d+3)^{\frac{3}{2}}}, \epsilon \sqrt{\left( \frac{16\lambda L_1 d + 16\lambda L_1^2 \rho(d+3)^3}{(\lambda^2 L_1 h_2^2 - 2\rho)} \right)^{-1}} \right\} \quad \text{and} \quad N \ge \left\lceil \left( \frac{8\lambda^2 L_1 r_0^2}{(\lambda^2 L_1 h_2^2 - 2\rho)} \right) \epsilon^{-2} - 1 \right\rceil,$$

then

$$\frac{1}{N+1} \sum_{k=0}^{N} E_{\mathcal{U}_k}[|F(\hat{z}_k)|^2] \le \epsilon^2 + \frac{6\lambda\sigma^2}{\lambda^2 L_1(L_1 h_2^2 - 2\rho)}$$

and thus the assertion follows. □

*Proof of Theorem 2.* The proof of Theorem 2 follows the proof of Theorem 1 from beginning to (63). Using (63) and by taking the expected value of (62) with respect to $u_k$ and then with respect to $\hat{u}_k$, noting that $E_{u_k}[\hat{\xi}_k] = 0$, $E_{u_k}[|\hat{\xi}_k|^2] \le \bar{\lambda} E_{u_k}[\|\hat{\xi}_k\|_*^2] \le \frac{\bar{\lambda}\sigma^2}{t_k}$, $E_{\hat{u}_k}[\xi_k] = 0$, and $E_{\hat{u}_k}[|\xi_k|^2] \le \bar{\lambda} E_{\hat{u}_k}[\|\xi_k\|_*^2] \le \frac{\bar{\lambda}\sigma^2}{t_k}$, we have

$$\left( \frac{h_2^2}{4} - \frac{\rho}{2L_1 \bar{\lambda}\underline{\lambda}\kappa} \right) E_{u_k, \hat{u}_k}[|F_\mu(\hat{z}_k)|^2] \le \frac{1}{2}(r_k^2 - E_{u_k, \hat{u}_k}[r_{k+1}^2]) + \frac{\mu^2 d}{2\bar{\lambda}\kappa} + \frac{\rho}{2\bar{\lambda}\kappa} \mu^2 L_1(d+3)^3$$

$$+ \frac{1}{4L_1^2 \bar{\lambda}\kappa^2 t_k} \sigma^2 + \frac{1}{2L_1^2 \bar{\lambda}\kappa^2 t_k} \sigma^2. \tag{67}$$

Since $u_k$ and $\hat{u}_k$ are independent by assumption, $\xi_k$, $z_k$, and $\hat{z}_k$ are independent of $u_k$, and $E_{u_k}[\hat{\xi}_k] = 0$, the expected value of the last four terms of (62) with respected to $u_k$ are zero.

Next, let $\mathcal{U}_k = [(u_0, \hat{u}_0), (u_1, \hat{u}_1), \cdots, (u_k, \hat{u}_k)]$ for $k \in \{0, \ldots, N\}$. Computing the expected value of (64) with respect to $\mathcal{U}_{k-1}$, letting $\phi_k = E_{\mathcal{U}_{k-1}}[r_k^2]$ and $\phi_0^2 = r_0^2$, we have

$$\left( \frac{h_2^2}{4} - \frac{\rho}{2L_1 \bar{\lambda}\underline{\lambda}\kappa} \right) E_{\mathcal{U}_k}[|F_\mu(\hat{z}_k)|^2] \le \frac{1}{2}(\phi_k^2 - \phi_{k+1}^2) + \frac{\mu^2 d}{2\bar{\lambda}\kappa} + \frac{\rho}{2\bar{\lambda}\kappa} \mu^2 L_1(d+3)^3 + \frac{1}{4L_1^2 \bar{\lambda}\kappa^2 t_k} \sigma^2 + \frac{1}{2L_1^2 \bar{\lambda}\kappa^2 t_k} \sigma^2. \tag{68}$$

Summing (68) from $k = 0$ to $k = N$, and dividing it by $N + 1$, yields

$$\frac{1}{N+1}\sum_{k=0}^{N} E_{\mathcal{U}_k}[|F_\mu(\hat{z}_k)|^2] \leq \frac{2\overline{\lambda}\underline{\lambda}L_1\kappa|z_0 - z^*|^2}{(\overline{\lambda}\underline{\lambda}L_1\kappa h_2^2 - 2\rho)(N+1)} + \frac{2\underline{\lambda}\mu^2 L_1 d}{(\overline{\lambda}\underline{\lambda}L_1\kappa h_2^2 - 2\rho)} + \frac{2\underline{\lambda}\mu^2 L_1^2\rho(d+3)^3}{(\overline{\lambda}\underline{\lambda}L_1\kappa h_2^2 - 2\rho)}$$

$$+ \frac{3\underline{\lambda}\sigma^2}{L_1\kappa(\overline{\lambda}\underline{\lambda}L_1\kappa h_2^2 - 2\rho)}\frac{1}{N+1}\sum_{k=0}^{N}\frac{1}{t_k}, \tag{69}$$

which completes the proof $\qquad\square$

*Proof of Corollary 4.* Considering Lemma 7 (and $L_1 = L_1(f)$) and let $B$ defined in (12) be the identity matrix, it can be seen that

$$\frac{1}{N+1}\sum_{k=0}^{N} E_{\mathcal{U}_k}[|F(\hat{z}_k)|^2] \leq \frac{2}{N+1}\sum_{k=0}^{N} E_{\mathcal{U}_k}[|F_\mu(\hat{z}_k)|^2] + \frac{\mu^2}{2}\lambda L_1^2(d+3)^3.$$

Considering Theorem 2 and (21), if $t_k = t \geq \frac{18\lambda\sigma^2}{L_1(\lambda^2 L_1 h_2^2 - 2\rho)}\epsilon^{-2}$ and

$$\mu \leq \min\left\{\frac{\epsilon}{\sqrt{3}\lambda L_1(d+3)^{\frac{3}{2}}}, \epsilon\sqrt{\left(\frac{24\lambda L_1 d + 24\lambda L_1^2\rho(d+3)^3}{(\lambda^2 L_1 h_2^2 - 2\rho)}\right)^{-1}}\right\} \quad \text{and} \quad N \geq \left\lceil\left(\frac{12\lambda^2 L_1 r_0^2}{(\lambda^2 L_1 h_2^2 - 2\rho)}\right)\epsilon^{-2} - 1\right\rceil,$$

then

$$\frac{1}{N+1}\sum_{k=0}^{N} E_{\mathcal{U}_k}[|F(\hat{z}_k)|^2] \leq \epsilon^2$$

and thus the assertion follows. $\qquad\square$

*Proof of Theorem 3.* In the following we use $L_1 = L_1(f)$ to shorten expressions. Considering $\xi_k = G_\mu(z_k) - F_\mu(z_k)$ and $\hat{\xi}_k = G_\mu(\hat{z}_k) - F_\mu(\hat{z}_k)$, $h_1 = h_2 = h$, $\sqrt{\frac{6\rho}{L_1\underline{\lambda}\overline{\lambda}\kappa}} < h \leq \frac{1}{2L_1\kappa\overline{\lambda}}$ and $\rho \leq \frac{1}{24L_1\kappa^2}$, the following estimate holds:

$$|z_{k+1} - z^*|^2 = |z_k - h\hat{s}_k - z^*|^2$$
$$= |z_k - z^*|^2 + h^2|\hat{s}_k|^2 - 2h\langle\hat{s}_k, z_k - z^*\rangle$$
$$= |z_k - z^*|^2 + h^2|\hat{s}_k|^2 - 2h\langle\hat{s}_k, z_k - \hat{z}_k\rangle - 2h\langle\hat{s}_k, \hat{z}_k - z^*\rangle$$
$$\leq |z_k - z^*|^2 + h^2|\hat{s}_k|^2 - 2h\langle\hat{s}_k, z_k - \hat{z}_k\rangle + 2h\Big(\rho\|\hat{s}_k\|_*^2$$
$$+ \frac{\mu}{2}D_z\kappa L_1(f)(d+3)^{\frac{3}{2}} + D_z\kappa\|\hat{\xi}_k\|_* + \frac{\mu^2}{2}\rho\kappa^2 L_1^2(f)(d+3)^3 + 2\rho\kappa^2\|\hat{\xi}_k\|_*^2\Big)$$
$$= |z_k - z^*|^2 + h^2|\hat{s}_k|^2 - 2h^2\langle\hat{s}_k, s_k\rangle + 2h\rho\|\hat{s}_k\|_*^2$$
$$+ 2h\left(\frac{\mu}{2}D_z\kappa L_1(f)(d+3)^{\frac{3}{2}} + D_z\kappa\|\hat{\xi}_k\|_* + \frac{\mu^2}{2}\rho\kappa^2 L_1^2(f)(d+3)^3 + 2\rho\kappa^2\|\hat{\xi}_k\|_*^2\right)$$
$$\leq |z_k - z^*|^2 + h^2(|\hat{s}_k - s_k|^2 - |s_k|^2) + \frac{4h\rho}{\underline{\lambda}}(|\hat{s}_k - s_k|^2 + |s_k|^2)$$
$$+ 2h\left(\frac{\mu}{2}D_z\kappa L_1(f)(d+3)^{\frac{3}{2}} + D_z\kappa\|\hat{\xi}_k\|_* + \frac{\mu^2}{2}\rho\kappa^2 L_1^2(f)(d+3)^3 + 2\rho\kappa^2\|\hat{\xi}_k\|_*^2\right). \tag{70}$$

The first inequality is obtained using Lemma 2 and the second inequality is obtained by completing squares. Next, we derive an upper bound for the term $\|\hat{s}_k - s_k\|^2$. We have

$$
\begin{aligned}
|\hat{s}_k - s_k|^2 &\le |(\hat{v}_k - v_k) + (\hat{s}_k - \hat{v}_k) + (v_k - s_k)|^2 \\
&\le 2|\hat{v}_k - v_k|^2 + 4|\hat{s}_k - \hat{v}_k|^2 + 4|v_k - s_k|^2 \\
&\le 2\overline{\lambda}\kappa^2 \|F_\mu(\hat{z}_k) - F_\mu(z_k)\|_*^2 + 4\overline{\lambda}\kappa\|G_\mu(z_k) - F_\mu(z_k)\|_*^2 + 4\overline{\lambda}\kappa\|G_\mu(\hat{z}_k) - F_\mu(\hat{z}_k)\|_*^2 \\
&\le 2L_1^2\overline{\lambda}^2\kappa^2 |\hat{z}_k - z_k|^2 + 4\overline{\lambda}\kappa\|\xi_k\|_*^2 + 4\overline{\lambda}\kappa\|\hat{\xi}_k\|_*^2 \\
&= 2h^2\overline{\lambda}^2\kappa^2 L_1^2 |s_k|^2 + 4\overline{\lambda}\kappa\|\xi_k\|_*^2 + 4\overline{\lambda}\kappa\|\hat{\xi}_k\|_*^2.
\end{aligned}
\tag{71}
$$

The third inequality is due to the inequalities $\|s_k - v_k\| \le \|G_\mu(z_k) - F_\mu(z_k)\|$, $\|\hat{s}_k - \hat{v}_k\| \le \|G_\mu(\hat{z}_k) - F_\mu(\hat{z}_k)\|$, and $\|\hat{v}_k - v_k\| \le \|F_\mu(\hat{z}_k) - F_\mu(z_k)\|$, which can be directly obtained from the non-expansiveness of the projection operator. The forth inequality is obtained using the fact that the gradient of the objective function is Lipchitz. Plugging (71) in (70), we have

$$
\begin{aligned}
|z_{k+1} - z^*|^2 &\le |z_k - z^*|^2 + h^2(2h^2\overline{\lambda}^2\kappa^2 L_1^2 |s_k|^2 + 4\overline{\lambda}\kappa\|\xi_k\|_*^2 + 4\overline{\lambda}\kappa\|\hat{\xi}_k\|_*^2 - \|s_k\|^2) \\
&\quad + \frac{4h\rho}{\underline{\lambda}}\left(2h^2\overline{\lambda}^2\kappa^2 L_1^2 |s_k|^2 + 4\overline{\lambda}\kappa\|\xi_k\|_*^2 + 4\overline{\lambda}\kappa\|\hat{\xi}_k\|_*^2 + \|s_k\|^2\right) + 2h\Big(\frac{\mu}{2}D_z\kappa L_1(f)(d+3)^{\frac{3}{2}} \\
&\quad + D_z\kappa\|\hat{\xi}_k\|_* + \frac{\mu^2}{2}\rho\kappa^2 L_1^2(f)(d+3)^3 + 2\rho\kappa^2\|\hat{\xi}_k\|_*^2\Big) \\
&\le |z_k - z^*|^2 + h^2(2L_1^2 h^2\overline{\lambda}^2\kappa^2 - 1)|s_k|^2 + \frac{4h\rho}{\underline{\lambda}}(2h^2 L_1^2\overline{\lambda}^2\kappa^2 + 1)|s_k|^2 + \mu h D_z\kappa L_1(d+3)^{\frac{3}{2}} \\
&\quad + \mu^2 h\rho\kappa^2 L_1^2(d+3)^3 + 2hD_z\kappa\|\hat{\xi}_k\|_* + (20h\rho\kappa^2 + 4\overline{\lambda}\kappa h^2)\|\hat{\xi}_k\|_*^2 + (16h\rho\kappa^2 + 4\overline{\lambda}\kappa h^2)\|\xi_k\|_*^2 \\
&\le |z_k - z^*|^2 - \frac{h^2}{2}|s_k|^2 + \frac{3\rho}{L_1\underline{\lambda}\overline{\lambda}\kappa}|s_k|^2 + \frac{\mu}{2\overline{\lambda}}D_z(d+3)^{\frac{3}{2}} + \frac{\mu^2}{2\overline{\lambda}}\rho\kappa L_1(d+3)^3 \\
&\quad + \frac{D_z}{L_1\overline{\lambda}}\|\hat{\xi}_k\| + \left(\frac{10\rho\kappa}{L_1\overline{\lambda}} + \frac{1}{L_1^2\overline{\lambda}\kappa}\right)\|\hat{\xi}_k\|_*^2 + \left(\frac{8\rho\kappa}{L_1\overline{\lambda}} + \frac{1}{L_1^2\overline{\lambda}\kappa}\right)\|\xi_k\|_*^2.
\end{aligned}
\tag{72}
$$

Letting $r_k^2 = |z_k - z^*|^2$, the inequality

$$
\begin{aligned}
\left(\frac{h^2}{2} - \frac{3\rho}{L_1\underline{\lambda}\overline{\lambda}\kappa}\right)|s_k|^2 &\le r_k^2 - r_{k+1}^2 + \frac{\mu}{2\overline{\lambda}}D_z(d+3)^{\frac{3}{2}} + \frac{\mu^2}{2\overline{\lambda}}\rho\kappa L_1(d+3)^3 + \frac{D_z}{L_1\overline{\lambda}}\|\hat{\xi}_k\| \\
&\quad + \left(\frac{10\rho\kappa}{L_1\overline{\lambda}} + \frac{1}{L_1^2\overline{\lambda}\kappa}\right)\|\hat{\xi}_k\|_*^2 + \left(\frac{8\rho\kappa}{L_1\overline{\lambda}} + \frac{1}{L_1^2\overline{\lambda}\kappa}\right)\|\xi_k\|_*^2
\end{aligned}
\tag{73}
$$

holds. As a next step, we compute the expected value of (73) with respect to $u_k$ and then with respect to $\hat{u}_k$, and we use the fact that $E_{u_k}[\|\hat{\xi}_k\|_*] \le \sigma$, $E_{\hat{u}_k}[\|\xi_k\|_*] \le \sigma$, which follows from the assumptions of Theorem 3 and Jensen's inequality. Let $\mathcal{U}_k = [(u_0, \hat{u}_0), (u_1, \hat{u}_1), \cdots, (u_k, \hat{u}_k)]$ for $k \in \{0, \ldots, N\}$ and $E_{\mathcal{U}_{k-1}}[r_k^2] = \phi_k^2$. Then, Computing expected value of (73) with respect to $\mathcal{U}_{k-1}$, summing from $k = 0$ to $k = N$, and dividing it by $N + 1$, yields

$$
\begin{aligned}
\frac{1}{N+1}\sum_{k=0}^{N} E_{\mathcal{U}_k}[|s_k|^2] &\le \frac{2L_1\underline{\lambda}\overline{\lambda}\kappa|z_0 - z^*|^2}{(L_1 h^2\underline{\lambda}\overline{\lambda}\kappa - 6\rho)(N+1)} + \frac{\mu D_z\underline{\lambda}\kappa L_1(d+3)^{3/2}}{L_1 h^2\underline{\lambda}\overline{\lambda}\kappa - 6\rho} + \frac{\mu^2\rho\underline{\lambda}\kappa^2 L_1^2(d+3)^3}{L_1 h^2\underline{\lambda}\overline{\lambda}\kappa - 6\rho} \\
&\quad + \frac{(36\rho\kappa^2\underline{\lambda} + \frac{4\underline{\lambda}}{L_1})\sigma^2}{L_1 h^2\underline{\lambda}\overline{\lambda}\kappa - 6\rho} + \frac{2D_z\underline{\lambda}\kappa\sigma}{L_1 h^2\underline{\lambda}\overline{\lambda}\kappa - 6\rho},
\end{aligned}
\tag{74}
$$

which completes the proof. $\qquad\square$

*Proof of Corollary 6.* The proof is similar to the proof of Corollary 2. Adopting the hypothesis of Theorem 3 (and $L_1 = L_1(f)$) and letting $B$ defined in (12) be $\lambda \mathbf{I}$ where $\lambda > 0$, we have

$$\frac{1}{N+1}\sum_{k=0}^{N} E_{\mathcal{U}_k}[|s_k|^2] \leq \frac{2\lambda^2 L_1 |z_0 - z^*|^2}{(\lambda^2 L_1 h^2 - 6\rho)(N+1)} + \frac{\mu\lambda D_z L_1 (d+3)^{3/2}}{\lambda^2 L_1 h^2 - 6\rho} + \frac{\mu^2 \lambda \rho L_1^2 (d+3)^3}{\lambda^2 L_1 h^2 - 6\rho}$$
$$+ \frac{(36\rho + \frac{4}{L_1})\lambda\sigma^2}{\lambda^2 L_1 h^2 - 6\rho} + \frac{2D_z \lambda\sigma}{\lambda^2 L_1 h^2 - 6\rho}.$$

We want to obtain a guideline on how to choose the parameters $N$ and $\mu$, given a measure of performance $\epsilon$, in order to bound the above inequality by $\epsilon$. Thus, by bounding terms $\frac{2\lambda^2 L_1 \|z_0 - z^*\|^2}{(\lambda^2 L_1 h^2 - 6\rho)(N+1)}$ and $\frac{\mu\lambda D_z L_1 (d+3)^{3/2}}{\lambda^2 L_1 h^2 - 6\rho} + \frac{\mu^2 \rho\lambda L_1^2 (d+3)^3}{\lambda^2 L_1 h^2 - 6\rho}$ separately by $\frac{\epsilon^2}{2}$, we obtain the lower bound on the number of iterations $N$ and upper bound on smoothing parameter $\mu$. Let $a = \frac{\rho\lambda L_1^2 (d+3)^3}{\lambda^2 L_1 h^2 - 6\rho}$, and $b = \frac{\lambda L_1 D_z (d+3)^{\frac{3}{2}}}{\lambda^2 L_1 h^2 - 6\rho}$. Thus if

$$\mu \leq \frac{-b + \sqrt{b^2 + 2a\epsilon^2}}{2a} \qquad \text{and} \qquad N \geq \left\lceil \left(\frac{4\lambda^2 L_1 r_0^2}{\lambda^2 L_1 h^2 - 6\rho}\right)\epsilon^{-2} - 1 \right\rceil,$$

then,

$$\frac{1}{N+1}\sum_{k=0}^{N} E_{\mathcal{U}_k}[|s_k|^2] \leq \epsilon^2 + \frac{(36\rho + \frac{4}{L_1})\lambda\sigma^2}{\lambda^2 L_1 h^2 - 6\rho} + \frac{2D_z \lambda\sigma}{\lambda^2 L_1 h^2 - 6\rho},$$

which completes the proof. $\qquad\square$

*Proof of Corollary 7.* Let $s_k$ be defined in (28), $p_k$ be defined in (34), $v_k$ be defined in (50), and $B$ defined in (12) be $\lambda \mathbf{I}$ where $\lambda > 0$. Adopting the hypothesis of Theorem 3 and considering the fact that $\|v_k - s_k\| \leq \|F_\mu(z_k) - G_\mu(z_k)\|$, which can be obtained directly from (Ghadimi et al., 2016, Proposition 1), we have

$$|v_k|^2 \leq 2|s_k|^2 + 2|v_k - s_k|^2 \leq 2|s_k|^2 + 2|F_\mu(z_k) - G_\mu(z_k)|^2 \leq 2|s_k|^2 + 2|\xi_k|^2.$$

Hence, taking the expected value with respect to $\mathcal{U}_k$, summing it from $k = 0$ to $k = N$, and dividing it by $N + 1$, yields

$$\frac{1}{N+1}\sum_{k=0}^{N} E_{\mathcal{U}_k}[|v_k|^2] \leq \frac{2}{N+1}\sum_{k=0}^{N} E_{\mathcal{U}_k}[|s_k|^2] + 2\lambda\sigma^2.$$

Similarly, the chain of inequalities

$$|p_k|^2 \leq 2|v_k|^2 + 2|p_k - v_k|^2 \leq 2|v_k|^2 + 2|F_\mu(z_k) - F(z_k)|^2 \leq 2|v_k|^2 + \frac{\mu^2\lambda L_1^2(f)(d+3)}{2}$$

is obtained and the last inequality follows from Lemma 7. Thus

$$\frac{1}{N+1}\sum_{k=0}^{N} E_{\mathcal{U}_k}[|p_k|^2] \leq \frac{2}{N+1}\sum_{k=0}^{N} E_{\mathcal{U}_k}[|v_k|^2] + \frac{\mu^2\lambda L_1^2(f)(d+3)}{2},$$

and

$$\frac{1}{N+1}\sum_{k=0}^{N} E_{\mathcal{U}_k}[|p_k|^2] \leq \frac{4}{N+1}\sum_{k=0}^{N} E_{\mathcal{U}_k}[|s_k|^2] + 4\lambda\sigma^2 + \frac{\mu^2\lambda L_1^2(f)(d+3)}{2}.$$

Let $a \stackrel{\text{def}}{=} \frac{4\rho\lambda L_1^2(f)(d+3)^3}{\lambda^2 L_1(f)h^2 - 6\rho}$, $b \stackrel{\text{def}}{=} \frac{4\lambda L_1(f)D_z(d+3)^{\frac{3}{2}}}{\lambda^2 L_1(f)h^2 - 6\rho}$. Considering Theorem 3 and (32), if

$$\mu \leq \min\left\{\frac{-b + \sqrt{b^2 + a\epsilon^2}}{2a}, \frac{\epsilon}{\sqrt{2}\lambda L_1(f)(d+3)^{\frac{3}{2}}}\right\} \qquad \text{and} \qquad N \geq \left\lceil \left(\frac{16\lambda^2 L_1(f)r_0^2}{\lambda^2 L_1(f)h^2 - 6\rho}\right)\epsilon^{-2} - 1 \right\rceil$$

then

$$\frac{1}{N+1}\sum_{k=0}^{N} E_{\mathcal{U}_k}[|p_k|^2] \le \epsilon^2 + \left(\frac{4(36\rho + \frac{4}{L_1(f)})}{\lambda^2 L_1(f)h^2 - 6\rho} + 4\right)\lambda\sigma^2 + \frac{8D_z\lambda\sigma}{\lambda^2 L_1(f)h^2 - 6\rho}$$

and thus the assertion follows. □

*Proof of Theorem 4.* The proof of Theorem 4 follows the proof of Theorem 3 from beginning to (73). As a next step, we compute the expected value of (73) with respect to $u_k$ and then with respect to $\hat{u}_k$, and we use the fact that $E_{u_k}[\|\hat{\xi}_k\|_*] \le \frac{\sigma}{\sqrt{t_k}}$, $E_{\hat{u}_k}[\|\xi_k\|_*] \le \frac{\sigma}{\sqrt{t_k}}$, which follows from the assumptions of Theorem 3 and Jensen's inequality. Let $\mathcal{U}_k = [(u_0, \hat{u}_0), (u_1, \hat{u}_1), \cdots, (u_k, \hat{u}_k)]$ for $k \in \{0, \ldots, N\}$ and $E_{\mathcal{U}_{k-1}}[r_k^2] = \phi_k^2$. Then, Computing expected value of (73) with respect to $\mathcal{U}_{k-1}$, summing from $k = 0$ to $k = N$, and dividing it by $N + 1$, yields

$$
\begin{aligned}
\frac{1}{N+1}\sum_{k=0}^{N} E_{\mathcal{U}_k}[|s_k|^2] \le & \frac{2L_1\underline{\lambda}\overline{\lambda}\kappa|z_0 - z^*|^2}{(L_1 h^2 \underline{\lambda}\overline{\lambda}\kappa - 6\rho)(N+1)} + \frac{\mu D_z\underline{\lambda}\kappa L_1(d+3)^{3/2}}{L_1 h^2 \underline{\lambda}\overline{\lambda}\kappa - 6\rho} + \frac{\mu^2\rho\underline{\lambda}\kappa^2 L_1^2(d+3)^3}{L_1 h^2 \underline{\lambda}\overline{\lambda}\kappa - 6\rho} \\
& + \frac{(36\rho\kappa^2\underline{\lambda} + \frac{4\underline{\lambda}}{L_1})\sigma^2}{L_1 h^2 \underline{\lambda}\overline{\lambda}\kappa - 6\rho}\frac{1}{N+1}\sum_{k=0}^{N}\frac{1}{t_k} + \frac{2D_z\underline{\lambda}\kappa\sigma}{L_1 h^2 \underline{\lambda}\overline{\lambda}\kappa - 6\rho}\frac{1}{N+1}\sum_{k=0}^{N}\frac{1}{\sqrt{t_k}},
\end{aligned}
\tag{75}
$$

which completes the proof □

*Proof of Corollary 8.* Let $s_k$ be defined in (28), $p_k$ be defined in (34), $v_k$ be defined in (50), and $B$ defined in (12) be $\lambda \mathbf{I}$ where $\lambda > 0$. Adopting the hypothesis of Theorem 4 and considering the proof of Corollary 7, we have

$$\frac{1}{N+1}\sum_{k=0}^{N} E_{\mathcal{U}_k}[|p_k|^2] \le \frac{4}{N+1}\sum_{k=0}^{N} E_{\mathcal{U}_k}[|s_k|^2] + 4\lambda\sigma^2 + \frac{\mu^2\lambda L_1^2(f)(d+3)}{2}.$$

Let $a \stackrel{\text{def}}{=} \frac{4\rho\lambda L_1^2(f)(d+3)^3}{\lambda^2 L_1(f)h^2 - 6\rho}$, $b \stackrel{\text{def}}{=} \frac{4\lambda L_1(f)D_z(d+3)^{\frac{3}{2}}}{\lambda^2 L_1(f)h^2 - 6\rho}$, $c = \frac{(36\rho + \frac{4}{L_1} + 4)\lambda\sigma^2}{\lambda^2 L_1 h^2 - 6\rho}$, and $e = \frac{2D_z\lambda\sigma}{\lambda^2 L_1 h^2 - 6\rho}$. Considering Theorem 3 and (32), if $t_k = t \ge \max\{32c\epsilon^{-2}, 32e^2\epsilon^{-4}\}$

$$\mu \le \min\left\{\frac{-b + \sqrt{b^2 + a\epsilon^2}}{2a}, \frac{\epsilon}{\sqrt{2}\lambda L_1(f)(d+3)^{\frac{3}{2}}}\right\} \quad \text{and} \quad N \ge \left\lceil\left(\frac{32\lambda^2 L_1(f)r_0^2}{\lambda^2 L_1(f)h^2 - 6\rho}\right)\epsilon^{-2} - 1\right\rceil$$

then

$$\frac{1}{N+1}\sum_{k=0}^{N} E_{\mathcal{U}_k}[|p_k|^2] \le \epsilon^2$$

and thus the assertion follows. □

*Proof of Theorem 5.* Considering Assumption 4, $h_2 > 0$, and letting $\xi_k = G_\mu(z_k) - F_\mu(z_k)$ and $\hat{\xi}_k = G_\mu(\hat{z}_k) - F_\mu(\hat{z}_k)$ (and recalling that $E_{u_k}[\hat{\xi}_k] = 0$ and $E_{\hat{u}_k}[\xi_k] = 0$), we have

$$
\begin{aligned}
0 \le & \; h_2\langle F_\mu(\hat{z}_k), \hat{z}_k - z^*\rangle + h_2\frac{\rho}{2}\|F_\mu(\hat{z}_k)\|_*^2 \\
= & \; h_2\langle G_\mu(\hat{z}_k), \hat{z}_k - z^*\rangle - h_2\langle\hat{\xi}_k, \hat{z}_k - z^*\rangle + h_2\frac{\rho}{2}\|F_\mu(\hat{z}_k)\|_*^2 \\
= & \; h_2\langle G_\mu(\hat{z}_k), z_{k+1} - z^*\rangle + h_2\langle G_\mu(\hat{z}_k), \hat{z}_k - z_{k+1}\rangle + h_2\langle G_\mu(\hat{z}_k) - G_\mu(z_k), \hat{z}_k - z_{k+1}\rangle \\
& \; - h_2\langle\hat{\xi}_k, \hat{z}_k - z^*\rangle + h_2\frac{\rho}{2}\|F_\mu(\hat{z}_k)\|_*^2.
\end{aligned}
\tag{76a}
$$

As a first step, we derive a bound for the first three terms in (76a). Considering that $\mathcal{Z} = \mathbf{Z}$, from Algorithm 1 line 8, we know that $z_k - z_{k+1} = h_2 G_\mu(\hat{z}_k)$. Thus it holds that

$$
h_2\langle G_\mu(\hat{z}_k), z_{k+1} - z^*\rangle = \langle z_k - z_{k+1}, z_{k+1} - z^*\rangle
$$
$$
= \frac{1}{2}|z^* - z_k|^2 - \frac{1}{2}|z^* - z_{k+1}|^2 - \frac{1}{2}|z_k - z_{k+1}|^2. \tag{77}
$$

Similarly, form Algorithm 1 line 5, we know that $z_k - \hat{z}_k = h_1 G_\mu(z_k)$, and thus the estimate

$$
h_2\langle G_\mu(z_k), \hat{z}_k - z_{k+1}\rangle = \frac{h_2}{h_1}\langle z_k - \hat{z}_k, \hat{z}_k - z_{k+1}\rangle
$$
$$
= \frac{h_2}{2h_1}(|z_k - z_{k+1}|^2 - |z_k - \hat{z}_k|^2 - |z_{k+1} - \hat{z}_k|^2)
$$
$$
= \frac{h_2}{2h_1}|z_k - z_{k+1}|^2 - \frac{h_2}{2h_1\overline{\lambda}}\|z_k - \hat{z}_k\|^2 - \frac{h_2}{2h_1\overline{\lambda}}\|z_{k+1} - \hat{z}_k\|^2 \tag{78}
$$

is obtained. For the third term in (76a), considering that the gradient of $f_\mu$ is Lipschitz continuous, for any $\alpha > 0$, we have

$$
h_2\langle G_\mu(\hat{z}_k) - G_\mu(z_k), \hat{z}_k - z_{k+1}\rangle \leq h_2|F_\mu(\hat{z}_k) - F_\mu(z_k)||\hat{z}_k - z_{k+1}| + h_2\langle \hat{\xi}_k - \xi_k, \hat{z}_k - z_{k+1}\rangle
$$
$$
\leq h_2 L_1(f_\mu)\kappa\|\hat{z}_k - z_k\|\|\hat{z}_k - z_{k+1}\| + h_2\langle \hat{\xi}_k - \xi_k, \hat{z}_k - z_{k+1}\rangle
$$
$$
\leq \frac{h_2 L_1(f_\mu)\alpha\kappa}{2}\|\hat{z}_k - z_k\|^2 + \frac{h_2 L_1(f_\mu)\kappa}{2\alpha}\|\hat{z}_k - z_{k+1}\|^2 + h_2\langle \hat{\xi}_k - \xi_k, \hat{z}_k - z_{k+1}\rangle. \tag{79}
$$

Substituting (77), (78), and (79) in (76a), letting $r_k = |z_k - z^*|$, and noting that $z_k - z_{k+1} = h_2 G_\mu(\hat{z}_k)$ and $z_k - \hat{z}_k = h_1 G_\mu(z_k)$, we get

$$
0 \leq \frac{1}{2}(r_k^2 - r_{k+1}^2) + \left(\frac{h_2}{2h_1} - \frac{1}{2}\right)|z_k - z_{k+1}|^2 + \left(\frac{h_2 L_1(f_\mu)\alpha\kappa}{2} - \frac{h_2}{2h_1\overline{\lambda}}\right)\|\hat{z}_k - z_k\|^2
$$
$$
+ \left(\frac{h_2 L_1(f_\mu)\kappa}{2\alpha} - \frac{h_2}{2h_1\overline{\lambda}}\right)\|\hat{z}_k - z_{k+1}\|^2 - h_2\langle \hat{\xi}_k, \hat{z}_k - z^*\rangle + h_2\langle \hat{\xi}_k - \xi_k, \hat{z}_k - z_{k+1}\rangle + h_2\frac{\rho}{2}\|F_\mu(\hat{z}_k)\|_*^2
$$
$$
= \frac{1}{2}(r_k^2 - r_{k+1}^2) + h_2^2\left(\frac{h_2}{2h_1} - \frac{1}{2}\right)|G_\mu(\hat{z}_k)|^2 + h_1^2\left(\frac{h_2 L_1(f_\mu)\alpha\kappa}{2} - \frac{h_2}{2h_1\overline{\lambda}}\right)\|G_\mu(z_k)\|^2
$$
$$
+ \left(\frac{h_2 L_1(f_\mu)\kappa}{2\alpha} - \frac{h_2}{2h_1\overline{\lambda}}\right)\|\hat{z}_k - z_{k+1}\|^2 - h_2\langle \hat{\xi}_k, \hat{z}_k - z^*\rangle + h_2\langle \hat{\xi}_k - \xi_k, \hat{z}_k - z_{k+1}\rangle + h_2\frac{\rho}{2}\|F_\mu(\hat{z}_k)\|_*^2.
$$

Rearranging the above terms, we have

$$
h_2^2\left(\frac{1}{2} - \frac{h_2}{2h_1}\right)|G_\mu(\hat{z}_k)|^2 \leq \frac{1}{2}(r_k^2 - r_{k+1}^2) + h_1^2\left(\frac{h_2 L_1(f_\mu)\alpha\kappa}{2} - \frac{h_2}{2h_1\overline{\lambda}}\right)\|G_\mu(z_k)\|^2 + h_2\frac{\rho}{2}\|F_\mu(\hat{z}_k)\|_*^2
$$
$$
+ \left(\frac{h_2 L_1(f_\mu)\kappa}{2\alpha} - \frac{h_2}{2h_1\overline{\lambda}}\right)\|\hat{z}_k - z_{k+1}\|^2 - h_2\langle \hat{\xi}_k, \hat{z}_k - z^*\rangle + h_2\langle \hat{\xi}_k - \xi_k, h_2 G_\mu(\hat{z}_k) - h_1 G(z_k)\rangle. \tag{80}
$$

Choosing $h_1 \leq \frac{1}{L_1(f_\mu)\overline{\lambda}\kappa}$ and $\alpha = 1$ ensures that the second and third right-hand side terms of (80) are non-positive. Also, we need $\frac{1}{2} - \frac{h_2}{2h_1} > 0$ and thus $h_2 < h_1$ needs to be satisfied. Considering $\sqrt{\frac{\rho}{L_1(f_\mu)\overline{\lambda}\underline{\lambda}\kappa}} \leq h_2 \leq \frac{h_1}{2}$, we have

$$
\frac{h_2^2}{4}|G_\mu(\hat{z}_k)|^2 \leq \frac{1}{2}(r_k^2 - r_{k+1}^2) - h_2\langle \hat{\xi}_k, \hat{z}_k - z^*\rangle + h_2\langle \hat{\xi}_k - \xi_k, h_2 G_\mu(\hat{z}_k) - h_1 G(z_k)\rangle + \frac{h_2\rho}{2}\|F_\mu(\hat{z}_k)\|_*^2
$$
$$
\leq \frac{1}{2}(r_k^2 - r_{k+1}^2) + h_2\langle \hat{\xi}_k, z^* - z_k + h_1 G_\mu(z_k)\rangle + \frac{\rho}{4L_1(f_\mu)\overline{\lambda}\underline{\lambda}\kappa}|F_\mu(\hat{z}_k)|^2 + h_2\langle \hat{\xi}_k - \xi_k, h_2 G_\mu(\hat{z}_k) - h_1 G(z_k)\rangle. \tag{81}
$$

The last term of the inequality above can be equivalently written as

$$
h_2\langle \hat{\xi}_k - \xi_k, h_2 G_\mu(\hat{z}_k) - h_1 G(z_k)\rangle = h_2\langle \hat{\xi}_k - \xi_k, h_2(\hat{\xi}_k + F_\mu(\hat{z}_k)) - h_1(\xi_k + F_\mu(z_k))\rangle
$$
$$
= h_2\langle \hat{\xi}_k - \xi_k, h_2\hat{\xi}_k - h_1\xi_k\rangle + h_2\langle \hat{\xi}_k - \xi_k, h_2 F_\mu(\hat{z}_k) - h_1 F_\mu(z_k)\rangle. \quad (82)
$$

Moreover, for $h_2\langle \hat{\xi}_k - \xi_k, h_2\hat{\xi}_k - h_1\xi_k\rangle$ the equality

$$
h_2\langle \hat{\xi}_k - \xi_k, h_2\hat{\xi}_k - h_1\xi_k\rangle = h_2^2|\hat{\xi}_k|^2 + h_2 h_1|\xi_k|^2 + h_2\langle \xi_k, -h_2\hat{\xi}_k\rangle + h_2\langle \hat{\xi}_k, -h_1\xi_k\rangle \quad (83)
$$

holds. Substituting (82) and (83) in (81), we have

$$
\frac{h_2^2}{4}|G_\mu(\hat{z}_k)|^2 - \frac{\rho}{4L_1(f_\mu)\overline{\lambda}\underline{\lambda}\kappa}|F_\mu(\hat{z}_k)|^2 \le \frac{1}{2}(r_k^2 - r_{k+1}^2) + h_2^2|\hat{\xi}_k|^2 + h_2 h_1|\xi_k|^2
$$
$$
+ h_2\langle \hat{\xi}_k, z^* - z_k + h_1 G_\mu(z_k)\rangle + h_2\langle \hat{\xi}_k - \xi_k, h_2 F_\mu(\hat{z}_k) - h_1 F_\mu(z_k)\rangle + h_2\langle \xi_k, -h_2\hat{\xi}_k\rangle + h_2\langle \hat{\xi}_k, -h_1\xi_k\rangle. \quad (84)
$$

From Jensen's inequality, we know that $E_{u_k}[|G_\mu(\hat{z}_k)|]^2 \le E_{u_k}[|G_\mu(\hat{z}_k)|^2]$. Also, it can be concluded that $E_{u_k}[|G_\mu(\hat{z}_k)|] \ge |E_{u_k}[G_\mu(\hat{z}_k)]| = |F_\mu(\hat{z}_k)|$, and thus

$$
E_{u_k}[|G_\mu(\hat{z}_k)|^2] \ge |F_\mu(\hat{z}_k)|^2. \quad (85)
$$

Using this inequality and the taking the expected value of (81) with respect to $u_k$ and then with respect to $\hat{u}_k$, noting that $E_{u_k}[\hat{\xi}_k] = 0$, $E_{u_k}[|\hat{\xi}_k|^2] \le \overline{\lambda}E_{u_k}[\|\hat{\xi}_k\|_*^2] \le \overline{\lambda}\sigma^2$, $E_{\hat{u}_k}[\xi_k] = 0$, and $E_{\hat{u}_k}[|\xi_k|^2] \le \overline{\lambda}E_{\hat{u}_k}[\|\xi_k\|_*^2] \le \overline{\lambda}\sigma^2$, we have

$$
\left(\frac{h_2^2}{4} - \frac{\rho}{4L_1(f_\mu)\overline{\lambda}\underline{\lambda}\kappa}\right)E_{u_k, \hat{u}_k}[|F_\mu(\hat{z}_k)|^2] \le \frac{1}{2}(r_k^2 - E_{u_k, \hat{u}_k}[r_{k+1}^2]) + h_2^2\overline{\lambda}\sigma^2 + \frac{h_2\overline{\lambda}}{L_1(f_\mu)\overline{\lambda}\kappa}\sigma^2. \quad (86)
$$

Since $u_k$ and $\hat{u}_k$ are independent by assumption, $\xi_k$, $z_k$, and $\hat{z}_k$ are independent of $u_k$, and $E_{u_k}[\hat{\xi}_k] = 0$, the expected value of the last four terms of (84) with respected to $u_k$ are zero.

Next, let $\mathcal{U}_k = [(u_0, \hat{u}_0), (u_1, \hat{u}_1), \cdots, (u_k, \hat{u}_k)]$ for $k \in \{0, \ldots, N\}$. Computing the expected value of (86) with respect to $\mathcal{U}_{k-1}$, letting $\phi_k = E_{\mathcal{U}_{k-1}}[r_k^2]$ and $\phi_0^2 = r_0^2$, we have

$$
\left(h_2^2 - \frac{\rho}{L_1(f_\mu)\overline{\lambda}\underline{\lambda}\kappa}\right)E_{\mathcal{U}_k}[|F_\mu(\hat{z}_k)|^2] \le 2(\phi_k^2 - \phi_{k+1}^2) + \frac{3}{L_1^2(f_\mu)\overline{\lambda}\kappa^2}\sigma^2. \quad (87)
$$

Summing (87) from $k = 0$ to $k = N$, and dividing it by $N + 1$, yields

$$
\frac{1}{N+1}\sum_{k=0}^N E_{\mathcal{U}_k}[|F_\mu(\hat{z}_k)|^2] \le \frac{2L_1(f_\mu)\overline{\lambda}\underline{\lambda}\kappa|z_0 - z^*|^2}{(L_1(f_\mu)\overline{\lambda}\underline{\lambda}\kappa h_2^2 - \rho)(N+1)} + \frac{3\underline{\lambda}}{L_1(f_\mu)\kappa(L_1(f_\mu)\overline{\lambda}\underline{\lambda}\kappa h_2^2 - \rho)}\sigma^2,
$$

which completes the proof. $\qquad \square$

*Proof of Corollary 10.* We adopt the hypothesis of Theorem 5 and use the definition $r_0 = \|z_0 - z^*\|$. Let $B$ defined in (12) be the identity matrix. From Lemma 3, we know that $L_1(f_\mu) = \frac{d^{1/2}}{\mu}L_0(f)$.

Hence, if

$$
N \ge \left\lceil \left(\frac{2r_0^2 L_1(f_\mu)}{(L_1(f_\mu)h_2^2 - \rho)}\right)\epsilon^{-2} - 1\right\rceil, \quad \text{then} \quad \frac{1}{N+1}\sum_{k=0}^N E_{\mathcal{U}_k}[\|F_\mu(\hat{z}_k)\|^2] \le \epsilon^2 + \frac{3}{L_1(f_\mu)(L_1(f_\mu)h_2^2 - \rho)}\sigma^2,
$$

where $\epsilon$ is a positive scalar. $\qquad \square$

*Proof of Corollary 11.* Adopting the hypothesis of Theorem 5 and letting $B$ defined in (12) be the identity matrix, we have

$$\frac{1}{N+1}\sum_{k=0}^{N}E_{\mathcal{U}_k}[\|F_\mu(\hat{z}_k)\|^2] \leq \frac{2r_0^2 L_1(f_\mu)}{(L_1(f_\mu)h_2^2-\rho)(N+1)} + \frac{3}{L_1(f_\mu)(L_1(f_\mu)h_2^2-\rho)}\sigma^2.$$

Thus, for $N \geq \left\lceil \frac{8r_0^2 L_1(f_\mu)}{(L_1(f_\mu)h_2^2-\rho)}\epsilon^{-2} - 1 \right\rceil$, there exists a point $\bar{z}$ in the sequence generated such that

$$E_{\mathcal{U}_k}[\|F_\mu(\bar{z})\|^2] \leq \frac{\epsilon^2}{4} + \frac{3}{L_1(f_\mu)(L_1(f_\mu)h_2^2-\rho))}\sigma^2,$$

which implies that

$$E_{\mathcal{U}_k}[\|F_\mu(\bar{z})\|] \leq \frac{\epsilon}{2} + \sqrt{\frac{3}{L_1(f_\mu)(L_1(f_\mu)h_2^2-\rho))}}\sigma. \tag{88}$$

From Lemma 4, we have

$$\nabla f_\mu(\bar{z}) \in \partial_\delta f(\bar{z}) + \mathbb{B}_\gamma(0).$$

This implies that

$$\text{dist}(0, \partial_\delta f(\bar{z})) \leq \|F_\mu(\bar{z})\| + \gamma, \tag{89}$$

where $\text{dist}(0, A) = \min_{a \in A} \|a\|$. Calculating the expected value of (89) and substituting (88) into the expected value, yields

$$E_{\mathcal{U}_k}[\text{dist}(0, \partial_\delta f(\bar{z}))] \leq \frac{\epsilon}{2} + \gamma + \sqrt{\frac{3}{L_1(f_\mu)(L_1(f_\mu)h_2^2-\rho))}}\sigma.$$

Let $\gamma \leq \frac{\epsilon}{2}$. Then , for $\mu \leq \frac{\delta}{\sqrt{d\pi e}}\left(\frac{\epsilon}{8L_0(f)}\right)^{1/d}$,

$$E_{\mathcal{U}_k}[\text{dist}(0, \partial_\delta f(\bar{z}))] \leq \bar{\epsilon}, \qquad \bar{\epsilon} = \epsilon + \sqrt{\frac{3}{L_1(f_\mu)(L_1(f_\mu)h_2^2-\rho))}}\sigma$$

i.e., $\bar{z}$ is a $(\delta, \bar{\epsilon})$-Goldstein stationary point of $f$. $\qquad\square$

*Proof of Theorem 6.* The proof of Theorem 6 follows the proof of Theorem 5 from beginning to (85). Using (85) and by taking the expected value of (84) with respect to $u_k$ and then with respect to $\hat{u}_k$, noting that $E_{u_k}[\hat{\xi}_k] = 0$, $E_{u_k}[|\hat{\xi}_k|^2] \leq \bar{\lambda}E_{u_k}[\|\hat{\xi}_k\|_*^2] \leq \frac{\bar{\lambda}\sigma^2}{t_k}$, $E_{\hat{u}_k}[\xi_k] = 0$, and $E_{\hat{u}_k}[|\xi_k|^2] \leq \bar{\lambda}E_{\hat{u}_k}[\|\xi_k\|_*^2] \leq \frac{\bar{\lambda}\sigma^2}{t_k}$, we have

$$\left(\frac{h_2^2}{4} - \frac{\rho}{4L_1(f_\mu)\bar{\lambda}\underline{\lambda}\kappa}\right)E_{u_k,\hat{u}_k}[|F_\mu(\hat{z}_k)|^2] \leq \frac{1}{2}(r_k^2 - E_{u_k,\hat{u}_k}[r_{k+1}^2]) + h_2^2\bar{\lambda}\sigma^2 + \frac{h_2\bar{\lambda}}{L_1(f_\mu)\bar{\lambda}\kappa t_k}\sigma^2. \tag{90}$$

Since $u_k$ and $\hat{u}_k$ are independent by assumption, $\xi_k$, $z_k$, and $\hat{z}_k$ are independent of $u_k$, and $E_{u_k}[\hat{\xi}_k] = 0$, the expected value of the last four terms of (84) with respected to $u_k$ are zero.

Next, let $\mathcal{U}_k = [(u_0, \hat{u}_0), (u_1, \hat{u}_1), \cdots, (u_k, \hat{u}_k)]$ for $k \in \{0, \ldots, N\}$. Computing the expected value of (90) with respect to $\mathcal{U}_{k-1}$, letting $\phi_k = E_{\mathcal{U}_{k-1}}[r_k^2]$ and $\phi_0^2 = r_0^2$, we have

$$\left(h_2^2 - \frac{\rho}{L_1(f_\mu)\bar{\lambda}\underline{\lambda}\kappa}\right)E_{\mathcal{U}_k}[|F_\mu(\hat{z}_k)|^2] \leq 2(\phi_k^2 - \phi_{k+1}^2) + \frac{3}{L_1^2(f_\mu)\bar{\lambda}\kappa^2 t_k}\sigma^2. \tag{91}$$

Summing (91) from $k = 0$ to $k = N$, and dividing it by $N + 1$, yields

$$\frac{1}{N+1}\sum_{k=0}^{N}E_{\mathcal{U}_k}[|F_\mu(\hat{z}_k)|^2] \leq \frac{2L_1(f_\mu)\bar{\lambda}\underline{\lambda}\kappa|z_0 - z^*|^2}{(L_1(f_\mu)\bar{\lambda}\underline{\lambda}\kappa h_2^2-\rho)(N+1)} + \frac{3\underline{\lambda}\sigma^2}{L_1(f_\mu)\kappa(L_1(f_\mu)\bar{\lambda}\underline{\lambda}\kappa h_2^2-\rho)}\frac{1}{N+1}\sum_{k=0}^{N}\frac{1}{t_k},$$

which completes the proof. $\qquad\square$

*Proof of Corollary 12.* Adopting the hypothesis of Theorem 6 and letting $B$ defined in (12) be the identity matrix, we have

$$\frac{1}{N+1}\sum_{k=0}^{N} E_{\mathcal{U}_k}[\|F_\mu(\hat{z}_k)\|^2] \leq \frac{2r_0^2 L_1(f_\mu)}{(L_1(f_\mu)h_2^2 - \rho)(N+1)} + \frac{3\sigma^2}{L_1(f_\mu)(L_1(f_\mu)h_2^2 - \rho)}\frac{1}{N+1}\sum_{k=0}^{N}\frac{1}{t_k}.$$

Thus, for $N \geq \left\lceil \frac{16r_0^2 L_1(f_\mu)}{(L_1(f_\mu)h_2^2 - \rho)}\epsilon^{-2} - 1 \right\rceil$ and $t_k = t \geq \lceil \frac{24\sigma^2}{L_1(f_\mu)(L_1(f_\mu)h_2^2 - \rho)}\epsilon^{-2}\rceil$, there exists a point $\bar{z}$ in the sequence generated such that

$$E_{\mathcal{U}_k}[\|F_\mu(\bar{z})\|^2] \leq \frac{\epsilon^2}{4},$$

which implies that

$$E_{\mathcal{U}_k}[\|F_\mu(\bar{z})\|] \leq \frac{\epsilon}{2}. \tag{92}$$

From Lemma 4, we have

$$\nabla f_\mu(\bar{z}) \in \partial_\delta f(\bar{z}) + \mathbb{B}_\gamma(0).$$

This implies that

$$\text{dist}(0, \partial_\delta f(\bar{z})) \leq \|F_\mu(\bar{z})\| + \gamma, \tag{93}$$

where $\text{dist}(0, A) = \min_{a \in A}\|a\|$. Calculating the expected value of (93) and substituting (92) into the expected value, yields

$$E_{\mathcal{U}_k}[\text{dist}(0, \partial_\delta f(\bar{z}))] \leq \frac{\epsilon}{2} + \gamma.$$

Let $\gamma \leq \frac{\epsilon}{2}$. Then , for $\mu \leq \frac{\delta}{\sqrt{d\pi e}}(\frac{\epsilon}{8L_0(f)})^{1/d}$,

$$E_{\mathcal{U}_k}[\text{dist}(0, \partial_\delta f(\bar{z}))] \leq \epsilon$$

i.e., $\bar{z}$ is a $(\delta, \epsilon)$-Goldstein stationary point of $f$. $\qquad\square$

## C   Choice of $B$

In this section we provide a discussion on the selection of the matrix $B$ defined in (12) for the different settings in Sections 3.1, 3.2, and 3.3. In particular, we discuss how the smallest and the largest eigenvalues of $B$ impact the results in Section 3 and we discuss how $B$ can be used as a design parameter.

### C.1   Unconstrained Differentiable Settings

To analyse the choice of minimum and maximum eigenvalues of $B$ in the unconstrained case, consider Theorem 2 with $t_k = t$ (the analysis holds for Theorem 1 with $t = 1$). To simplify the notation, let $L_1(f) = L_1$ and let $\nu_1(\cdot)$ denote the right-hand side of (21), i.e., we define

$$\nu_1(\underline{\lambda}, \overline{\lambda}, h_2) = \frac{\underline{\lambda}\overline{\lambda}L_1\kappa a}{\underline{\lambda}\overline{\lambda}L_1\kappa h_2^2 - 2\rho} + \frac{\underline{\lambda}L_1 b + \underline{\lambda}L_1^2 c}{\underline{\lambda}\overline{\lambda}L_1\kappa h_2^2 - 2\rho} + \frac{\underline{\lambda}e}{L_1\kappa(\underline{\lambda}\overline{\lambda}L_1\kappa h_2^2 - 2\rho)}, \tag{94}$$

where $a = \frac{2|z_0 - z^*|^2}{N+1}$, $b = 2\mu^2 d$, $c = 2\mu^2\rho(d+3)^3$, and $e = \frac{3\sigma^2}{t}$. We are interested in minimising $\nu_1(\cdot)$ with respect to all of its variables subject to $0 < \underline{\lambda} \leq \overline{\lambda}$ and $h_2 \in \left(\sqrt{\frac{\rho}{L_1\overline{\lambda}\underline{\lambda}\kappa}}, \frac{h_1}{2}\right]$ where $h_1 \leq \frac{1}{L_1\overline{\lambda}\kappa}$. Moreover, considering (7), we know $L_1$ is dependent on $B$. Thus converting the norms in (7) to $|\cdot|$, we get $|\nabla f(z_2) - \nabla f(z_1)| \leq \overline{L}|z_2 - z_1|$ where $L_1 = \frac{\overline{L}}{\underline{\lambda}}$. Rearranging $\nu_1(\cdot)$ we get

$$\nu_1(\underline{\lambda}, \overline{\lambda}, h_2) = \frac{\underline{\lambda}\overline{L}\kappa a}{\underline{\lambda}\overline{L}\kappa h_2^2 - 2\rho} + \frac{\overline{L}\kappa^{-1}b + \overline{L}^2(\overline{\lambda}\kappa)^{-1}c + \underline{\lambda}^2\overline{L}^{-1}e}{\underline{\lambda}\overline{L}\kappa h_2^2 - 2\rho} = \frac{\overline{\lambda}\bar{a} + \bar{b}\kappa^{-1} + \bar{c}(\overline{\lambda}\kappa)^{-1} + \underline{\lambda}^2\bar{e}}{\overline{L}\overline{\lambda}h_2^2 - 2\rho}, \tag{95}$$

where $\bar{a} = \bar{L}a$, $\bar{b} = \bar{L}b$, $\bar{c} = \bar{L}^2 c$, and $\bar{e} = \bar{L}^{-1}e$. Now, to minimise $\nu_1(\cdot)$ with respect to $h_2$, it can be seen that $h_2$ only appears in denominator and $\frac{\partial \nu_1}{\partial h_2}(\underline{\lambda}, \overline{\lambda}, h_2) < 0$. Thus the optimal $h_2$ is the maximum feasible value or $h_2^* = \frac{1}{2L_1\overline{\lambda}\kappa} = \frac{1}{2\bar{L}\kappa}$. Substituting $h_2^*$ in $\nu_1(\cdot)$ we get

$$\nu_1(\underline{\lambda}, \overline{\lambda}, h_2^*) = \frac{\overline{\lambda}\bar{a} + \bar{b}\kappa^{-1} + \bar{c}(\overline{\lambda}\kappa)^{-1} + \underline{\lambda}^2\bar{e}}{\frac{\underline{\lambda}^2}{4\bar{L}\overline{\lambda}} - 2\rho} = 4\bar{L}\frac{\bar{a}\underline{\lambda}\kappa^2 + \bar{b} + \bar{c}(\underline{\lambda}\kappa)^{-1} + \bar{e}\underline{\lambda}^2\kappa}{\underline{\lambda} - 8\rho\bar{L}\kappa}. \tag{96}$$

Minimising $\nu_1(\cdot)$ with respect to $\underline{\lambda}$ and $\overline{\lambda}$ leads to the problem of finding the roots of two polynomials of degree 4 and above. According to the Abel-Ruffini theorem, no closed form solution to characterise the roots of the polynomials exist.

We thus proceed by numerically evaluating $\nu_1(\underline{\lambda}, \overline{\lambda}, h_2^*)$ for 500 different randomly selected values of parameters $\bar{L}, \rho, |z_0 - z^*|^2, N, \mu, d, \sigma$ and the numerical evaluation indicates that $\kappa^* = 1$ provides the best bound. In particular, we can conclude that, based on numerical experiments, matrices $B \succ 0$ with condition number $\kappa = 1$ and thus

$$\nu_1(\underline{\lambda}, \underline{\lambda}, h_2^*) = 4\bar{L}\frac{\bar{a}\underline{\lambda} + \bar{b} + \bar{c}(\underline{\lambda})^{-1} + \bar{e}\underline{\lambda}^2}{\underline{\lambda} - 8\rho\bar{L}}$$

provide the best bound and the selection of $\underline{\lambda}$ depends on the parameters of a specific optimisation problem.

As a next step, we focus on the analysis of the selection of $B$ under the assumption that $\rho = 0$. Thus, (96) simplifies to

$$\nu_1(\underline{\lambda}, \overline{\lambda}, h_2^*) = \frac{\overline{\lambda}\bar{a} + \bar{b}\kappa^{-1} + \underline{\lambda}^2\bar{e}}{\frac{\underline{\lambda}^2}{4\bar{L}\overline{\lambda}}} = 4\bar{L}(\bar{a}\kappa^2 + \frac{\bar{b}}{\underline{\lambda}} + \bar{e}\overline{\lambda}). \tag{97}$$

Due to the relationship $\overline{\lambda} = \kappa\underline{\lambda}$, we can remove $\overline{\lambda}$ in (97) and minimise

$$\nu_1(\underline{\lambda}, \kappa\underline{\lambda}, h_2^*) = 4\bar{L}(\bar{a}\kappa^2 + \frac{\bar{b}}{\underline{\lambda}} + \bar{e}\underline{\lambda}\kappa)$$

subject to $\kappa \geq 1$ and $\underline{\lambda} > 0$. Taking the derivative of $\nu_1(\cdot)$ with respect to $\underline{\lambda}$ and equalling it to zero, we get $\underline{\lambda}^* = \sqrt{\frac{\bar{b}}{\kappa\bar{e}}}$ and thus we have

$$\nu_1(\underline{\lambda}^*, \kappa\underline{\lambda}^*, h_2^*) = 4\bar{L}(\bar{a}\kappa^2 + 2\sqrt{\bar{b}\bar{e}\kappa}).$$

Finally, from $\kappa \geq 1$ and $\frac{\partial \nu_1}{\partial \kappa}(\underline{\lambda}^*, \kappa\underline{\lambda}^*, h_2^*) > 0$ for $\kappa > 0$, we can conclude that $\kappa^* = 1$. Summarising the above discussion, in the unconstrained case, $\underline{\lambda}^* = \overline{\lambda}^* = \sqrt{\frac{\bar{b}}{\bar{e}}}$

## C.2   Constrained Differentiable Settings

To analyse the choice of minimum and maximum eigenvalues of $B$ in the constrained case we proceed as in the unconstrained case in Appendix C.1 and we consider Theorem 4 with $t_k = t$ (the analysis holds for Theorem 3 with $t = 1$). As in the unconstrained setting, we use the notation $L_1(f) = L_1$ and and define

$$\nu_2(\underline{\lambda}, \overline{\lambda}, h) = \frac{\underline{\lambda}\overline{\lambda}L_1\kappa a + \underline{\lambda}\kappa L_1 b + \underline{\lambda}\kappa^2 L_1^2 c + (36\rho\kappa^2\underline{\lambda} + \frac{4\underline{\lambda}}{L_1})e + \underline{\lambda}\kappa p}{\underline{\lambda}\overline{\lambda}L_1\kappa h^2 - 6\rho} \tag{98}$$

to denote the right-hand side of (35) with $a = \frac{2|z_0 - z^*|^2}{N+1}$, $b = \mu D_z(d+3)^{3/2}$, $c = \mu^2\rho(d+3)^3$, $e = \frac{\sigma^2}{t}$, and $p = \frac{2D_z\sigma}{\sqrt{t}}$. We are interested in minimising $\nu_2(\cdot)$ with respect to all of its variables subject to $0 < \underline{\lambda} \leq \overline{\lambda}$ and $h \in \left(\sqrt{\frac{\rho}{L_1\overline{\lambda}\underline{\lambda}\kappa}}, \frac{1}{2L_1\overline{\lambda}\kappa}\right]$. Moreover, considering (7), we know $L_1$ is dependent on $B$. Thus converting the norms in (7) to $|\cdot|$, we get $|\nabla f(z_2) - \nabla f(z_1)| \leq \bar{L}|z_2 - z_1|$ where $L_1 = \frac{\bar{L}}{\underline{\lambda}}$. Rearranging the individual terms

in $\nu_2(\cdot)$ we get

$$\nu_2(\underline{\lambda}, \overline{\lambda}, h) = \frac{\overline{\lambda}\bar{a} + \bar{b} + \bar{c}\underline{\lambda}^{-1} + p\underline{\lambda}\kappa + (36\rho\kappa^2\underline{\lambda} + \frac{4\underline{\lambda}\overline{\lambda}}{\bar{L}})e}{\bar{L}\overline{\lambda}h^2 - 6\rho}, \tag{99}$$

where $\bar{a} = \bar{L}a$, $\bar{b} = \bar{L}b$, and $\bar{c} = \bar{L}^2 c$. Now, to minimise $\nu_2(\cdot)$ with respect to $h$, it can be seen that $h$ only appears in denominator and $\frac{\partial \nu_2}{\partial h}(\underline{\lambda}, \overline{\lambda}, h) < 0$. Thus the optimal $h$ is the maximum feasible value or $h^* = \frac{1}{2L_1\overline{\lambda}\kappa} = \frac{1}{2\bar{L}\kappa}$. Substituting $h^*$ in $\nu_2$ we get

$$\nu_2(\underline{\lambda}, \overline{\lambda}, h^*) = 4\bar{L}\frac{\overline{\lambda}\kappa\bar{a} + \bar{b}\kappa + \bar{c}\kappa\underline{\lambda}^{-1} + p\underline{\lambda}\kappa^2 + (36\rho\kappa^3\underline{\lambda} + \frac{4\overline{\lambda}^2}{\bar{L}})e}{\underline{\lambda} - 24\rho\bar{L}\kappa},$$

Minimising $\nu_2(\cdot)$ with respect to $\underline{\lambda}$ and $\overline{\lambda}$ leads to the problem of finding the roots of two polynomials of degree 5 and above. As before, we thus proceed by numerically evaluating $\nu_2(\underline{\lambda}, \overline{\lambda}, h^*)$ for 500 different randomly picked values of parameters $\bar{L}, \rho, |z_0 - z^*|^2, N, \mu, d, \sigma$ and the numerical evaluation again indicates that $\kappa^* = 1$ provides the best bound. In particular, we can conclude that, based on numerical experiments, matrices $B \succ 0$ with condition number $\kappa = 1$ and thus

$$\nu_2(\underline{\lambda}, \underline{\lambda}, h^*) = 4\bar{L}\frac{\overline{\lambda}\bar{a} + \bar{b} + \bar{c}\underline{\lambda}^{-1} + p\underline{\lambda} + (36\rho\underline{\lambda} + \frac{4\underline{\lambda}^2}{\bar{L}})e}{\underline{\lambda} - 24\rho\bar{L}}$$

provide the best bound and the selection of $\underline{\lambda}$ depends on the parameters of a specific optimisation problem. We continue with the analysis for the case that $\rho = 0$. Thus, it holds that

$$\nu_2(\underline{\lambda}, \overline{\lambda}, h^*) = 4\bar{L}\frac{\overline{\lambda}\kappa\bar{a} + \bar{b}\kappa + p\underline{\lambda}\kappa^2 + (\frac{4\overline{\lambda}^2}{\bar{L}})e}{\underline{\lambda}} = 4\bar{L}\left(\bar{a}\kappa^2 + \frac{\bar{b}\kappa}{\underline{\lambda}} + p\kappa^2 + \bar{e}\frac{\overline{\lambda}^2}{\underline{\lambda}}\right),$$

where $\bar{e} = \frac{4}{\bar{L}}e$. We use again the relationship $\overline{\lambda} = \kappa\underline{\lambda}$ to replace $\overline{\lambda}$ and consider

$$\nu_2(\underline{\lambda}, \kappa\underline{\lambda}, h^*) = 4\bar{L}(\bar{a}\kappa^2 + \frac{\bar{b}\kappa}{\underline{\lambda}} + p\kappa^2 + \bar{e}\kappa^2\underline{\lambda})$$

subject to $\kappa \geq 1$ and $\underline{\lambda} > 0$. Continuing with the same steps as in the unconstrained setting, we get $\underline{\lambda}^* = \sqrt{\frac{\bar{b}}{\kappa\bar{e}}}$ and

$$\nu_2(\underline{\lambda}^*, \kappa\underline{\lambda}^*, h^*) = 4\bar{L}(p\kappa^2 + \bar{a}\kappa^2 + 2\sqrt{\bar{b}\bar{e}\kappa^3}),$$

and from $\kappa \geq 1$ and $\frac{\partial \nu_2}{\partial \kappa}(\underline{\lambda}^*, \kappa\underline{\lambda}^*, h^*) > 0$ for positive $\kappa$ it follows that $\kappa^* = 1$ and $\underline{\lambda}^* = \overline{\lambda}^* = \sqrt{\frac{\bar{b}}{\bar{e}}}$.

### C.3 Unconstrained Non-Differentiable Settings

To analyse the choice of minimum and maximum eigenvalues of $B$ in the non-differentiable case, consider Theorem 6 with $t_k = t$ (the analysis holds for Theorem 5 with $t = 1$). Let $L_1(f) = L_1$ and let $\nu_3(\cdot)$ denote right-hand side of (38), i.e., we define

$$\nu_3(\underline{\lambda}, \overline{\lambda}, h_2) = \frac{\underline{\lambda}\overline{\lambda}L_1\kappa a + \underline{\lambda}(L_1\kappa)^{-1}b}{\underline{\lambda}\overline{\lambda}L_1\kappa h_2^2 - \rho} \tag{100}$$

where $a = \frac{2|z_0 - z^*|^2}{N+1}$ and $b = \frac{3\sigma^2}{t}$. We are interested in minimising $\nu_3(\cdot)$ with respect to all of its variables subject to $0 < \underline{\lambda} \leq \overline{\lambda}$ and $h_2 \in \left(\sqrt{\frac{\rho}{L_1\underline{\lambda}\kappa}}, \frac{h_1}{2}\right]$ where $h_1 \leq \frac{1}{L_1\overline{\lambda}\kappa}$. Moreover, considering (7), we know $L_1$ is dependent on $B$. Thus converting the norms in (7) to $|\cdot|$, we get $|\nabla f(z_2) - \nabla f(z_1)| \leq \bar{L}|z_2 - z_1|$ where $L_1 = \frac{\bar{L}}{\underline{\lambda}}$. Rearranging $\nu_3(\cdot)$ we get

$$\nu_3(\underline{\lambda}, \overline{\lambda}, h_2) = \frac{\overline{\lambda}\bar{a} + \underline{\lambda}^2\bar{b}}{\bar{L}\overline{\lambda}h_2^2 - \rho}, \tag{101}$$

where $\bar{a} = \bar{L}a$ and $\bar{b} = \bar{L}^{-1}b$. Now to minimise $\nu_3(\cdot)$ with respect to $h_2$, it can be seen that $h_2$ only appears in denominator and $\frac{\partial \nu_3}{\partial h_2}(\underline{\lambda}, \overline{\lambda}, h_2) < 0$. Thus the optimal $h_2$ is the maximum feasible value or $h_2^* = \frac{1}{2L_1 \overline{\lambda} \kappa} = \frac{1}{2\bar{L}\kappa}$. Substituting $h_2^*$ in $\nu_3(\cdot)$ we get

$$\nu_3(\underline{\lambda}, \overline{\lambda}, h_2^*) = \frac{\overline{\lambda}\bar{a} + \underline{\lambda}^2 \bar{b}}{\frac{\underline{\lambda}^2}{4\bar{L}\overline{\lambda}} - \rho} = 4\bar{L}\frac{\bar{a}\underline{\lambda}\kappa^2 + \bar{b}\underline{\lambda}^2 \kappa}{\underline{\lambda} - 4\rho\bar{L}\kappa}.$$

As before, we analyse $\nu_3(\underline{\lambda}, \overline{\lambda}, h_2^*)$ numerically for 500 different randomly picked values of parameters $\bar{L}, \rho, |z_0 - z^*|^2, N, \sigma$, indicating that $\kappa^* = 1$ is optimal. In particular, we conclude that, based on numerical experiments, matrices $B \succ 0$ with condition number $\kappa = 1$ are optimal and thus

$$\nu_3(\underline{\lambda}, \underline{\lambda}, h_2^*) = 4\bar{L}\frac{\bar{a}\underline{\lambda} + \bar{b}\underline{\lambda}^2}{\underline{\lambda} - 4\rho\bar{L}}$$

provide the best bound. The optimal selection of $\underline{\lambda}$ again depends on the parameters of a specific optimisation problem.

For the case $\rho = 0$, we have

$$\nu_3(\underline{\lambda}, \overline{\lambda}, h_2^*) = \frac{\overline{\lambda}\bar{a} + \underline{\lambda}^2 \bar{b}}{\frac{\underline{\lambda}^2}{4\bar{L}\overline{\lambda}}} = 4\bar{L}(\bar{a}\kappa^2 + \bar{b}\overline{\lambda}),$$

which is equivalent to

$$\nu_3(\underline{\lambda}, \kappa\underline{\lambda}, h_2^*) = 4\bar{L}(\bar{a}\kappa^2 + \bar{b}\underline{\lambda}\kappa)$$

for $\kappa \geq 1$ and $\underline{\lambda} > 0$. We observe that the optimal value for $\nu_3(\cdot)$ is obtained when $\underline{\lambda}$ goes to zero. Moreover, from $\kappa \geq 1$ and $\frac{\partial \nu_1}{\partial \kappa}(\underline{\lambda}, \kappa\underline{\lambda}, h_2^*) > 0$ for positive $\kappa$ it follows that $\kappa^* = 1$. Summarising the above discussion, in the non-differentiable case, $\underline{\lambda}^* = \overline{\lambda}^* \to 0$. To give an intuition for this phenomena, note that $u_k$ and $\hat{u}_k$ in Algorithm 1 and 2 are sampled from $\mathcal{N}(0, B^{-1})$ and the smaller the eigenvalues of $B$ get, the directions are sampled from a larger area and the smoothing process will be done over a larger domain. This leads to a smaller Lipschitz constant of the smoothed version of the non-differentiable function.

# D   A Non-differentiable Loss Function Satisfying MVI

In this section we prove that $f(x, y) = |x| - |y|$, $x, y \in \mathbb{R}$ along with $\mathcal{Z} = \mathbb{R}^2$ satisfies Assumption 4. Let $u_1, u_2 \sim \mathcal{N}(0, \sigma^2)$ and $z^* = (0, 0)$. Using (10), we know that

$$f_\mu(x, y) = E_{u_1, u_2}[f(x + \mu u_1, y + \mu u_2)] = E_{u_1}[|x + \mu u_1|] + E_{u_2}[|y + \mu u_2|].$$

To calculate these expected values, we note that $x + \mu u_1 \sim \mathcal{N}(x, \mu^2 \sigma^2)$. We define the random variable $Y \stackrel{\text{def}}{=} |x + \mu u_1|$. It is well known that $Y$ has a folded normal distribution and the intended expected value is the mean of $Y$. Thus,

$$E_{u_1}[|x + \mu u_1|] = \mu\sigma\sqrt{\frac{2}{\pi}} \exp\left(-\frac{x^2}{2\mu^2\sigma^2}\right) + x\left(1 - 2\Phi\left(-\frac{x}{\mu\sigma}\right)\right),$$

where $\Phi(x) = \frac{1}{\sqrt{2\pi}}\int_{-\infty}^{x} \exp(-\frac{t^2}{2})dt$ is the cumulative distribution function of a Gaussian distribution. Similarly we can obtain $E_{u_2}[|y + \mu u_2|]$ and we have

$$f_\mu(x, y) = \mu\sigma\sqrt{\frac{2}{\pi}}\left(\exp\left(-\frac{x^2}{2\mu^2\sigma^2}\right) - \exp\left(-\frac{y^2}{2\mu^2\sigma^2}\right)\right) + x - y - 2x\Phi\left(-\frac{x}{\mu\sigma}\right) + 2y\Phi\left(-\frac{y}{\mu\sigma}\right).$$

To obtain $F_\mu(z) = \begin{bmatrix} \nabla_x f_\mu(x, y) \\ -\nabla_y f_\mu(x, y) \end{bmatrix}$, we need to calculate $\nabla_x f_\mu(x, y)$ and $\nabla_y f_\mu(x, y)$. Taking the derivative of $f_\mu(x, y)$ with respect to $x$, we have

$$\nabla_x f_\mu(x, y) = \frac{-x}{\mu\sigma}\sqrt{\frac{2}{\pi}}\exp\left(-\frac{x^2}{2\mu^2\sigma^2}\right) + 1 - 2\Phi\left(\frac{-x}{\mu\sigma}\right) + \frac{2x}{\mu\sigma}\frac{1}{\sqrt{2\pi}}\exp\left(-\frac{x^2}{2\mu^2\sigma^2}\right) = 1 - 2\Phi\left(\frac{-x}{\mu\sigma}\right).$$

Similarly, the expression

$$\nabla_y f_\mu(x, y) = -1 + 2\Phi\left(\frac{-y}{\mu\sigma}\right),$$

is obtained. We are interested in checking if $\langle F_\mu(z), z - z^* \rangle \geq 0$ for all $x, y \in \mathbb{R}$. Substituting the terms, we have

$$\langle F_\mu(z), z - z^* \rangle = x\left(1 - 2\Phi\left(\frac{-x}{\mu\sigma}\right)\right) + y\left(1 - 2\Phi\left(\frac{-y}{\mu\sigma}\right)\right).$$

It is well known that $\Phi(-x) = 1 - \phi(x)$. Also, if $x < 0$ then $\phi(x) < 1/2$, if $x > 0$ then $\phi(x) > 1/2$, and $\phi(0) = 1/2$. Considering these facts, $x$ and $1 - 2\Phi(\frac{-x}{\mu\sigma})$ have the same sign and the same holds for $y$ and $1 - 2\Phi(\frac{-y}{\mu\sigma})$. Thus,

$$x\left(1 - 2\Phi\left(\frac{-x}{\mu\sigma}\right)\right) + y\left(1 - 2\Phi\left(\frac{-y}{\mu\sigma}\right)\right) \geq 0 \qquad \text{or} \qquad \langle F_\mu(z), z - z^* \rangle \geq 0.$$

## E $\;B$-Invariant upper bounds

In this section, we analyse how one can obtain upper bounds for $\frac{1}{N+1}\sum_{k=0}^N E_{\mathcal{U}_k}[\|F(\hat{z}_k)\|^2]$ or $\frac{1}{N+1}\sum_{k=0}^N E_{\mathcal{U}_k}[\|F_\mu(\hat{z}_k)\|^2]$ invariant of minimum and maximum eigenvalues of $B$. Towards this end, we modify Algorithm 2 as described below. The main differences between Algorithms 2 and 3 are the up-

---

**Algorithm 3** Modified Variance-Reduced ZO-EG

1: **Input**: $z_0 = (x_0, y_0) \in \mathcal{Z}$; $N \in \mathbb{N}$; $\{h_{1,k}\}_{k=0}^N, \{h_{2,k}\}_{k=0}^N \subset \mathbb{R}_{>0}$; $\mu > 0$; $B_1 = B_1^\top \succ 0$; $B_2 = B_2^\top \succ 0$; $\{t_k\}_{k=0}^N \subset \mathbb{N}$
2: **for** $k = 0, \ldots, N$ **do**
3: $\quad$ Sample $\hat{u}_{1,k}^0, \cdots, \hat{u}_{1,k}^{t_k}$ and $\hat{u}_{2,k}^0, \cdots, \hat{u}_{2,k}^{t_k}$ from $\mathcal{N}(0, B_1^{-1})$ and $\mathcal{N}(0, B_2^{-1})$
4: $\quad$ Calculate $G_\mu^0(z_k), \cdots, G_\mu^{t_k}(z_k)$ using $u^i = \hat{u}_k^i$, $i = 0, \ldots, t_k$, (14) and (13)
5: $\quad$ Compute $G_\mu(z_k) = \frac{1}{t_k}\sum_{i=0}^{t_k} G_\mu^i(z_k)$
6: $\quad$ Compute $\hat{z}_k = \text{Proj}_{\mathcal{Z}}(z_k - h_1(k)B^{-1}G_\mu(z_k))$
7: $\quad$ Sample $u_{1,k}^0, \cdots, u_{1,k}^{t_k}$ and $u_{2,k}^0, \cdots, u_{2,k}^{t_k}$ from $\mathcal{N}(0, B_1^{-1})$ and $\mathcal{N}(0, B_2^{-1})$
8: $\quad$ Calculate $G_\mu^0(\hat{z}_k), \cdots, G_\mu^{t_k}(\hat{z}_k)$ using $u^i = u_k^i$, $i = 0, \ldots, t_k$, (14) and (13)
9: $\quad$ Compute $G_\mu(\hat{z}_k) = \frac{1}{t_k}\sum_{i=0}^{t_k} G_\mu^i(\hat{z}_k)$
10: $\quad$ Compute $z_{k+1} = \text{Proj}_{\mathcal{Z}}(z_k - h_2(k)B^{-1}G_\mu(\hat{z}_k))$
11: **end for**
12: **return** $z_1, \ldots, z_N$

---

dates changed from $\hat{z}_k = \text{Proj}_{\mathcal{Z}}(z_k - h_1(k)G_\mu(z_k))$ and $z_{k+1} = \text{Proj}_{\mathcal{Z}}(z_k - h_2(k)G_\mu(\hat{z}_k))$ in Algorithm 2 to $\hat{z}_k = \text{Proj}_{\mathcal{Z}}(z_k - h_1(k)B^{-1}G_\mu(z_k))$ and $z_{k+1} = \text{Proj}_{\mathcal{Z}}(z_k - h_2(k)B^{-1}G_\mu(\hat{z}_k))$ in Algorithm 3 as well as the left multiplication of the random oracle by $B^{-1}$ in Algorithm 3. These modifications are similar to the one in (Nesterov & Spokoiny, 2017, (66)). Using these small modifications, we can obtain upper bounds for the average expected norm of the gradient operator invariant of eigenvalues of $B$. In the next subsections, we will show how the results will change alongside with their proofs.

### E.1 Unconstrained Settings

In this section, we present an analogue of Theorem 2 for the case where Algorithm 3 instead of Algorithm 2 is employed.

**Theorem 7.** *Let $f \colon \mathbf{Z} \to \mathbb{R}$ be continuously differentiable with Lipschitz continuous gradients with constant $L_1(f) > 0$. Let $\sigma^2$ be an upper bound on the variance of the random oracle defined in Assumption 1, $N \geq 0$ be*

*the number of iterations, $t_k \in \mathbb{N}$ be the number of samples in each iteration of Algorithm 2, $F_\mu$ be defined in (15) with smoothing parameter $\mu > 0$, $\mathcal{U}_k = [(u_0, \hat{u}_0), (u_1, \hat{u}_1), \cdots, (u_k, \hat{u}_k)]$, $k \in \{0, \ldots, N\}$, and $\rho$ denotes the weak MVI parameter in Definition 10. Moreover, let $\{z_k\}_{k \geq 0}$ and $\{\hat{z}_k\}_{k \geq 0}$ be the sequences generated by Algorithm 3, lines 6 and 10, respectively, and suppose that Assumption 2 is satisfied. Then, for any iteration $N$, with*

$$h_{1,k} = h_1 \leq \frac{1}{L_1(f)} \quad and \quad h_{2,k} = h_2 \in \left( \sqrt{\frac{2\rho}{L_1(f)}}, \frac{h_1}{2} \right],$$

*we have*

$$\frac{1}{N+1} \sum_{k=0}^{N} E_{\mathcal{U}_k}[\|F_\mu(\hat{z}_k)\|_*^2] \leq \frac{2L_1(f)\|z_0 - z^*\|^2}{(L_1(f)h_2^2 - 2\rho)(N+1)} + \frac{2\mu^2 L_1(f)d}{(L_1(f)h_2^2 - 2\rho)} + \frac{2\mu^2 L_1(f)^2 \rho (d+3)^3}{(L_1(f)h_2^2 - 2\rho)}$$
$$+ \frac{3\sigma^2}{L_1(f)(L_1(f)h_2^2 - 2\rho)} \frac{1}{(N+1)} \sum_{k=0}^{N} \frac{1}{t_k}. \tag{102}$$

*Proof of Theorem 7.* Considering Lemma 1, $h_2 > 0$, and letting

$$\xi_k := G_\mu(z_k) - F_\mu(z_k) \quad and \quad \hat{\xi}_k := G_\mu(\hat{z}_k) - F_\mu(\hat{z}_k) \tag{103}$$

(with $E_{u_k}[\hat{\xi}_k] = 0$ and $E_{\hat{u}_k}[\xi_k] = 0$), we have

$$0 \leq h_2 \langle F_\mu(\hat{z}_k), \hat{z}_k - z^* \rangle + h_2\rho\|F_\mu(\hat{z}_k)\|_*^2 + h_2\mu^2 L_1 d + h_2\rho\mu^2 L_1^2(d+3)^3$$
$$= h_2 \langle G_\mu(\hat{z}_k), \hat{z}_k - z^* \rangle - h_2 \langle \hat{\xi}_k, \hat{z}_k - z^* \rangle + h_2\rho\|F_\mu(\hat{z}_k)\|_*^2 + h_2\mu^2 L_1 d + h_2\rho\mu^2 L_1^2(d+3)^3$$
$$= h_2 \langle G_\mu(\hat{z}_k), z_{k+1} - z^* \rangle + h_2 \langle G_\mu(z_k), \hat{z}_k - z_{k+1} \rangle + h_2 \langle G_\mu(\hat{z}_k) - G_\mu(z_k), \hat{z}_k - z_{k+1} \rangle \tag{104a}$$
$$- h_2 \langle \hat{\xi}_k, \hat{z}_k - z^* \rangle + h_2\rho\|F_\mu(\hat{z}_k)\|_*^2 + h_2\mu^2 L_1 d + h_2\rho\mu^2 L_1^2(d+3)^3 \tag{104b}$$

Here, we have additionally used $L_1 = L_1(f)$ to shorten the expressions. As a next step, we derive a bound for the three terms in (104a). Considering $\mathcal{Z} = \mathbf{Z}$ and from Algorithm 2 line 8, we know that $z_k - z_{k+1} = h_2 B^{-1} G_\mu(\hat{z}_k)$. Thus, it holds that

$$h_2 \langle G_\mu(\hat{z}_k), z_{k+1} - z^* \rangle = \langle B(z_k - z_{k+1}), z_{k+1} - z^* \rangle$$
$$= \frac{1}{2}\|z^* - z_k\|^2 - \frac{1}{2}\|z^* - z_{k+1}\|^2 - \frac{1}{2}\|z_k - z_{k+1}\|^2. \tag{105}$$

Similarly, from Algorithm 2 line 5, we know that $z_k - \hat{z}_k = h_1 B^{-1} G_\mu(z_k)$ and thus the estimate

$$h_2 \langle G_\mu(z_k), \hat{z}_k - z_{k+1} \rangle = \frac{h_2}{h_1} \langle B(z_k - \hat{z}_k), \hat{z}_k - z_{k+1} \rangle$$
$$= \frac{h_2}{2h_1} (\|z_k - z_{k+1}\|^2 - \|z_k - \hat{z}_k\|^2 - \|z_{k+1} - \hat{z}_k\|^2) \tag{106}$$

is obtained. For the third term in (104a), we use the fact that the gradient of $f$ is Lipschitz continuous. Hence, for any $\alpha > 0$, the chain of inequalities

$$h_2 \langle G_\mu(\hat{z}_k) - G_\mu(z_k), \hat{z}_k - z_{k+1} \rangle \leq h_2 \|F_\mu(\hat{z}_k) - F_\mu(z_k)\|_* \|\hat{z}_k - z_{k+1}\| + h_2 \langle \hat{\xi}_k - \xi_k, \hat{z}_k - z_{k+1} \rangle$$
$$\leq h_2 L_1 \|\hat{z}_k - z_k\| \|\hat{z}_k - z_{k+1}\| + h_2 \langle \hat{\xi}_k - \xi_k, \hat{z}_k - z_{k+1} \rangle$$
$$\leq \frac{h_2 L_1 \alpha}{2} \|\hat{z}_k - z_k\|^2 + \frac{h_2 L_1}{2\alpha} \|\hat{z}_k - z_{k+1}\|^2 + h_2 \langle \hat{\xi}_k - \xi_k, \hat{z}_k - z_{k+1} \rangle, \tag{107}$$

is satisfied. The first inequality is due to the fact that for all $z \in \mathbf{Z}$ and $\tilde{z} \in \mathbf{Z}^*$ we have $\langle z, \tilde{z} \rangle \leq \|z\| \|\tilde{z}\|_*$ and this comes from the definition of the dual norm (Boyd, 2004, A.1.6).

Substituting (105), (106), and (107) in (104a), letting $r_k = \|z_k - z^*\|$, and noting that $z_k - z_{k+1} = h_2 B^{-1} G_\mu(\hat{z}_k)$ and $z_k - \hat{z}_k = h_1 B^{-1} G_\mu(z_k)$, we get

$$
\begin{aligned}
0 \le{}& \frac{1}{2}(r_k^2 - r_{k+1}^2) + \left(\frac{h_2}{2h_1} - \frac{1}{2}\right)\|z_k - z_{k+1}\|^2 + \left(\frac{h_2 L_1 \alpha}{2} - \frac{h_2}{2h_1}\right)\|\hat{z}_k - z_k\|^2 + h_2 \rho \|F_\mu(\hat{z}_k)\|_*^2 \\
&+ \left(\frac{h_2 L_1}{2\alpha} - \frac{h_2}{2h_1}\right)\|\hat{z}_k - z_{k+1}\|^2 - h_2 \langle \hat{\xi}_k, \hat{z}_k - z^* \rangle + h_2 \mu^2 L_1 d + h_2 \rho \mu^2 L_1^2 (d+3)^3 \\
&+ h_2 \langle \hat{\xi}_k - \xi_k, \hat{z}_k - z_{k+1} \rangle \\
={}& \frac{1}{2}(r_k^2 - r_{k+1}^2) + h_2^2 \left(\frac{h_2}{2h_1} - \frac{1}{2}\right)\|B^{-1}G_\mu(\hat{z}_k)\|^2 + h_1^2 \left(\frac{h_2 L_1 \alpha}{2} - \frac{h_2}{2h_1}\right)\|B^{-1}G_\mu(z_k)\|^2 + h_2 \rho \|F_\mu(\hat{z}_k)\|_*^2 \\
&+ \left(\frac{h_2 L_1}{2\alpha} - \frac{h_2}{2h_1}\right)\|\hat{z}_k - z_{k+1}\|^2 - h_2 \langle \hat{\xi}_k, \hat{z}_k - z^* \rangle + h_2 \mu^2 L_1 d + h_2 \rho \mu^2 L_1^2 (d+3)^3 \\
&+ h_2 \langle \hat{\xi}_k - \xi_k, \hat{z}_k - z_{k+1} \rangle. \tag{108}
\end{aligned}
$$

Rearranging the above terms and noting that $\|B^{-1}G_\mu(z_k)\| = \|G_\mu(z_k)\|_*$ and $\|B^{-1}G_\mu(\hat{z}_k)\| = \|G_\mu(\hat{z}_k)\|_*$, we have

$$
\begin{aligned}
h_2^2 \left(\frac{1}{2} - \frac{h_2}{2h_1}\right)\|G_\mu(\hat{z}_k)\|_*^2 \le{}& \frac{1}{2}(r_k^2 - r_{k+1}^2) + h_1^2 \left(\frac{h_2 L_1 \alpha}{2} - \frac{h_2}{2h_1}\right)\|G_\mu(z_k)\|_*^2 + h_2 \rho \|F_\mu(\hat{z}_k)\|_*^2 \\
&+ \left(\frac{h_2 L_1}{2\alpha} - \frac{h_2}{2h_1}\right)\|\hat{z}_k - z_{k+1}\|^2 - h_2 \langle \hat{\xi}_k, \hat{z}_k - z^* \rangle + h_2 \mu^2 L_1 d + h_2 \rho \mu^2 L_1^2 (d+3)^3 \\
&+ h_2 \langle \hat{\xi}_k - \xi_k, h_2 B^{-1} G_\mu(\hat{z}_k) - h_1 B^{-1} G_\mu(z_k) \rangle. \tag{109}
\end{aligned}
$$

Choosing $h_1 \le \frac{1}{L_1}$ and $\alpha = 1$ ensures that the second and third right-hand side terms of (109) are non-positive. For $\frac{1}{2} - \frac{h_2}{2h_1} > 0$ to hold, $h_2$ needs to satisfy $h_2 < h_1$. Considering these facts, we choose $\sqrt{\frac{2\rho}{L_1}} \le h_2 \le \frac{h_1}{2}$ and we have

$$
\begin{aligned}
\frac{h_2^2}{4}\|G_\mu(\hat{z}_k)\|_*^2 - \frac{\rho}{2L_1}|F_\mu(\hat{z}_k)|_*^2 \le{}& \frac{1}{2}(r_k^2 - r_{k+1}^2) - h_2 \langle \hat{\xi}_k, \hat{z}_k - z^* \rangle + \frac{\mu^2 d}{2} + \frac{\rho}{2}\mu^2 L_1 (d+3)^3 \\
&+ h_2 \langle \hat{\xi}_k - \xi_k, h_2 B^{-1} G_\mu(\hat{z}_k) - h_1 B^{-1} G_\mu(z_k) \rangle \\
\le{}& \frac{1}{2}(r_k^2 - r_{k+1}^2) + h_2 \langle \hat{\xi}_k, z^* - z_k + h_1 B^{-1} G_\mu(z_k) \rangle + \frac{\mu^2 d}{2\bar{\lambda}\kappa} + \frac{\rho}{2\bar{\lambda}\kappa}\mu^2 L_1 (d+3)^3 \\
&+ h_2 \langle \hat{\xi}_k - \xi_k, h_2 B^{-1} G_\mu(\hat{z}_k) - h_1 B^{-1} G_\mu(z_k) \rangle. \tag{110}
\end{aligned}
$$

For the last term of the inequality above, we have

$$
\begin{aligned}
h_2 \langle \hat{\xi}_k - \xi_k, h_2 B^{-1} G_\mu(\hat{z}_k) - h_1 B^{-1} G(z_k) \rangle &= h_2 \langle \hat{\xi}_k - \xi_k, h_2 B^{-1}(\hat{\xi}_k + F_\mu(\hat{z}_k)) - h_1 B^{-1}(\xi_k + F_\mu(z_k)) \rangle \\
&= h_2 \langle \hat{\xi}_k - \xi_k, h_2 B^{-1}\hat{\xi}_k - h_1 B^{-1}\xi_k \rangle + h_2 \langle \hat{\xi}_k - \xi_k, h_2 B^{-1} F_\mu(\hat{z}_k) - h_1 B^{-1} F_\mu(z_k) \rangle. \tag{111}
\end{aligned}
$$

For $h_2 \langle \hat{\xi}_k - \xi_k, h_2 B^{-1}\hat{\xi}_k - h_1 B^{-1}\xi_k \rangle$, we have

$$
h_2 \langle \hat{\xi}_k - \xi_k, h_2 B^{-1}\hat{\xi}_k - h_1 B^{-1}\xi_k \rangle = h_2^2 \|\hat{\xi}_k\|_*^2 + h_2 h_1 \|\xi_k\|_*^2 + h_2 \langle \xi_k, -h_2 B^{-1}\hat{\xi}_k \rangle + h_2 \langle \hat{\xi}_k, -h_1 B^{-1}\xi_k \rangle. \tag{112}
$$

Substituting (111) and (112) in (110), we have

$$
\begin{aligned}
\frac{h_2^2}{4}\|G_\mu(\hat{z}_k)\|_*^2 - \frac{\rho}{2L_1}\|F_\mu(\hat{z}_k)\|_*^2 \le{}& \frac{1}{2}(r_k^2 - r_{k+1}^2) + \frac{\mu^2 d}{2} + \frac{\rho}{2}\mu^2 L_1 (d+3)^3 + h_2^2 \|\hat{\xi}_k\|_*^2 + h_2 h_1 \|\xi_k\|_*^2 \\
&+ h_2 \langle \hat{\xi}_k, z^* - z_k + h_1 B^{-1} G_\mu(z_k) \rangle + h_2 \langle \hat{\xi}_k - \xi_k, h_2 B^{-1} F_\mu(\hat{z}_k) - h_1 B^{-1} F_\mu(z_k) \rangle \\
&+ h_2 \langle \xi_k, -h_2 B^{-1}\hat{\xi}_k \rangle + h_2 \langle \hat{\xi}_k, -h_1 B^{-1}\xi_k \rangle. \tag{113}
\end{aligned}
$$

From Jensen's inequality, we know that $E_{u_k}[\|G_\mu(\hat{z}_k)\|_*]^2 \leq E_{u_k}[\|G_\mu(\hat{z}_k)\|_*^2]$. Additionally, it can be concluded that $E_{u_k}[\|G_\mu(\hat{z}_k)\|_*] \geq \|E_{u_k}[G_\mu(\hat{z}_k)]\|_* = \|F_\mu(\hat{z}_k)\|_*$, and thus

$$E_{u_k}[\|G_\mu(\hat{z}_k)\|_*^2] \geq \|F_\mu(\hat{z}_k)\|_*^2. \tag{114}$$

Using this inequality and by taking the expected value of (113) with respect to $u_k$ and then with respect to $\hat{u}_k$, noting that $E_{u_k}[\hat{\xi}_k] = 0$, $E_{u_k}[\|\hat{\xi}_k\|_*^2] \leq \sigma^2/t_k$, $E_{\hat{u}_k}[\xi_k] = 0$, and $E_{\hat{u}_k}[\|\xi_k\|_*^2] \leq \sigma^2/t_k$, we have

$$\left(\frac{h_2^2}{4} - \frac{\rho}{2L_1}\right) E_{u_k,\hat{u}_k}[\|F_\mu(\hat{z}_k)\|_*^2] \leq \frac{1}{2}(r_k^2 - E_{u_k,\hat{u}_k}[r_{k+1}^2]) + \frac{\mu^2 d}{2} + \frac{\rho}{2}\mu^2 L_1(d+3)^3 + \frac{1}{4L_1^2 t_k}\sigma^2 + \frac{1}{2L_1^2 t_k}\sigma^2. \tag{115}$$

Since $u_k$ and $\hat{u}_k$ are independent by assumption, $\xi_k$, $z_k$, and $\hat{z}_k$ are independent of $u_k$, and $E_{u_k}[\hat{\xi}_k] = 0$, the expected value of the last four terms of (113) with respected to $u_k$ are zero.

Next, let $\mathcal{U}_k = [(u_0, \hat{u}_0), (u_1, \hat{u}_1), \cdots, (u_k, \hat{u}_k)]$ for $k \in \{0, \dots, N\}$. Computing the expected value of (115) with respect to $\mathcal{U}_{k-1}$, letting $\phi_k = E_{\mathcal{U}_{k-1}}[r_k^2]$ and $\phi_0^2 = r_0^2$, we have

$$\left(\frac{h_2^2}{4} - \frac{\rho}{2L_1}\right) E_{\mathcal{U}_k}[\|F_\mu(\hat{z}_k)\|_*^2] \leq \frac{1}{2}(\phi_k^2 - \phi_{k+1}^2) + \frac{\mu^2 d}{2} + \frac{\rho}{2}\mu^2 L_1(d+3)^3 + \frac{1}{4L_1^2 t_k}\sigma^2 + \frac{1}{2L_1^2 t_k}\sigma^2. \tag{116}$$

Summing (116) from $k = 0$ to $k = N$, and dividing it by $N + 1$, yields

$$\frac{1}{N+1} \sum_{k=0}^{N} E_{\mathcal{U}_k}[\|F_\mu(\hat{z}_k)\|_*^2] \leq \frac{2L_1\|z_0 - z^*\|^2}{(L_1 h_2^2 - 2\rho)(N+1)} + \frac{2\mu^2 L_1 d}{(L_1 h_2^2 - 2\rho)} + \frac{2\mu^2 L_1^2 \rho(d+3)^3}{(L_1 h_2^2 - 2\rho)} \tag{117}$$

$$+ \frac{3\sigma^2}{L_1(L_1 h_2^2 - 2\rho)} \frac{1}{(N+1)} \sum_{k=0}^{N} \frac{1}{t_k}, \tag{118}$$

which completes the proof $\qquad\qquad\square$

It can be seen that (102) is in the primal and dual norms, and the right-hand side of (102) is not dependent on the minimum and maximum eigenvalues of $B$ directly. Thus, with an appropriate $B$ invariant choice of hyperparameters, similar to the process in Section 3.1, we can show the convergence of the sequence generated by Algorithm 3 to an $\epsilon$-stationary point of the objective function in the expectation sense.

## E.2 Constrained Settings

In this section, we will show how Theorem 4 will change by using Algorithm 3 instead of Algorithm 2. Before proceeding further, we need to redefine (24) and (27) as below.

$$Q_\ell(z, a, F(\bar{z})) \stackrel{\text{def}}{=} -\frac{B}{a}(\text{Prox}_\ell(z - aB^{-1}F(\bar{z})) - z), \qquad \forall z, \bar{z} \in \mathcal{Z}, \tag{119}$$

and

$$P_{\mathcal{Z}}(z, h, g(\bar{z})) \stackrel{\text{def}}{=} \frac{B}{h} \left[z - \text{Proj}_{\mathcal{Z}}(z - hB^{-1}g(\bar{z}))\right], \tag{120}$$

where $h$ and $a$ are positive scalars. We note that when $\ell$ is the indicator function, then $P_{\mathcal{Z}}(z, h, g(\bar{z})) = Q_\ell(z, h, g(\bar{z}))$. Then we can define the below auxiliary variables with respect to (119) and (120):

$$s_k \stackrel{\text{def}}{=} P_{\mathcal{Z}}(z_k, h_{1,k}, G_\mu(z_k)), \qquad \hat{s}_k \stackrel{\text{def}}{=} P_{\mathcal{Z}}(z_k, h_{2,k}, G_\mu(\hat{z}_k)). \tag{121}$$

Hence, using above auxiliary variables in the constrained case of Problem 1, then the update steps in lines 5 and 8 in Algorithm 3 can be written as

$$z_{k+1} = z_k - h_{2,k} B^{-1} \hat{s}_k \quad \text{and} \quad \hat{z}_k = z_k - h_{1,k} B^{-1} s_k. \tag{122}$$

Next, we have the modified version of Lemma 2 with respect to (119) and (120).

**Lemma 8.** *Let $f(z)$ defined in Problem 1, be continuously differentiable with Lipschitz continuous gradient with constant $L_1(f) > 0$. Moreover, let $\hat{s}_k$ and $s_k$ be defined in (121), $\xi_k \stackrel{\text{def}}{=} G_\mu(z_k) - F_\mu(z_k)$, $\hat{\xi}_k \stackrel{\text{def}}{=} G_\mu(\hat{z}_k) - F_\mu(\hat{z}_k)$, $G_\mu$ and $F_\mu$ be defined in (13) and (15) with smoothing parameter $\mu > 0$, $\rho$ denote the proximal weak MVI parameter defined in Definition 11, and $D_z$ be the diameter of $\mathcal{Z} \subset \mathbb{R}^d$. If there exists $z^* \in \mathcal{Z}$ such that Assumption 3 is satisfied, then it holds that*

$$\langle s_k, z_k - z^* \rangle + \rho \|s_k\|_*^2 + \frac{\mu}{2} D_z L_1(f)(d+3)^{\frac{3}{2}} + D_z \|\xi_k\|_* + \frac{\mu^2}{2} \rho L_1^2(f)(d+3)^3 + 2\rho \|\xi_k\|_*^2 \geq 0, \qquad (123)$$

$$\langle \hat{s}_k, \hat{z}_k - z^* \rangle + \rho \|\hat{s}_k\|_*^2 + \frac{\mu}{2} D_z L_1(f)(d+3)^{\frac{3}{2}} + D_z \|\hat{\xi}_k\|_* + \frac{\mu^2}{2} \rho L_1^2(f)(d+3)^3 + 2\rho \|\hat{\xi}_k\|_*^2 \geq 0. \qquad (124)$$

*Proof of Lemma 8.* Let $s_k$ and $\hat{s}_k$, be defined in (28) and

$$v_k \stackrel{\text{def}}{=} P_{\mathcal{Z}}(z_k, h_{1,k}, F_\mu(z_k)), \ \hat{v}_k \stackrel{\text{def}}{=} P_{\mathcal{Z}}(z_k, h_{2,k}, F_\mu(\hat{z}_k)), \ p_k \stackrel{\text{def}}{=} P_{\mathcal{Z}}(z_k, h_{1,k}, F(z_k)), \ \hat{p}_k \stackrel{\text{def}}{=} P_{\mathcal{Z}}(z_k, h_{1,k}, F(\hat{z}_k)) \qquad (125)$$

Considering Algorithm 3 and $\Gamma(z)$ along with $\mathcal{Z}$ as defined in Section 3.2 when $\ell(z) = I_{\mathcal{Z}}(z)$, we have $\text{Prox}_\ell(\cdot) = \text{Proj}_{\mathcal{Z}}(\cdot)$, $Q_\ell(z_k, h_1, F(z_k)) = p_k$ and $Q_\ell(z_k, h_2, F(\hat{z}_k)) = \hat{p}_k$. Thus, when Assumption 3 is satisfied, we have

$$\langle p_k, z_k - z^* \rangle + \frac{\rho}{2} \|p_k\|_*^2 \geq 0, \qquad \langle \hat{p}_k, \hat{z}_k - z^* \rangle + \frac{\rho}{2} \|\hat{p}_k\|_*^2 \geq 0.$$

Considering the above inequalities, we have

$$0 \leq \langle p_k, z_k - z^* \rangle + \frac{\rho}{2} \|p_k\|_*^2$$

$$= \langle s_k, z_k - z^* \rangle + \langle p_k - v_k, z_k - z^* \rangle + \langle v_k - s_k, z_k - z^* \rangle + \frac{\rho}{2} \|s_k + (p_k - v_k) + (v_k - s_k)\|_*^2$$

$$\leq \langle s_k, z_k - z^* \rangle + \|p_k - v_k\|_* \|z_k - z^*\| + \langle v_k - s_k, z_k - z^* \rangle + \rho \|s_k\|_*^2 + \rho \|(p_k - v_k) + (v_k - s_k)\|_*^2$$

$$\leq \langle s_k, z_k - z^* \rangle + D_z \|F(z_k) - F_\mu(z_k)\|_* + D_z \|\xi_k\|_* + \rho \|s_k\|_*^2 + 2\rho \|F(z_k) - F_\mu(z_k)\|_*^2 + 2\rho \|\xi_k\|_*^2$$

$$\leq \langle s_k, z_k - z^* \rangle + \frac{\mu}{2} D_z L_1(f)(d+3)^{3/2} + D_z \|\xi_k\|_* + \rho \|s_k\|_*^2 + \frac{\mu^2}{2} \rho L_1^2(f)(d+3)^3 + 2\rho \|\xi_k\|_*^2. \qquad (126)$$

The third inequality is due to $\|p_k - v_k\|_* \leq \|F(z_k) - F_\mu(z_k)\|_*$ and $\|v_k - s_k\|_* \leq \|F_\mu(z_k) - G_\mu(z_k)\|_*$ which can be obtained directly from the non-expansiveness of the projection operator. The last inequality is due to Lemma 7. Inequality (126) proves (123). A proof of (124) follows the same arguments. □

Next, we will have the modified version of Theorem 4.

**Theorem 8.** *Let $f(z)$, defined in Problem 1, be continuously differentiable with Lipschitz continuous gradient with constant $L_1(f) > 0$. Let $\sigma^2$ be an upper bound on the variance of the random oracle defined in Assumption 1, $N \geq 0$ be the number of iterations, $t_k$ be the number of samples in each iteration of Algorithm 3, $s_k$ be defined in (28) with smoothing parameter $\mu > 0$, $\mathcal{U}_k = [(u_0, \hat{u}_0), (u_1, \hat{u}_1), \cdots, (u_k, \hat{u}_k)], \ k \in \{0, \ldots, N\}$, $\rho$ denotes the proximal weak MVI parameter defined in Definition 11, and $D_z$ be diameter of the compact and convex set $\mathcal{Z} \subset \mathbb{R}^d$. Moreover, let $\{z_k\}_{k \geq 0}$ and $\{\hat{z}_k\}_{k \geq 0}$ be the sequences generated by Algorithm 3, respectively, suppose that Assumption 2 is satisfied. Then, for any iteration $N$, with $h_{1,k} = h_{2,k} = h$ and $h \in \left( \sqrt{\frac{6\rho}{L_1(f)}}, \frac{1}{2L_1(f)} \right]$, we have*

$$\frac{1}{N+1} \sum_{k=0}^N E_{\mathcal{U}_k}[\|s_k\|_*^2] \leq \frac{2L_1(f)\|z_0 - z^*\|_*^2}{(L_1(f)h^2 - 6\rho)(N+1)} + \frac{\mu D_z L_1(f)(d+3)^{3/2}}{L_1(f)h^2 - 6\rho} + \frac{\mu^2 \rho L_1(f)^2(d+3)^3}{L_1(f)h^2 - 6\rho}$$

$$+ \frac{(36\rho + \frac{4}{L_1(f)})\sigma^2}{L_1(f)h^2 - 6\rho} \frac{1}{N+1} \sum_{k=0}^N \frac{1}{t_k} + \frac{2D_z\sigma}{L_1(f)h^2 - 6\rho} \frac{1}{N+1} \sum_{k=0}^N \frac{1}{\sqrt{t_k}}. \qquad (127)$$

*Proof of Theorem 8.* In the following, we use $L_1 = L_1(f)$ to shorten expressions. Considering $\xi_k = G_\mu(z_k) - F_\mu(z_k)$ and $\hat{\xi}_k = G_\mu(\hat{z}_k) - F_\mu(\hat{z}_k)$, $h_1 = h_2 = h$, $\sqrt{\frac{6\rho}{L_1}} < h \le \frac{1}{2L_1}$ and $\rho \le \frac{1}{24L_1}$, the following estimate holds:

$$
\begin{aligned}
\|z_{k+1} - z^*\|^2 &= \|z_k - hB^{-1}\hat{s}_k - z^*\|^2 \\
&= \|z_k - z^*\|^2 + h^2\|B^{-1}\hat{s}_k\|^2 - 2h\langle BB^{-1}\hat{s}_k, z_k - z^*\rangle \\
&= \|z_k - z^*\|^2 + h^2\|\hat{s}_k\|_*^2 - 2h\langle \hat{s}_k, z_k - \hat{z}_k\rangle - 2h\langle \hat{s}_k, \hat{z}_k - z^*\rangle \\
&\le \|z_k - z^*\|^2 + h^2\|\hat{s}_k\|_*^2 - 2h\langle \hat{s}_k, z_k - \hat{z}_k\rangle + 2h\Big(\rho\|\hat{s}_k\|_*^2 \\
&\quad + \frac{\mu}{2}D_z L_1(f)(d+3)^{\frac{3}{2}} + D_z\|\hat{\xi}_k\|_* + \frac{\mu^2}{2}\rho L_1^2(f)(d+3)^3 + 2\rho\|\hat{\xi}_k\|_*^2\Big) \\
&= \|z_k - z^*\|^2 + h^2\|\hat{s}_k\|_*^2 - 2h^2\langle \hat{s}_k, B^{-1}s_k\rangle + 2h\rho\|\hat{s}_k\|_*^2 \\
&\quad + 2h\Big(\frac{\mu}{2}D_z L_1(f)(d+3)^{\frac{3}{2}} + D_z\|\hat{\xi}_k\|_* + \frac{\mu^2}{2}\rho L_1^2(f)(d+3)^3 + 2\rho\|\hat{\xi}_k\|_*^2\Big) \\
&\le \|z_k - z^*\|^2 + h^2(\|\hat{s}_k - s_k\|_*^2 - \|s_k\|_*^2) + 4h\rho(\|\hat{s}_k - s_k\|_*^2 + \|s_k\|_*^2) \\
&\quad + 2h\Big(\frac{\mu}{2}D_z L_1(f)(d+3)^{\frac{3}{2}} + D_z\|\hat{\xi}_k\|_* + \frac{\mu^2}{2}\rho L_1^2(f)(d+3)^3 + 2\rho\|\hat{\xi}_k\|_*^2\Big).
\end{aligned}
\tag{128}
$$

The first inequality is obtained using Lemma 8, and the second inequality is obtained by completing squares. Next, we derive an upper bound for the term $\|\hat{s}_k - s_k\|_*^2$. We have

$$
\begin{aligned}
\|\hat{s}_k - s_k\|_*^2 &\le \|(\hat{v}_k - v_k) + (\hat{s}_k - \hat{v}_k) + (v_k - s_k)\|_*^2 \\
&\le 2\|\hat{v}_k - v_k\|_*^2 + 4\|\hat{s}_k - \hat{v}_k\|_*^2 + 4\|v_k - s_k\|_*^2 \\
&\le 2\|F_\mu(\hat{z}_k) - F_\mu(z_k)\|_*^2 + 4\|G_\mu(z_k) - F_\mu(z_k)\|_*^2 + 4\|G_\mu(\hat{z}_k) - F_\mu(\hat{z}_k)\|_*^2 \\
&\le 2L_1^2\|\hat{z}_k - z_k\|^2 + 4\|\xi_k\|_*^2 + 4\|\hat{\xi}_k\|_*^2 \\
&= 2h^2 L_1^2\|s_k\|_*^2 + 4\|\xi_k\|_*^2 + 4\|\hat{\xi}_k\|_*^2.
\end{aligned}
\tag{129}
$$

The third inequality is due to the inequalities $\|s_k - v_k\|_* \le \|G_\mu(z_k) - F_\mu(z_k)\|_*$, $\|\hat{s}_k - \hat{v}_k\|_* \le \|G_\mu(\hat{z}_k) - F_\mu(\hat{z}_k)\|_*$, and $\|\hat{v}_k - v_k\|_* \le \|F_\mu(\hat{z}_k) - F_\mu(z_k)\|_*$, which can be directly obtained from the non-expansiveness of the projection operator. The forth inequality is obtained using the fact that the gradient of the objective function is Lipchitz. Plugging (129) in (128), we have

$$
\begin{aligned}
\|z_{k+1} - z^*\|^2 &\le \|z_k - z^*\|^2 + h^2(2h^2 L_1^2\|s_k\|_*^2 + 4\|\xi_k\|_*^2 + 4\|\hat{\xi}_k\|_*^2 - \|s_k\|_*^2) \\
&\quad + 4h\rho\Big(2h^2 L_1^2\|s_k\|_*^2 + 4\|\xi_k\|_*^2 + 4\|\hat{\xi}_k\|_*^2 + \|s_k\|_*^2\Big) + 2h\Big(\frac{\mu}{2}D_z L_1(f)(d+3)^{\frac{3}{2}} \\
&\quad + D_z\|\hat{\xi}_k\|_* + \frac{\mu^2}{2}\rho L_1^2(f)(d+3)^3 + 2\rho\|\hat{\xi}_k\|_*^2\Big) \\
&\le \|z_k - z^*\|^2 + h^2(2L_1^2 h^2 - 1)\|s_k\|_*^2 + 4h\rho(2h^2 L_1^2 + 1)\|s_k\|_*^2 + \mu h D_z L_1(d+3)^{\frac{3}{2}} \\
&\quad + \mu^2 h\rho L_1^2(d+3)^3 + 2h D_z\|\hat{\xi}_k\|_* + (20h\rho + 4h^2)\|\hat{\xi}_k\|_*^2 + (16h\rho + 4h^2)\|\xi_k\|_*^2 \\
&\le \|z_k - z^*\|^2 - \frac{h^2}{2}\|s_k\|_*^2 + \frac{3\rho}{L_1}\|s_k\|_*^2 + \frac{\mu}{2}D_z(d+3)^{\frac{3}{2}} + \frac{\mu^2}{2}\rho L_1(d+3)^3 \\
&\quad + \frac{D_z}{L_1}\|\hat{\xi}_k\|_* + \Big(\frac{10\rho}{L_1} + \frac{1}{L_1^2}\Big)\|\hat{\xi}_k\|_*^2 + \Big(\frac{8\rho}{L_1} + \frac{1}{L_1^2}\Big)\|\xi_k\|_*^2.
\end{aligned}
\tag{130}
$$

Letting $r_k^2 = |z_k - z^*|^2$, the inequality

$$
\begin{aligned}
\Big(\frac{h^2}{2} - \frac{3\rho}{L_1}\Big)\|s_k\|_*^2 &\le r_k^2 - r_{k+1}^2 + \frac{\mu}{2}D_z(d+3)^{\frac{3}{2}} + \frac{\mu^2}{2}\rho L_1(d+3)^3 + \frac{D_z}{L_1}\|\hat{\xi}_k\| \\
&\quad + \Big(\frac{10\rho}{L_1} + \frac{1}{L_1^2}\Big)\|\hat{\xi}_k\|_*^2 + \Big(\frac{8\rho}{L_1} + \frac{1}{L_1^2}\Big)\|\xi_k\|_*^2
\end{aligned}
\tag{131}
$$

holds. As a next step, we compute the expected value of (131) with respect to $u_k$ and then with respect to $\hat{u}_k$, and we use the fact that $E_{u_k}[\|\hat{\xi}_k\|_*] \leq \sigma/t_k$, $E_{\hat{u}_k}[\|\xi_k\|_*] \leq \sigma/t_k$, which follows from the assumptions of Theorem 8 and Jensen's inequality. Let $\mathcal{U}_k = [(u_0, \hat{u}_0), (u_1, \hat{u}_1), \cdots, (u_k, \hat{u}_k)]$ for $k \in \{0, \ldots, N\}$ and $E_{\mathcal{U}_{k-1}}[r_k^2] = \phi_k^2$. Then, Computing expected value of (131) with respect to $\mathcal{U}_{k-1}$, summing from $k = 0$ to $k = N$, and dividing it by $N + 1$, yields

$$
\frac{1}{N+1} \sum_{k=0}^{N} E_{\mathcal{U}_k}[\|s_k\|_*^2] \leq \frac{2L_1\|z_0 - z^*\|^2}{(L_1 h^2 - 6\rho)(N+1)} + \frac{\mu D_z L_1 (d+3)^{3/2}}{L_1 h^2 - 6\rho} + \frac{\mu^2 \rho L_1^2 (d+3)^3}{L_1 h^2 - 6\rho}
$$
$$
+ \frac{(36\rho + \frac{4}{L_1})\sigma^2}{L_1 h^2 - 6\rho} \frac{1}{N+1} \sum_{k=0}^{N} \frac{1}{t_k} + \frac{2D_z \sigma}{L_1 h^2 - 6\rho} \frac{1}{N+1} \sum_{k=0}^{N} \frac{1}{\sqrt{t_k}},
$$
(132)

which completes the proof. $\qquad\square$

It can be seen that (127) is in the primal and dual norms, and the right-hand side of (127) is not dependent on the minimum and maximum eigenvalues of $B$ directly. Thus, with an appropriate $B$ invariant choice of hyperparameters, similar to the process in Section 3.2, we can show the convergence of the sequence generated by Algorithm 3 to an $\epsilon$-stationary point of the objective function in the expectation sense.

### E.3    Non-differentiable Settings

In this section, we will show how Theorem 6 will change by using Algorithm 3 instead of Algorithm 2.

**Theorem 9.** *Let $f(z)$, defined in Problem 1, be Lipschitz continuous with constant $L_0(f) > 0$. Let $\sigma^2$ be an upper bound on the variance of the random oracle defined in Assumption 1, $N \geq 0$ be the number of iterations, $t_k$ be the number of samples in each iteration of Algorithm 3, $F_\mu$ be defined in (15) with smoothing parameter $\mu > 0$, $\mathcal{U}_k = [(u_0, \hat{u}_0), (u_1, \hat{u}_1), \cdots, (u_k, \hat{u}_k)]$, $k \in \{0, \ldots, N\}$, $\rho$ denotes the weak MVI parameter defined in Assumption 4, and $L_1(f_\mu)$ be the Lipschitz constant of the gradient of $f_\mu$. Moreover, let $\{z_k\}_{k \geq 0}$ and $\{\hat{z}_k\}_{k \geq 0}$ be the sequences generated by Algorithm 1 (see lines 5 and 8) and suppose Assumption 4 is satisfied. Then, for any number of iterations $N$, with $h_{1,k} = h_1 \leq \frac{1}{L_1(f_\mu)}$ and $h_{2,k} = h_2 \in \left( \sqrt{\frac{\rho}{L_1(f_\mu)}}, \frac{h_1}{2} \right]$, we have*

$$
\frac{1}{N+1} \sum_{k=0}^{N} E_{\mathcal{U}_k}[\|F_\mu(\hat{z}_k)\|_*^2] \leq \frac{2L_1(f_\mu)\|z_0 - z^*\|^2}{(L_1(f_\mu)h_2^2 - \rho)(N+1)} + \frac{3\sigma^2}{L_1(f_\mu)(L_1(f_\mu)h_2^2 - \rho)} \frac{1}{N+1} \sum_{k=0}^{N} \frac{1}{t_k}.
$$
(133)

*Proof of Theorem 9.* Considering Assumption 4, $h_2 > 0$, and letting $\xi_k = G_\mu(z_k) - F_\mu(z_k)$ and $\hat{\xi}_k = G_\mu(\hat{z}_k) - F_\mu(\hat{z}_k)$ (and recalling that $E_{u_k}[\hat{\xi}_k] = 0$ and $E_{\hat{u}_k}[\xi_k] = 0$), we have

$$
0 \leq h_2 \langle F_\mu(\hat{z}_k), \hat{z}_k - z^* \rangle + h_2 \frac{\rho}{2} \|F_\mu(\hat{z}_k)\|_*^2
$$
$$
= h_2 \langle G_\mu(\hat{z}_k), \hat{z}_k - z^* \rangle - h_2 \langle \hat{\xi}_k, \hat{z}_k - z^* \rangle + h_2 \frac{\rho}{2} \|F_\mu(\hat{z}_k)\|_*^2
$$
$$
= h_2 \langle G_\mu(\hat{z}_k), z_{k+1} - z^* \rangle + h_2 \langle G_\mu(\hat{z}_k), \hat{z}_k - z_{k+1} \rangle + h_2 \langle G_\mu(\hat{z}_k) - G_\mu(z_k), \hat{z}_k - z_{k+1} \rangle
$$
$$
- h_2 \langle \hat{\xi}_k, \hat{z}_k - z^* \rangle + h_2 \frac{\rho}{2} \|F_\mu(\hat{z}_k)\|_*^2.
$$
(134a)

As a first step, we derive a bound for the first three terms in (134a). Considering that $\mathcal{Z} = \mathbf{Z}$, from Algorithm 3 line 8, we know that $z_k - z_{k+1} = h_2 B^{-1} G_\mu(\hat{z}_k)$. Thus it holds that

$$
h_2 \langle G_\mu(\hat{z}_k), z_{k+1} - z^* \rangle = \langle B(z_k - z_{k+1}), z_{k+1} - z^* \rangle
$$
$$
= \frac{1}{2}\|z^* - z_k\|^2 - \frac{1}{2}\|z^* - z_{k+1}\|^2 - \frac{1}{2}\|z_k - z_{k+1}\|^2.
$$
(135)

Similarly, form Algorithm 3 line 5, we know that $z_k - \hat{z}_k = h_1 B^{-1} G_\mu(z_k)$, and thus the estimate

$$
h_2 \langle G_\mu(z_k), \hat{z}_k - z_{k+1} \rangle = \frac{h_2}{h_1} \langle B(z_k - \hat{z}_k), \hat{z}_k - z_{k+1} \rangle
$$
$$
= \frac{h_2}{2h_1} (\|z_k - z_{k+1}\|^2 - \|z_k - \hat{z}_k\|^2 - \|z_{k+1} - \hat{z}_k\|^2) \tag{136}
$$

is obtained. For the third term in (134a), considering that the gradient of $f_\mu$ is Lipschitz continuous, for any $\alpha > 0$, we have

$$
h_2 \langle G_\mu(\hat{z}_k) - G_\mu(z_k), \hat{z}_k - z_{k+1} \rangle \leq h_2 \|F_\mu(\hat{z}_k) - F_\mu(z_k)\|_* \|\hat{z}_k - z_{k+1}\| + h_2 \langle \hat{\xi}_k - \xi_k, \hat{z}_k - z_{k+1} \rangle
$$
$$
\leq h_2 L_1(f_\mu) \|\hat{z}_k - z_k\| \|\hat{z}_k - z_{k+1}\| + h_2 \langle \hat{\xi}_k - \xi_k, \hat{z}_k - z_{k+1} \rangle
$$
$$
\leq \frac{h_2 L_1(f_\mu) \alpha}{2} \|\hat{z}_k - z_k\|^2 + \frac{h_2 L_1(f_\mu)}{2\alpha} \|\hat{z}_k - z_{k+1}\|^2 + h_2 \langle \hat{\xi}_k - \xi_k, \hat{z}_k - z_{k+1} \rangle. \tag{137}
$$

The first inequality is due to the fact that for all $z \in \mathbf{Z}$ and $\tilde{z} \in \mathbf{Z}^*$ we have $\langle z, \tilde{z} \rangle \leq \|z\| \|\tilde{z}\|_*$ and this comes from the definition of the dual norm (Boyd, 2004, A.1.6).

Substituting (135), (136), and (137) in (134a), letting $r_k = \|z_k - z^*\|$, and noting that $z_k - z_{k+1} = h_2 B^{-1} G_\mu(\hat{z}_k)$ and $z_k - \hat{z}_k = h_1 B^{-1} G_\mu(z_k)$, we get

$$
0 \leq \frac{1}{2}(r_k^2 - r_{k+1}^2) + \left(\frac{h_2}{2h_1} - \frac{1}{2}\right) \|z_k - z_{k+1}\|^2 + \left(\frac{h_2 L_1(f_\mu)\alpha}{2} - \frac{h_2}{2h_1}\right) \|\hat{z}_k - z_k\|^2
$$
$$
+ \left(\frac{h_2 L_1(f_\mu)}{2\alpha} - \frac{h_2}{2h_1}\right) \|\hat{z}_k - z_{k+1}\|^2 - h_2 \langle \hat{\xi}_k, \hat{z}_k - z^* \rangle + h_2 \langle \hat{\xi}_k - \xi_k, \hat{z}_k - z_{k+1} \rangle + h_2 \frac{\rho}{2} \|F_\mu(\hat{z}_k)\|_*^2
$$
$$
= \frac{1}{2}(r_k^2 - r_{k+1}^2) + h_2^2 \left(\frac{h_2}{2h_1} - \frac{1}{2}\right) \|B^{-1} G_\mu(\hat{z}_k)\|^2 + h_1^2 \left(\frac{h_2 L_1(f_\mu)\alpha}{2} - \frac{h_2}{2h_1}\right) \|B^{-1} G_\mu(z_k)\|^2
$$
$$
+ \left(\frac{h_2 L_1(f_\mu)}{2\alpha} - \frac{h_2}{2h_1}\right) \|\hat{z}_k - z_{k+1}\|^2 - h_2 \langle \hat{\xi}_k, \hat{z}_k - z^* \rangle + h_2 \langle \hat{\xi}_k - \xi_k, \hat{z}_k - z_{k+1} \rangle + h_2 \frac{\rho}{2} \|F_\mu(\hat{z}_k)\|_*^2.
$$

Rearranging the above terms and noting that $\|B^{-1} G_\mu(z_k)\| = \|G_\mu(z_k)\|_*$ and $\|B^{-1} G_\mu(\hat{z}_k)\| = \|G_\mu(\hat{z}_k)\|_*$, we have

$$
h_2^2 \left(\frac{1}{2} - \frac{h_2}{2h_1}\right) \|G_\mu(\hat{z}_k)\|_*^2 \leq \frac{1}{2}(r_k^2 - r_{k+1}^2) + h_1^2 \left(\frac{h_2 L_1(f_\mu)\alpha}{2} - \frac{h_2}{2h_1}\right) \|G_\mu(z_k)\|_*^2 + h_2 \frac{\rho}{2} \|F_\mu(\hat{z}_k)\|_*^2
$$
$$
+ \left(\frac{h_2 L_1(f_\mu)}{2\alpha} - \frac{h_2}{2h_1}\right) \|\hat{z}_k - z_{k+1}\|^2 - h_2 \langle \hat{\xi}_k, \hat{z}_k - z^* \rangle + h_2 \langle \hat{\xi}_k - \xi_k, h_2 B^{-1} G_\mu(\hat{z}_k) - h_1 B^{-1} G(z_k) \rangle. \tag{138}
$$

Choosing $h_1 \leq \frac{1}{L_1(f_\mu)}$ and $\alpha = 1$ ensures that the second and third right-hand side terms of (138) are non-positive. Also, we need $\frac{1}{2} - \frac{h_2}{2h_1} > 0$ and thus $h_2 < h_1$ needs to be satisfied. Considering $\sqrt{\frac{\rho}{L_1(f_\mu)}} \leq h_2 \leq \frac{h_1}{2}$, we have

$$
\frac{h_2^2}{4} \|G_\mu(\hat{z}_k)\|_*^2 \leq \frac{1}{2}(r_k^2 - r_{k+1}^2) - h_2 \langle \hat{\xi}_k, \hat{z}_k - z^* \rangle + h_2 \langle \hat{\xi}_k - \xi_k, h_2 B^{-1} G_\mu(\hat{z}_k) - h_1 B^{-1} G(z_k) \rangle + \frac{h_2 \rho}{2} \|F_\mu(\hat{z}_k)\|_*^2
$$
$$
\leq \frac{1}{2}(r_k^2 - r_{k+1}^2) + h_2 \langle \hat{\xi}_k, z^* - z_k + h_1 B^{-1} G_\mu(z_k) \rangle + \frac{\rho}{4L_1(f_\mu)} \|F_\mu(\hat{z}_k)\|_*^2 + h_2 \langle \hat{\xi}_k - \xi_k, h_2 B^{-1} G_\mu(\hat{z}_k) - h_1 B^{-1} G(z_k) \rangle. \tag{139}
$$

The last term of the inequality above can be equivalently written as

$$
h_2 \langle \hat{\xi}_k - \xi_k, h_2 B^{-1} G_\mu(\hat{z}_k) - h_1 B^{-1} G(z_k) \rangle = h_2 \langle \hat{\xi}_k - \xi_k, h_2 B^{-1}(\hat{\xi}_k + F_\mu(\hat{z}_k)) - h_1 B^{-1}(\xi_k + F_\mu(z_k)) \rangle
$$
$$
= h_2 \langle \hat{\xi}_k - \xi_k, h_2 B^{-1} \hat{\xi}_k - h_1 B^{-1} \xi_k \rangle + h_2 \langle \hat{\xi}_k - \xi_k, h_2 B^{-1} F_\mu(\hat{z}_k) - h_1 B^{-1} F_\mu(z_k) \rangle. \tag{140}
$$

Moreover, for $h_2 \langle \hat{\xi}_k - \xi_k, h_2 B^{-1} \hat{\xi}_k - h_1 B^{-1} \xi_k \rangle$ the equality

$$
h_2 \langle \hat{\xi}_k - \xi_k, h_2 B^{-1} \hat{\xi}_k - h_1 B^{-1} \xi_k \rangle = h_2^2 \|\hat{\xi}_k\|_*^2 + h_2 h_1 \|\xi_k\|_*^2 + h_2 \langle \xi_k, -h_2 B^{-1} \hat{\xi}_k \rangle + h_2 \langle \hat{\xi}_k, -h_1 B^{-1} \xi_k \rangle \tag{141}
$$

holds. Substituting (140) and (141) in (139), we have

$$
\begin{aligned}
\frac{h_2^2}{4}\|G_\mu(\hat{z}_k)\|_*^2 - \frac{\rho}{4L_1(f_\mu)}\|F_\mu(\hat{z}_k)\|_*^2 &\leq \frac{1}{2}(r_k^2 - r_{k+1}^2) + h_2^2\|\hat{\xi}_k\|_*^2 + h_2 h_1\|\xi_k\|_*^2 \\
&+ h_2\langle\hat{\xi}_k, z^* - z_k + h_1 B^{-1}G_\mu(z_k)\rangle + h_2\langle\hat{\xi}_k - \xi_k, h_2 B^{-1}F_\mu(\hat{z}_k) - h_1 B^{-1}F_\mu(z_k)\rangle \\
&+ h_2\langle\xi_k, -h_2 B^{-1}\hat{\xi}_k\rangle + h_2\langle\hat{\xi}_k, -h_1 B^{-1}\xi_k\rangle.
\end{aligned}
\tag{142}
$$

From Jensen's inequality, we know that $E_{u_k}[\|G_\mu(\hat{z}_k)\|_*]^2 \leq E_{u_k}[\|G_\mu(\hat{z}_k)\|_*^2]$. Also, it can be concluded that $E_{u_k}[\|G_\mu(\hat{z}_k)\|_*] \geq \|E_{u_k}[G_\mu(\hat{z}_k)]\|_* = \|F_\mu(\hat{z}_k)\|_*$, and thus

$$
E_{u_k}[\|G_\mu(\hat{z}_k)\|_*^2] \geq \|F_\mu(\hat{z}_k)\|_*^2.
\tag{143}
$$

Using this inequality and the taking the expected value of (139) with respect to $u_k$ and then with respect to $\hat{u}_k$, noting that $E_{u_k}[\hat{\xi}_k] = 0$, $E_{u_k}E_{u_k}[\|\hat{\xi}_k\|_*^2] \leq \sigma^2/t_k$, $E_{\hat{u}_k}[\xi_k] = 0$, and $E_{\hat{u}_k}[\|\xi_k\|_*^2] \leq \sigma^2/t_k$, we have

$$
\left(\frac{h_2^2}{4} - \frac{\rho}{4L_1(f_\mu)}\right)E_{u_k,\hat{u}_k}[\|F_\mu(\hat{z}_k)\|_*^2] \leq \frac{1}{2}(r_k^2 - E_{u_k,\hat{u}_k}[r_{k+1}^2]) + h_2^2\frac{\sigma^2}{t_k} + \frac{h_2}{L_1(f_\mu)t_k}\sigma^2.
\tag{144}
$$

Since $u_k$ and $\hat{u}_k$ are independent by assumption, $\xi_k$, $z_k$, and $\hat{z}_k$ are independent of $u_k$, and $E_{u_k}[\hat{\xi}_k] = 0$, the expected value of the last four terms of (142) with respected to $u_k$ are zero.

Next, let $\mathcal{U}_k = [(u_0, \hat{u}_0), (u_1, \hat{u}_1), \cdots, (u_k, \hat{u}_k)]$ for $k \in \{0, \ldots, N\}$. Computing the expected value of (144) with respect to $\mathcal{U}_{k-1}$, letting $\phi_k = E_{\mathcal{U}_{k-1}}[r_k^2]$ and $\phi_0^2 = r_0^2$, we have

$$
\left(h_2^2 - \frac{\rho}{L_1(f_\mu)}\right)E_{\mathcal{U}_k}[\|F_\mu(\hat{z}_k)\|_*^2] \leq 2(\phi_k^2 - \phi_{k+1}^2) + \frac{3}{L_1^2(f_\mu)t_k}\sigma^2.
\tag{145}
$$

Summing (145) from $k = 0$ to $k = N$, and dividing it by $N + 1$, yields

$$
\frac{1}{N+1}\sum_{k=0}^{N}E_{\mathcal{U}_k}[\|F_\mu(\hat{z}_k)\|_*^2] \leq \frac{2L_1(f_\mu)\|z_0 - z^*\|^2}{(L_1(f_\mu)h_2^2 - \rho)(N+1)} + \frac{3\sigma^2}{L_1(f_\mu)(L_1(f_\mu)h_2^2 - \rho)}\frac{1}{N+1}\sum_{k=0}^{N}\frac{1}{t_k},
$$

which completes the proof. $\qquad\square$

It can be seen that (133) is in the primal and dual norms, and the right-hand side of (133) is not dependent on the minimum and maximum eigenvalues of $B$ directly. Thus, with an appropriate $B$ invariant choice of hyperparameters, similar to the process in Section 3.3, we can show the convergence of the sequence generated by Algorithm 3 to n $(\delta, \epsilon)$-stationary point of the objective function in the expectation sense.

