# OpenReview forum: "Min-Max Optimisation for Nonconvex-Nonconcave Functions Using a Random Zeroth-Order Extragradient Algorithm"
_TMLR — Accepted by TMLR_

### Review · Reviewer_HpAX · 2025-05-12

**Summary Of Contributions:**

This paper analyzes the theoretical performance of ZO-EG (zeroth-order extragradient), which uses random Gaussian smoothing. The main problem setting is minimax optimization of (possibly) nonconvex-noncave objective function, either having globally Lipschitz continuous gradients (thus, continuously differentiable) or being globally Lipschitz continuous (and possibly not differentiable everywhere), defined on either entire d-dimensional Euclidean space (i.e., unconstrained case) or a convex and compact constraint set with finite diameter (i.e. constrained case). The main goal is to show the convergence to a neighborhood of an $\epsilon$-stationary point.

This paper shows that, for an unconstrained minimax problem on an objective function with Lipschitz continuous gradients, the squared gradient norm of (smoothed) function (taking average over iterations and mean over iid sampling of direction vectors) is bounded by three terms: one vanishing as taking many iterations, one vanishing as taking a small Gaussian smoothing parameter, and one proportional to the variance of the zeroth-order random gradient oracle. The first and the second terms can be made small ($O(\epsilon)$) by taking the number of iterations as $\Omega(\epsilon^{-2})$ and the Gaussian smoothing parameter as $O(d^{-3/2} \epsilon)$, while the third term can be made small if we use variance reduction scheme (sampling many random directions to compute gradient oracle).

The second result is quite similar for constrained problems with a finite diameter of the given constraint set. One difference is that the convergence metric is slightly different from the gradient norm; instead, a sequence of auxiliary variables defined with the proximal (or projection) operator is used, which subsumes the gradient operator in the unconstrained case. Another difference is that the third term of the convergence upper bound is also proportional to the diameter of the constraint set.

For unconstrained and non-differentiable problems (with Lipschitz continuity), a similar convergence guarantee towards a neighborhood of the Goldstein stationary point of the objective function is proved. A crucial step is to utilize the relationship between the Goldstein subdifferential of the objective function and the gradient of the Gaussian-smoothed objective function.

Lastly, the paper showcases a bunch of numerical evaluations of ZO-EG.

**Audience:**

Yes

**Broader Impact Concerns:**

Since this paper mostly focuses on the theoretical aspects of optimization algorithms, the Broader Impact Statement section seems unnecessary.

**Claims And Evidence:**

Yes

**Requested Changes:**

**Required Changes**

- Every point I listed above in the weaknesses should be resolved.

**Recommended Changes & Some Questions expected to be answered**

- As far as I know, the gradient approximation used in the paper is not that stable: the first-order Taylor expansion error remains. As a remedy, another approximation method is widely used: $\frac{f(z+\mu u) - f(z- \mu u)}{2\mu} Bu$. Why did the author not use this option? Was there any technical difficulty?
- Can a variance reduction scheme be conducted adaptively & iteration-dependently? Can we have any computational benefits by periodically increasing the number of sampled vectors?
- Minor typos I found in the main text are listed below:
    - Two lines below Eq. (2): Put a space (”For$k\in \mathbb{N}$…” → ”For $k\in \mathbb{N}$…”)
    - Definition 8, 9, 10: Erase the commas at the end of the math equations.
    - Corollary 2: redundant parenthesis in “$(L_1(f)h_2^2 - 2\rho$”
    - Eq. (20): Is it a typo? In “$\langle Q_\ell (z, a, F(\bar{z})), z-z^*\rangle…$”, should “$z-z^*$” become “$\bar{z}-z^*$”?
    - Eq. (23): Two equations must be exchanged with each other.

**Strengths And Weaknesses:**

**Strengths**

1. I find this paper interesting because it dives deeply into the ZO-EG algorithm under various problem settings, across differentiability of the objective function and the existence of the constraint set. Each problem setting might be able to become a single paper, but this paper covers all of them.
2. Remark 1 about scheduling the smoothing parameter is particularly interesting, having a practical implication.
3. Numerical examples are abundant, providing many examples to future researchers to evaluate their minimax optimization algorithms.

**Weaknesses**

1. All the results are not exactly the convergence result towards an $\epsilon$-stationary point. Instead, they only guarantee the near-convergence to a neighborhood of it. Even in the easiest setting, the unconstrained setting with Lipschitz continuous gradients, the exact convergence is not guaranteed in the paper. How optimal is it? Is the non-exact convergence inevitable?
    * I don’t think this is usual for a zeroth-order optimization algorithm (e.g., [1, 2, 3]).
2. Despite the non-exact convergence, the authors provide a remedy using a variance reduction scheme, by sampling many direction vectors (from a centered Gaussian) and taking the average of gradient oracles associated with the direction vectors $u$. I think this idea should be more analyzed. For instance, although the scheme helps reduce the radius of the convergent neighborhood by shrinking the variance, there must be a trade-off of sampling many direction vectors, let’s say, in a computational burden. If so, shouldn’t the convergence bound be described in the overall amount of computation?
3. Another concern is that the constrained case analysis relies on the constraint set's finite diameter. However, there are many examples of constraint sets with infinite volume in the real world, which may have $D_z = \infty$. I wonder whether the analysis can be extended to such cases.
4. The paper lacks discussions in terms of other optimality criteria discussed in minimax optimization and multi-player game literature (e.g., local Nash [4] and $\Phi$-equilibrium [5]).
5. How do you choose the (PSD) matrix $B$ for Gaussian smoothing? I find it difficult to find the choice of $B_1$ and $B_2$ in both the theorem statement and the proof.

---

[1] Balasubramanian & Ghadimi, “Zeroth-order Nonconvex Stochastic Optimization: Handling Constraints”, High-Dimensionality and Saddle-Points, arXiv:1809.06474

[2] Malladi et al., “Fine-Tuning Language Models with Just Forward Passes”, NeurIPS 2023

[3] Wang & Feng, “Convergence Rates of Stochastic Zeroth-order Gradient Descent for Łojasiewicz Functions”, arXiv:2210.16997

[4] Jin, Netrapalli, & Jordan, “What is Local Optimality in Nonconvex-Nonconcave Minimax Optimization?”, arXiv:1902.00618

[5] Cai et al., “On Tractable Φ-Equilibria in Non-Concave Games”, arXiv:2403.08171

---

> ### Author Response · Authors · 2025-07-29
> **Addressing the issues raised by the reviewer**
>
> Thank you for the precise summary and the valuable comments and suggestions. We have submitted a revised manuscript after addressing all the issues raised by the reviewers. The parts of the revised manuscript that underwent significant changes as compared to our original submission are highlighted in blue. We address your comments below.
>
> **Weaknesses**
> 1.  Please note that in the papers you mentioned, [2,3] considered solving a minimisation problem with an extra assumption on the structure of the objective functions (satisfying Polyak Lowasewicz inequality (PL)).  Also in [3], the gradient estimator in (3) is an average over \\(k\\) different directions (similar to the variance reduction technique). Moreover, [1] considers solving the unconstrained non-convex minimisation problem without any further assumptions on the structure of the objective function. But, they use a similar variance reduction technique. In each iteration, they have $m_k$ different sampled directions and use the average of the gradient estimator (2.4).  Here, we consider the min-max problem without any further assumptions like PL, which introduces further challenges. In the main body of the first submission, we use the algorithm with only 1 sampled direction in each iteration. Using the mentioned variance reduction technique (Appendix C in the first submission), similar to [1], the variance dependent term can go to zero with extra computational load. To see the results of using the variance reduction, please see the next response. For the constrained case, even in minimisation problems with extra structure assumptions, the variance dependent term can be seen in the literature. Please see [4,5,6].
> 2. You are correct that using the variance reduction technique introduces additional computational expense.  Based on the number of samples in each iteration, the bound on the number of iterations and number of function calls change. To avoid confusion, in the revised manuscript, we brought the variance reduction technique (Appendix C in the first submission) to Section 2 and added Theorems 2, 4, and 6 and Corollaries 4, 8, and 12 to Section 3, which explain the result in the case where the variance reduction technique is used. We illustrate the findings via a numerical example in Section 4.3. Moreover, Section 1.1 has also been revised accordingly.
> 3. Our analysis is only for compact convex constraint set (a common assumption in the literature, e.g., [1]). We have added Remark 5 to Section 3.2 in the revised manuscript and explained more about this assumption.
> 4. In the last paragraph of Section 3.4, we discussed and explained more on this issue.
> 5. You are correct. We chose $B_1$ and $B_2$ as identity in our first submission, as it is the common choice in the literature [1,2,3,4,5,6] and we failed to explicitly mention it in the original submission. However, your comment made us to consider generalising the results for any suitable choices of $B_{1}$ and $B_{2}$. Accordingly, the notation section is altered slightly in the revised manuscript and all the main results in Section 3 have been has been updated and now employ a general matrix $B = B^\ast \succ0$ instead of $B = I.$ We explained more on choice of $B$ after each main theorem, in Appendix C, and in Section 3.4 and illustrate it via a numerical example in Section 4.3. Moreover, we added extensions for $B$-invariant results in Appendix E.
>
> **Recommended Changes**
> 1. We explained more on this issue in the first and second paragraphs of Section 3.4 and illustrated it via a numerical example in Section 4.3.
> 2. Indeed, a variance reduction technique can be implemented adaptively. Please see Remark 2 in the revised manuscript.
> 3. The typos are fixed in the revised manuscript. In the fourth mentioned typology, it is not a typology and it is how we defined it. On the fifth mentioned typo, there is no exchange needed in (23) ((29) in the revised version) as to go from point $z_k$ to $\hat{z}_k,$ we need to use $s_k$ which is based on $G(z_k)$ and to go from point $\hat{z}_k$ to $z_k+1$, we need to use $\hat{s}_k$ which is based on $G(\hat{z}_k)$.
>
> [1] Balasubramanian and Ghadimi, “Zeroth-order Nonconvex Stochastic Optimization: Handling Constraints”, High-Dimensionality and Saddle-Points, arXiv:1809.06474.
>
> [2] Malladi et al., “Fine-Tuning Language Models with Just Forward Passes”, NeurIPS 2023.
>
> [3] Wang and Feng, “Convergence Rates of Stochastic Zeroth-order Gradient Descent for Łojasiewicz Functions”, arXiv:2210.16997.
>
> [4] Farzin and Shames, I. "Minimisation of Polyak-Łojasewicz Functions Using Random Zeroth-Order Oracles." 2024 European Control Conference (ECC). IEEE.
>
> [5] Maass et al. "Zeroth-order optimization on subsets of symmetric matrices with application to mpc tuning." IEEE Transactions on Control Systems Technology 30.4 (2021) .
>
> [6] Maass et al. "Tracking and regret bounds for online zeroth-order Euclidean and Riemannian optimization." SIAM Journal on Optimization 32.2 (2022).

---

### Review · Reviewer_jZZx · 2025-05-25

**Summary Of Contributions:**

This paper studies min-max optimization problems involving nonconvex-nonconcave functions, with a particular focus on the zeroth-order optimization setting, where only function values of the objective are accessible. To address this problem, the authors propose a zeroth-order extragradient method based on random Gaussian smoothing, and provide theoretical analysis for both unconstrained and constrained settings. In addition, they validate the effectiveness of the proposed method through numerical experiments.

**Audience:**

Yes

**Claims And Evidence:**

No

**Requested Changes:**

I would appreciate it if the authors could verify whether there are any misunderstandings in the points of concern I raised above, and provide clarifications or revise the manuscript accordingly as appropriate.

**Strengths And Weaknesses:**

The problem setting and motivation are clearly explained, and the related work is discussed to a sufficient extent, which I consider to be a strength of the paper. The empirical validation through numerical experiments also appears to be convincing. Moreover, I believe the topic of this research will be of interest to a broad audience.
That said, I have some concerns regarding the soundness of the theoretical analysis and the comparison with prior work.

First, I would like to raise a concern regarding the usage of results from (Nesterov & Spokoiny, 2017), which are cited at multiple points throughout the paper. In particular, the norm $\\| \cdot \\|$ used in that work differs from the one adopted in the submitted manuscript, and this discrepancy does not appear to have been properly accounted for. Specifically, as can be seen from Section 1.3 of (Nesterov & Spokoiny, 2017), the norm is defined as $\\| x \\| = \sqrt{x^\top B x}$ for some matrix $B$, whereas in the submitted paper, according to the "Notation" paragraph in Section 2, the norm $\\| x \\|$ refers to the standard Euclidean norm.

With this in mind, it seems that expressions such as $\\| u \\|^2$ in Equation (6), and the norms appearing in Equation (13) as well as in Lemmas 5 and 7, should be interpreted in terms of the $B$-induced norm rather than the Euclidean one. Furthermore, (Nesterov & Spokoiny, 2017) appears to define Lipschitz constants and other quantities with respect to the $B$-induced norm, which also needs to be carefully considered. However, I could not find evidence that this distinction has been taken into account in the stated assumptions or inequalities in the main results.

Next, in the Contributions section, the authors claim that their method finds an $\epsilon$-stationary point within $O(\epsilon^{-2})$ iterations. However, in Theorems 1 and 2, as well as Corollaries 1 through 4, the obtained upper bounds on the gradient norm appear to contain residual terms that depend on $\sigma$. This suggests that the bound does not vanish as the number of iterations increases, which casts doubt on whether the claimed convergence to an $\epsilon$-stationary point is actually achieved in the usual sense.

Given this, I am concerned about the validity of the statement in the Contributions section. It would be desirable for the authors to clarify whether this issue also exists in the related work they cite as comparisons. If those previous studies suffer from a similar limitation, it should be acknowledged; otherwise, the comparison may be misleading and would require appropriate qualification.

Third, as mentioned multiple times in the paper, zeroth-order methods are often designed to handle noisy feedback, i.e., they are commonly applied in stochastic optimization settings. However, it appears that the current analysis does not actually support such scenarios. An explicit clarification on this point would be helpful.

For example, the last sentence of page 1 states, *"Unlike first-order methods, ZO methods only require access to (often noisy) evaluations of the objective function,"* and the **Related Work** section refers to methods for *"stochastic min-max optimization problems with similar problem structures."* These statements may give the impression that the proposed method can handle stochastic settings with noisy function evaluations.

However, as I understand it, the gradient estimator defined in Equation (10) would suffer from high variance when the two function evaluations involved contain independent noise, which is typically the case in stochastic settings. In such cases, the variance would scale with $1/\mu$, potentially invalidating Assumption 1. This suggests that the current analysis does not cover the noisy evaluation setting.

If this is indeed the case, then directly comparing the results of this paper to prior work on stochastic ZO optimization may not be entirely fair, and the authors should clarify this distinction explicitly in the paper.

---

> ### Author Response · Authors · 2025-07-29
> **Addressing the issues raised by the reviewer**
>
> Thank you the precise summary and the valuable comments and suggestions. We have submitted a revised manuscript after addressing all the issues raised by the reviewers. The parts of the revised manuscript that underwent significant changes as compared to our original submission are highlighted in blue. We address your comments below.
>
> **Weaknesses**
>
> 1. You are correct. We chose $B_1$ and $B_2$ as identity in our first submission, as it is the common choice in the literature [1,2,3,4,5,6] and we failed to explicitly mention it in the original submission. However, your comment made us to think on generalising the results for any suitable choices of $B_{1}$ and $B_{2}$. Accordingly, the notation section is altered slightly in the revised manuscript and all the main results in Section 3 have been updated and now employ a general matrix $B = B^\ast \succ0$ instead of $B = I.$ We explained more on choice of $B$ after each main theorem, in Appendix C, and in in Section 3.4 and illustrate it via a numerical example in Section 4.3. Moreover, we added extensions for $B$-invariant results in Appendix E.
> 2.  You are correct. We remark that in the contribution section of the first submission, for all three settings (unconstrained, constrained, and non-differentiable), we claim that "we prove the convergence of the algorithm to a neighbourhood of the
> $\epsilon$-stationary point of $f$ in at most $O(\epsilon^{-2})$ iterations". Here, the word "neighbourhood" refers to the $\sigma$-dependent term in the bounds. Also, we claim that "The size of the neighbourhood of convergence in both settings depends on the variance of the ZO random oracle, which can be controlled using variance-reduction techniques (Appendix C in the first submission)." The $\sigma$-dependent term can go to zero using the variance reduction technique, but with extra computational load. Please note that the variance reduction technique is leveraged in the literature of non-convex optimisation to control the variance-dependent term [1]. Other works like [2,3,4] assume extra structure like Polyak-Lowasewicz (PL) on their objective function to overcome these challenges. To the best of our knowledge, this work is the first work on non-convex non-concave ZO minimax optimisation without any further assumptions like PL or strong concavity. We only rely on the existence of a weak solution to the problem (existence of a solution satisfying the weak MVI inequality). Thus, we can not compare the related works directly.
>
>     To avoid confusion, in the revised manuscript, we brought the variance reduction technique (Appendix C in the first submission) to Section 2 and added Theorems 2, 4, and 6 and Corollaries 4, 8, and 12 to Section 3, which explain the result in the case where the variance reduction technique is used. We illustrate the findings via a numerical example in Section 4.3. Section 1.1 has also been revised accordingly.
> 3. You are correct, and the current analysis does not cover the noisy evaluation setting. Please note that we do not claim that the results of this work include the case of noisy feedback anywhere in the first submission. We mentioned  "ZO methods only require access to (often noisy) evaluations of the objective function" in the introduction section just for completeness of the definition. Moreover, we clarified that if the results of related works are stochastic or deterministic in Section 1.2 and emphasised more on the fact that we have considered the deterministic case in Section 1 and the Abstract.
>
>     Please note that the forward approximation oracle is commonly used in the ZO optimisation literature and assumption of bounded variance of the random oracle even in the noisy feedback case is common and crucial in the literature of ZO stochastic optimisation. Using the forward approximation oracle the bound on variance will be scaled by $\frac{1}{\mu^2}$ but still it is bounded as long as the variance of the noise is bounded. We explained more on this issue in the first and second paragraph of Section 3.4 and illustrated it via a numerical example in Section 4.3.
>
> [1] Balasubramanian and Ghadimi, “Zeroth-order Nonconvex Stochastic Optimization: Handling Constraints”, High-Dimensionality and Saddle-Points, arXiv:1809.06474
>
> [2] Malladi et al., “Fine-Tuning Language Models with Just Forward Passes”, NeurIPS 2023
>
> [3] Wang and Feng, “Convergence Rates of Stochastic Zeroth-order Gradient Descent for Łojasiewicz Functions”, arXiv:2210.16997
>
> [4] Farzin and Shames, I. "Minimisation of Polyak-Łojasewicz Functions Using Random Zeroth-Order Oracles." 2024 European Control Conference (ECC). IEEE.
>
> [5] Maass, Alejandro I., et al., 2021. "Zeroth-order optimization on subsets of symmetric matrices with application to mpc tuning." IEEE Transactions on Control Systems Technology 30 (4)
>
> [6] Maass, Alejandro I., et al., 2022. "Tracking and regret bounds for online zeroth-order Euclidean and Riemannian optimization." SIAM Journal on Optimization 32 (2)

---

### Review · Reviewer_zS4u · 2025-07-22

**Summary Of Contributions:**

This submitted manuscript has the following key points:

(1) considering zeroth-order (extra)gradient based method to solve a min-max optimization problem;

(2) the first-order stationary criterion authors claim to focus on is Goldstein stationary point

(3) assumption of Lipschitz continuity for both objective and its derivative

(4) proof of convergence relying on Gaussian smoothed version of an integrable function and the Minty variational equality

**Audience:**

Yes

**Claims And Evidence:**

No

**Requested Changes:**

1. Define "locally Lipschitz" in definition 4 on page 5

2. Define "extra"gradient

3. Add citation for "Rademacher's theorem" mentioned under definition 4 on page 5

4. Show deduction of inequality (13) in subsection 3.1 under assumption 1 on page 8

5. On page 28, answer the following questions:

- It is implicitly assumed that $\sqrt{\frac{2\rho}{L_1}} \le \frac{h_1}{2} \le h_2 \le \frac{1}{2L_1} $. There is no guarantee that this will hold because the relationship between $\rho$ and $L$ is undetermined.

- Supplement missing subscript $G_{\mu}$ in inequality (48)

- In inequality (48), last row, the term $h_1 G_{\mu}(z_k)$ comes out from nowhere

- In inequality (51), I do not think the left hand side should be $ || F_{\mu} ( \hat{z}_{k} ) || $. It does not follow from $ \mathbb{E}_{u_k} || G_{\mu}(\hat{z}_k) ||^2 \ge || F_{\mu}(\hat{z}_k) || $

- $\xi_k$ and $\hat{\xi}_k$ are not independent, therefore the inner product terms in equation (50) do not vanish under expectation.

- Most importantly, Lines 5 and 8 in Algorithm 1 on page 7 clearly entail a projection operator. The domain $\mathcal{Z}$ is not clearly defined in the statement of the theorem, either. This projection operator is directly ignored in the proof process of this work. The claims, under inequality (45), that $z_k - z_{k+1} = h_2 G_{\mu}(\hat{z}_k)$ and $z_k - \hat{z}_k = h_1 G_{\mu} (z_k)$ do not hold.

**Strengths And Weaknesses:**

Strengths:

- Authors tries to set up a clear structure of the paper

Weaknesses:

- Several technical aspects should be further clarified

---

> ### Author Response · Authors · 2025-07-29
> **Addressing the issues raised by the reviewer**
>
> Thanks for your summary and valuable comments. We point out that your summary of contributions is not altogether accurate. We clarify a misconception in the statements here. Specifically, we should note that the stationarity criterion we used in Sections 3.1 and 3.2 is convergence to $\epsilon$-stationary points, and only in Section 3.3 (non-differentiable case), we focus on convergence to $(\delta,\epsilon)$-Goldstein stationary point. The assumption of Lipschitz continuity of the gradients of the function is used in Sections 3.1 and 3.2, and the assumption of Lipschitz continuity of the function is only used in Section 3.3.  We have submitted a revised manuscript after addressing all the issues raised by the reviewers. The parts of the revised manuscript that underwent significant changes compared to our original submission are highlighted in blue. We address your comments in the following.
>
> **Requested Changes**
> 1. We have modified Definition 1 to include the definition of locally Lipschitz.
> 2. We have added an explanation for the word "extra" to the introduction section before Section 1.1.
> 3. A reference is added after Definition 4.
> 4. Inequality (13) (inequality (17) in the revised version) follows from the fact that $E_u[G_\mu(z)]=F_\mu(z).$ More details are added in the revised version after (17).
> 5. Your next comments on page 28 of the original submission are regarding the proof of Theorem 1, which is on page 37 in the revised submission.
> -   The relationship between $\rho$ and $L$ is given in Definition 10 ($\rho\in[0,\frac{1}{8L})$ in first and $\rho\in[0,\frac{1}{8L\kappa^2})$ in the revised version), which guarantees $\sqrt{\frac{2\rho}{L}}\leq h_2\leq \frac{h_1}{2}\leq\frac{1}{2L}$ in the first and $\sqrt{\frac{2\rho}{L\overline{\lambda}\underline{\lambda}\kappa}}\leq h_2\leq \frac{h_1}{2}\leq\frac{1}{2L\overline{\lambda}\kappa}$ in the revised submission hold. We have added more details to the proof of Theorem 1 before (59).
> - Thanks. The typo is fixed.
> - The term $h_1G_\mu(z_k)$ in (48) ((59) in the revised version) is not coming out of nowhere. It follows
>  $$-h_2\\langle\\hat{\xi}_k,\\hat{z}_k-z^\\ast\\rangle = -h_2\\langle\\hat{\\xi}_k,z_k - h_1 G\_\\mu (z_k) -z^\\ast\\rangle = h_2\\langle\\hat{\\xi}_k,z^\\ast-z_k + h_1 G\_\mu (z_k) \\rangle$$
> and this is due to $\hat{z}_k=z_k-h_1 G\_\mu (z_k)$ according to ZO-EG algorithm line 5 and the fact that the problem is unconstrained, i.e., $\\text{Proj}\_\mathcal{Z}(z)=z$. More explanation is added in the revised version after (59).
> - We are not sure about what you meant by this comment, but we double-checked our calculations. In the inequality above (51) ((64) in the revised submission) we show $E_{u_k}[\\|G\_\mu(\hat{z}_k)\\|^2]\geq\\|F\_\mu(\hat{z}_k)\\|^2.$  Also, we know $\frac{h_2}{4}$ is positive and thus after taking the expectation from (48) ((62) in the revised version) we can lower bound $E\_{u_k}[\\|G\_\mu(\hat{z}_k)\\|^2]$ by $\\|F\_\mu(\hat{z}_k)\\|^2$ using the mentioned inequality and reach (51) ((64) in the revised submission). We added more explanation before (63) in the revised version.
> - You are right that $\xi_k$ and $\hat{\xi}_k$ are not independent but $\hat{\xi}_k$ depends on $u_k$ ($E\_{u_k}[\hat{\xi}_k]=0$) and $\xi_k$ does not ($E\_{u_k}[\xi_k]=\xi_k$). Thus, taking the expectation with respect to $u_k$ first, which is the last sampled random variable, makes the expectation of the inner product zero. The same technique (Taking the expectation with respect to the last sampled random variable and then the history) has been leveraged commonly before in the literature, e.g., Section 7 before (67) of [2]. We added more explanation after (64) in the revised version.
> - In the statement of Theorem 1, it is said suppose that Assumption 2 is satisfied and in Assumption 2, we have declared that $\mathcal{Z}=\mathbb{R}^d$ (in the revised version  $\mathcal{Z}=\mathbf{Z}$ where $\mathbf{Z} = (\mathbb{R}^d,\|z\| = \langle z,Bz\rangle^{\frac{1}{2}})$) as we consider the unconstrained case in Section 3.1. Moreover, we did specify that $\mathcal{Z}=\mathbb{R}^d$ before (43) in the first submission. Thus in this setting $\text{Proj}_{\mathcal{Z}}(z)=z.$ Hence, $z_k-z\_{k+1} = h_2G\_\mu(\hat{z}_k)$ and $z_k-\hat{z}_k=h_1G\_\mu(z_k)$.
>
>     Please note that we have presented the ZO-EG algorithm in the general form as we analyse the min-max problem in 3 different settings (differentiable objective function with unconstrained and constrained problem and non-differentiable objective function with unconstrained problem). The projection operator plays its role in the constrained settings, which is analysed in Section 3.2. We have added more explanation before (54) in the revised version.
>
> [1] Korpelevich, G. M., 1977. “Extragradient method for finding saddle points and other problems,” Matekon, 13(4)
>
> [2] Nesterov, Y. and Spokoiny, V., 2017. Random gradient-free minimization of convex functions. Foundations of Computational Mathematics, 17(2)

---

### Decision · Action_Editor_in8p · 2025-09-03

**Recommendation:** Accept as is

**Audience:**

Yes

**Audience Explanation:**

The considered problem (zeroth-order min-max optimization) is of interest to the ML community, and papers on this topic regularly appear in top ML conferences. The reviewers also acknowledged that the topic is relevant to TMLR's audience.

**Claims And Evidence:**

Yes

**Claims Explanation:**

The authors provide detailed convergence proofs and extensive numerical experiments.